# Uniquely preserved gut contents illuminate trilobite palaeophysiology

Petr Kraft[1,5], Valéria Vaškaninová[1,5], Michal Mergl[2], Petr Budil[3], Oldřich Fatka[1] & Per E. Ahlberg[4 ✉]

Trilobites are among the most iconic of fossils and formed a prominent component of marine ecosystems during most of their 270-million-year-long history from the early Cambrian period to the end Permian period[1]. More than 20,000 species have been described to date, with presumed lifestyles ranging from infaunal burrowing to a planktonic life in the water column[2]. Inferred trophic roles range from detritivores to predators, but all are based on indirect evidence such as body and gut morphology, modes of preservation and attributed feeding traces; no trilobite specimen with internal gut contents has been described[3,4]. Here we present the complete and fully itemized gut contents of an Ordovician trilobite, *Bohemolichas incola*, preserved three-dimensionally in a siliceous nodule and visualized by synchrotron microtomography. The tightly packed, almost continuous gut fill comprises partly fragmented calcareous shells indicating high feeding intensity. The lack of dissolution of the shells implies a neutral or alkaline environment along the entire length of the intestine supporting digestive enzymes comparable to those in modern crustaceans or chelicerates. Scavengers burrowing into the trilobite carcase targeted soft tissues below the glabella but avoided the gut, suggesting noxious conditions and possibly ongoing enzymatic activity.

Siliceous nodules, nicknamed 'Rokycany Balls', weather out of shales of the Šárka Formation (Darriwilian, Middle Ordovician, approximately 465 million years ago (Ma)) of the Prague Basin, Czech Republic[5,6] and provide abundant well-preserved three-dimensional (3D) fossils without secondary deformation due to their very early diagenetic origin (Supplementary Information). One was found to contain a complete specimen of the infrequent trilobite *Bohemolichas incola* with preserved gut content visible in the exfoliated parts of the thoracic axis and occipital ring (Fig. 1a). This specimen was selected for investigation by propagation phase–contrast synchrotron microtomography (PPC-SRμCT) at the European Synchrotron Radiation Facility (ESRF) in Grenoble, France (Extended Data Table 1). The specimen comprises an almost undisturbed, articulated exoskeleton. Only the librigenae are dislocated laterally to the left (Fig. 1b and Extended Data Fig. 1) and the almost in situ conterminant[7] hypostome is slightly displaced below the glabella (Fig. 1c and Extended Data Fig. 1b). In lateral view, the sixth segment of the thorax is markedly deflected from the normal position causing a slight vaulting of the central portion of the thorax (Fig. 1d and Extended Data Fig. 1c).

## Gut contents

Small fossils and fragments of shells are densely distributed exclusively along the exoskeletal axial lobe reflecting an almost continual infill of the digestive tract (Figs. 1c and 2a, Extended Data Fig. 1a,b and Supplementary Fig. 1). The largest concentration is situated in the cephalic region between the glabella and the hypostome where it forms an arc connecting two wide clusters, one pressed to the hypostome and the other close to the surface of the central and posterior parts of the glabella (Fig. 3a–d and Extended Data Figs. 1d,e and 2). The clusters are confined within the vaulted regions of the glabella and hypostome. The displacement of the hypostome deformed the space below the glabella, affecting the proximal part of the infilled digestive tract, which is compressed and shifted (Extended Data Fig. 3). A linear accumulation of shell remains along the midline of the thoracic and pygidial axes, pressed to the ventral side of the vaulted axis, is dorsoventrally flattened resulting in an oval cross-section (Fig. 2a). This postcephalic accumulation is almost continuous, though three indistinct clusters can be discerned (Extended Data Fig. 1a,b).

The most abundant determinable elements of the digestive tract infill are ostracods (Fig. 2b) characterized by a typical shape of the margin, vaulting of valves and occasional surface ornamentation. Although fragments dominate, several complete valves (Fig. 2e and Extended Data Fig. 4), randomly distributed along the entire digestive tract, can be identified as different early instars of *Conchoprimitia osekensis*[8]. One hyolith conch near the posterior end of the accumulation represents most likely the genus *Elegantilites* with a moderate vaulting of the dorsal side, becoming steeper laterally, and a slightly convex ventral side in cross-section (Fig. 2d). A fragment of a hyolithid operculum with typical cardinal processes is also present (Fig. 2d).

[1]Institute of Geology and Palaeontology, Charles University, Prague, Czech Republic. [2]Centre of Biology, Geosciences and Environmental Sciences, University of West Bohemia in Plzeň, Plzeň, Czech Republic. [3]Czech Geological Survey, Prague, Czech Republic. [4]Department of Organismal Biology, Uppsala University, Uppsala, Sweden. [5]These authors contributed equally: Petr Kraft, Valéria Vaškaninová. ✉e-mail: per.ahlberg@ebc.uu.se

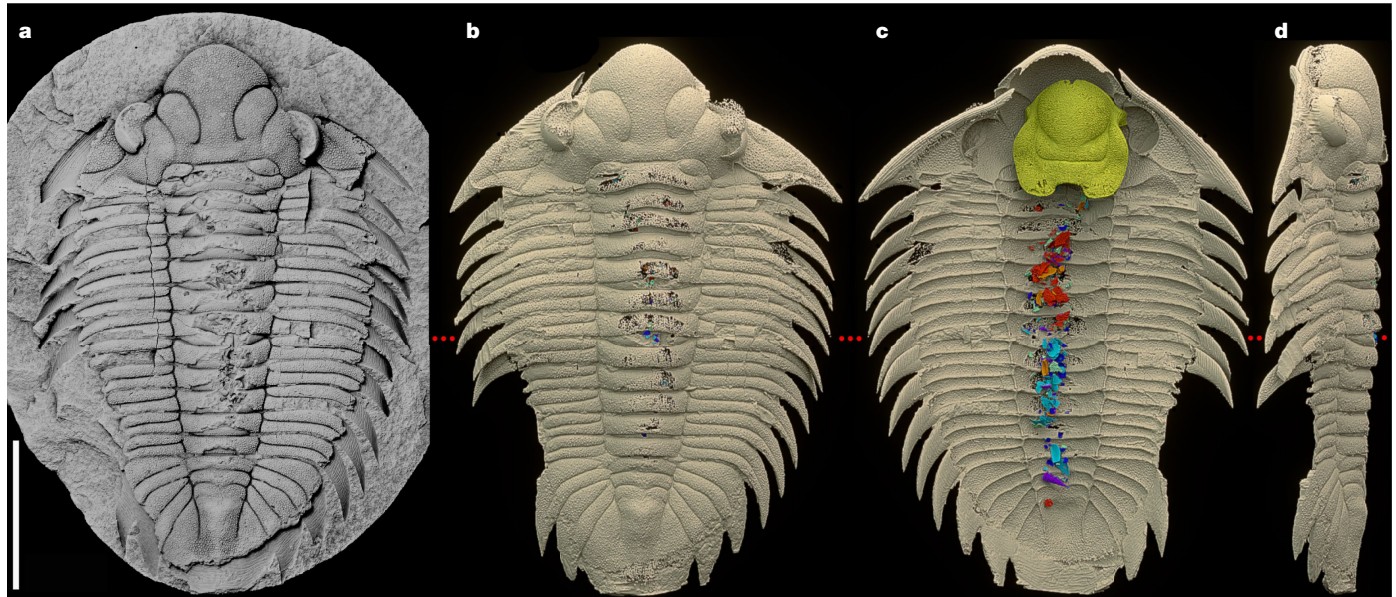

**Fig. 1 | *Bohemolichas incola* (Barrande, 1872). a**, Internal mould of specimen (inventory no. 8) in the nodule (coated with ammonium chloride). **b–d**, Scan model of the same specimen in dorsal (**b**), ventral (**c**) and left lateral (**d**) view. Exoskeleton in cream, hypostome in gold, digestive tract contents in shades of red and blue. The red dotted line indicates an anomalous position of segments five and six. Voxel size, 11.35 μm (applies for all figures and extended data). Scale bar, 10 mm.

Remains belonging most likely to a stylophoran echinoderm are scattered through the anterior part of the digestive tract and occur in the cephalic clusters and the anterior cluster in the gut. Flat plates with a regular dense reticulation on their surfaces, some very thin with a sieve-like perforation by circular pores, apparently represent central plates (Fig. 2c and Extended Data Fig. 5). Accompanying girder-like or vertebra-like plates, which are more complex and massive, can be identified as marginal plates (Fig. 2b and Extended Data Fig. 5a–c). The difference in robustness and resistance was apparently the reason the central plates are fragmented while the marginals are complete. The association of central and marginal plates probably represents a single small disintegrated individual of this echinoderm, implying that the trilobite consumed at least one half of the digestive tract content in one place and time, thus indicating a fast-occasional feeding behaviour. Large fragments of thin-walled shells, most probably bivalves, occur predominantly in the glabellar cluster and two posterior clusters in the gut (Fig. 2b). Indeterminable shell fragments show a regular distribution along the gut (Fig. 2b).

## Reconstructing the digestive tract

Inferred from its preserved content, the digestive tract was spacious, and the intestine had a wide diameter. The accumulations of ingested shells in the cephalic region form two ventriculi positioned dorsoventrally above each other between the hypostome and glabella (Fig. 3 and Extended Data Figs. 1e, 2 and 3). This part of the digestive tract represents the foregut composed of a two-chamber stomach (crop). Based on the extent of the clusters of shell fragments, the dorsal ('glabellar') ventriculus was larger than the ventral ('hypostomal'). The dorsal ventriculus is located in the central and posterior parts of glabella (Fig. 3b and Extended Data Fig. 2); the ventral ventriculus was secondarily shifted but its original position can be inferred as more antero-ventral in relation to the dorsal ventriculus. The ventriculi were interconnected anteriorly by a bent segment. Secondarily, the connecting segment was dislocated to the left and the originally sack-like ventriculi were taphonomically flattened, resulting in a horizontal, U-shaped transverse cross-section of the foregut. Posterior to the dorsal

ventriculus, the digestive tract became slightly narrower, indicating the midgut comprising a short segment below the occipital ring at the posterior edge of the cephalon and first two pleural rings. The width of the intestine decreases considerably below the ring of the third thoracic segment, probably indicating the beginning of the hindgut (Fig. 2a and Extended Data Fig. 1a). The last gut infill fragments are located in the middle (sagittal) part of the pygidial axis, which locates the anus behind the axial lobe termination (Fig. 1c).

Besides the lobate glabella, another indication of the considerable volume of the foregut is the original life position of the hypostome, as reconstructed from 3D models by fitting the corresponding edges, forming a vaulted ventral chamber of the cephalon and a bulbous space of the axial cephalic lobe (Fig. 5 and Extended Data Fig. 6b). The locomotory appendages (endopods) of the trilobite must have been long and sturdy enough to raise the vaulted ventral side of the body off the substrate while moving. Such long locomotory endopods have been recorded in some exceptionally preserved Ordovician trilobites[9].

A two-chambered ventriculus is known in extant arthropods. Decapod crustaceans possess a so-called gastric mill, composed of solid circular teeth, situated in the narrowing separating the cardiac and pyloric stomach[10]. In the extant horseshoe crab, a similar configuration is observed, composed of a ventral proventriculus (crop) for storing food items, followed by a ventriculus (gizzard) for mechanical grinding of food items with hard shells[11]. The presence of such a grinding apparatus in *B. incola* was tested by comparing the relative size of fragments in both ventriculi and the gut. However, the results of the analysis show a statistically insignificant difference (Supplementary Table 1). The ventral ventricular chamber of *B. incola* could represent a collecting space for ingested material before its passage into the dorsal chamber and gut for final digestion without subsequent grinding.

Blurry aggregates of porous and grainy matter occur very frequently in the nodules of the Šárka Formation, usually associated with body fossils and ichnofossils. The position and extent of this material allow us to interpret it as a diagenetic product of substances produced by tissue decay[12]. Informally referred to as 'crumble', after its appearance, it provides important clues to the position, size and morphology of the

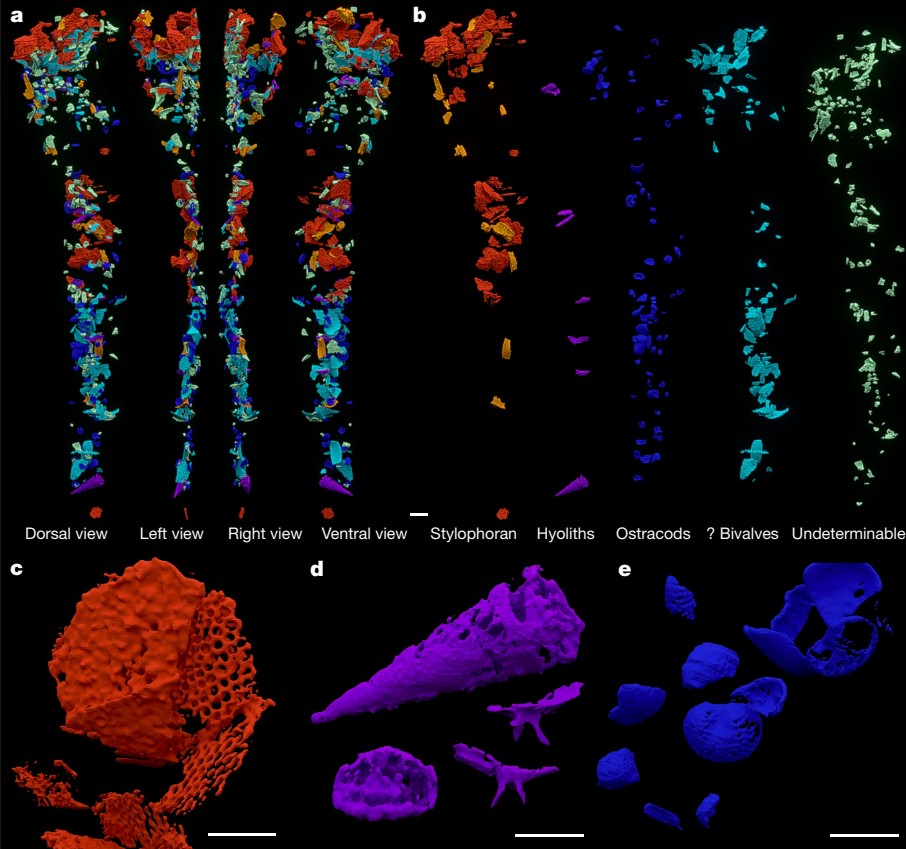

Dorsal view    Left view    Right view    Ventral view    Stylophoran    Hyoliths    Ostracods    ? Bivalves    Undeterminable

**Fig. 2 | Digestive tract content composition. a**, Scan model of the full digestive tract infill. **b**, Digestive tract infill separated by content in series of dorsal views (stylophoran marginal plates in yellow, stylophoran central plates in red). **c**, Detail of stylophoran echinoderm central plates. **d**, Detail of hyolithid conch in lateral view (top), cross-section (bottom left) and a fragment of hyolithid operculum with cardinal processes in different views (bottom right). **e**, Detail of ostracod shells. Scale bars, 1 mm (**a**,**b**), 500 μm (**c**–**e**).

soft body parts buried with the exoskeleton. In the thorax of our specimen, the 'crumble' is limited to the axial part reflecting the decaying tissue of the gut (Fig. 4c and Extended Data Figs. 2–3 and 7–9). However, below the occipital ring at the posterior edge of the cephalon and two adjacent thoracic anterior axial rings, it spreads into lobe-like extensions connected directly to the gut. These are indications of digestive caecae (or diverticula), of which there are typically three pairs in the anterior midgut of lichid trilobites[13] (Fig. 4c,d). Remains of laterally extended 'crumble' are visible in other axial rings along the entire length of the thorax, possibly representing traces of midgut glands or other less extensive diverticula (Extended Data Fig. 8a,b). The almost entirely filled digestive tract combined with the 'crumble' reflects its simple J-like shape and allows the tract to be reconstructed (Fig. 5).

## Feeding strategy

The studied specimen of *B. incola* represents a specific feeding adaptation on organic remains including shells and has, so far, no parallel either in the coeval fossil communities or among the trilobites. All identified fragments inside the digestive system belong to benthic invertebrates possessing calcium carbonate shells (Fig. 2). They are represented by small taxa (ostracods), small specimens of taxa producing larger individuals (hyoliths), fragments of larger, thin-walled shells (probably bivalves), or specimens composed of small elements that easily become disintegrated (echinoderm plates). The diet composition indicates a limited ability of the trilobite to break firm or robust inorganic shells, based on the absence of an assemblage of fragments in the intestine that can be securely identified as crushed pieces originating from a single shell. This aligns with the sclerotised nature of trilobite gnathobases that were used for food processing/mastication (although some trilobites, for example, *Redlichia*, are considered shell crushers based on the presence of reinforced gnathobasic spines[14]). Generally, the more fragile thin-shelled elements are fragmented, whereas the solid elements are complete. The size range of vertebra-like marginal plates of stylophorans indicates a preference for small ingestible solid elements (Fig. 2b and Extended Data Fig. 5). This feeding mode is comparable to extant xiposurid chelicerates[11] in many aspects, especially considering the incorporation of gnathobases[15].

Animals feeding by crushing shells, so-called durophagous predation, have a special position in ecosystems. Durophagy requires special morphological adaptations in the mouth region or appendages that provide the organism with the necessary biting force to overcome the physical limitations of hard tissues[16]. The primary aim is to crush open the shell or exoskeleton, to reach the soft, digestible tissues. The ingestion of shell fragments is undesirable and usually occurs in small quantities. The continuous accumulation of hard particles throughout the intestine of *B. incola* reflects a different feeding strategy (in comparison to ref. 17). The studied trilobite was polyphagous with a low degree of food particle preference in their positive frequency-dependent proportion[18], which means that the food selection was based on size and shell resistance rather than taxonomical composition. By contrast, durophages usually specialize on a limited variety of shelly animals. The non-selective feeding behaviour of *B. incola* suggests that it was predominantly an opportunistic scavenger, because predatory arthropods are usually selective in their diet preferences[18,19]. In summary, *B. incola* can be considered as a light crusher and chance feeder

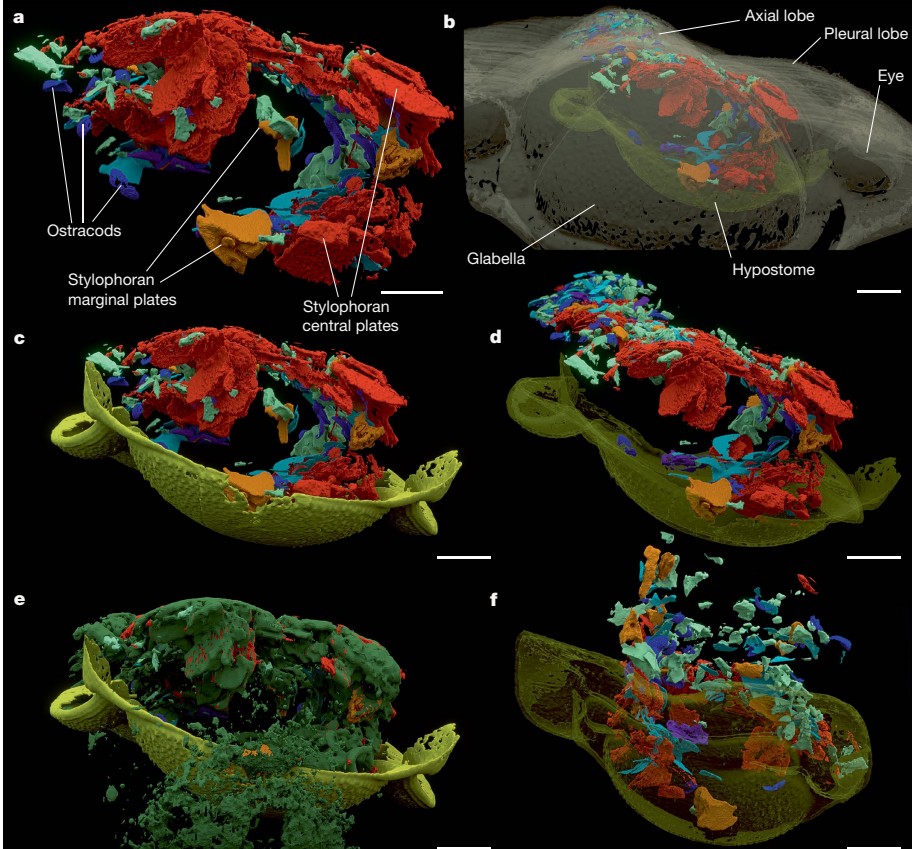

**Fig. 3 | Details of contents in the anterior digestive tract. a–e**, Anterior oblique view of the head region: ventriculi (**a**); with transparent exoskeleton and hypostome (**b**); with hypostome (**c**); with hypostome transparent (**d**); with hypostome and 'crumble' (**e**). **f**, Posterior oblique view of the head region, hypostome transparent. Colour coding as in Fig. 1, 'crumble' in dark green; labelling in **a** and **b** for orientation. Scale bars, 1 mm.

hoovering up dead or living animals that were either easily disintegrated or small enough to be swallowed whole.

## Food processing and digestive physiology

Digestion of food comprising organic tissues swallowed along with the associated inorganic shells is highly specialized and requires particular adaptations but offers a food supply with limited competition. Certain peculiarities of the dorsal exoskeleton of *B. incola* and other lichid trilobites may represent adaptations to accommodate an enlarged and specialized digestive tract. These include the distinctive glabellar morphology, with an inflated, often dorsally strongly vaulted and anteriorly expanded frontal glabellar lobe and a unique configuration of the other glabellar lobes (Fig. 1b; compare with a different morphology related to a dissimilar digestive tract in a dalmanitid trilobite[20]). The hypostome of *B. incola* is relatively flat, only with a slightly protuberant antero-central region (Extended Data Fig. 6a). However, its tilted life position (Fig. 5 and Extended Data Fig. 6b) expanded the space for the two-chambered ventriculus associated with musculature while covering and protecting the foregut. The blunt end of the pygidial axis (Fig. 1b,d), frequently occurring in lichid trilobites[21] may represent a further adaptation, allowing the passage of large undigested particles through the anal opening, including the necessary space for extensor muscles.

Adequate enzymatic support, such as a hepatopancreas and other digestive glands[22], is required for efficient energy extraction from a diet rich in indigestible fragments[23]. The lobate regions on the sides of the glabella (for example, specific bullar lobes; Fig. 1b), unknown in other trilobite groups[21], probably housed such glands, as they are close to the stomach and the anterior midgut[22]. External organs for manipulation and mechanical grinding of inorganic shelly material must also have been present. Gnathobases are known to have been used for this purpose in trilobites[14,24]. In addition, the morphology of the relatively robust hypostome (Extended Data Fig. 6a) suggests its possible participation in food processing, as a functional analogue to the gastric mill or gizzard of extant arthropods. In this interpretation, the wide, double-walled posterior margin of the hypostome would have operated as a cutting edge, with the inward midline curvature forming an efficient crusher. Even the prominent rough sculpture of the hypostome could have been used in food processing[24]. However, the associated limb specializations required for this functional interpretation have not been fully demonstrated[25].

The clusters of shell fragments below the axis are distributed in indistinct faecal pellets produced by peristalsis in the gut. They are long (up to five thoracic segments) and their width indicates that the gut had a large diameter or was extensible (Fig. 2a and Extended Data Fig. 1a,b). A disordered orientation of the shells or fragments of anisotropic shapes indicates the presence and essential role of a cover of the large pellets, such as a peritrophic membrane, protecting the inner surface of the gut.

The presence of undissolved sharp-edged calcareous shells in all parts of the gut shows that the digestive tract had an alkaline or almost pH-neutral inner environment; in an acid environment the shells in the posterior part of the gut would have become noticeably etched. This was functionally important because the dissolution of a large amount of calcareous shells in an acid-gut environment would lead to an ion imbalance of the organism and could cause a high concentration of extracellular $Ca^{2+}$ (hypercalcaemia). This unusual piece of evidence has far-reaching implications for the digestive physiology of the trilobite,

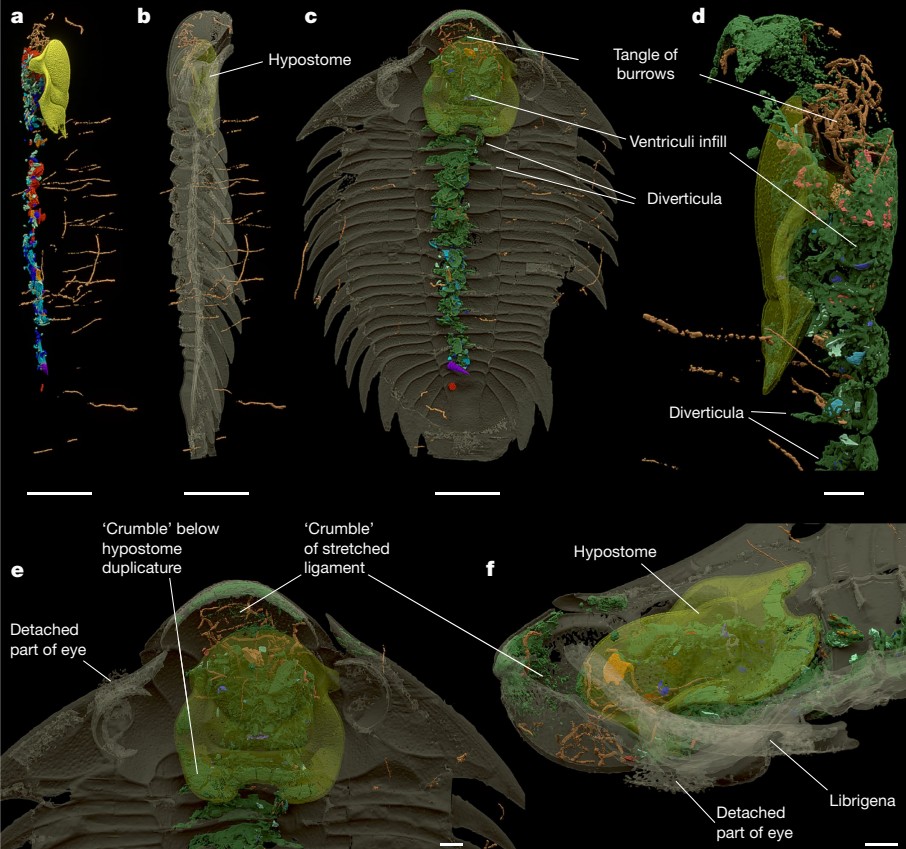

**Fig. 4 | Associated ichnofossils. a,b,** Distribution of trace fossils in right lateral view with hypostome (**a**) and transparent exoskeleton (**b**). **c,** Ventral view with 'crumble'. **d–f,** Details of head region in left lateral (**d**); ventral (**e**) and oblique ventrolateral (**f**) views. Colour coding as in Fig. 1, trace fossils in brown, exoskeleton and hypostome transparent. Scale bars, 5 mm (**a–c**), 1 mm (**d–f**).

as digestive enzymes are highly pH specific. A comparison with extant arthropods, important for anchoring the palaeophysiological interpretation in real biological systems, proves surprisingly challenging. The digestive tract of recent arthropods has been studied in detail with respect to its morphology, anatomy, food content, enzymatic processes and other features (for example, for decapod crustaceans[22,26]). However, there are unexpectedly few data on the pH values and dynamics inside their digestive tract, even though they directly influence all the mentioned characteristics. The discussion therefore has to be limited to a few model organisms.

Relevant data exist for the mud crab (*Scylla serrata* and related taxa) because of its economic importance, and the horseshoe crab (*Limulus polyphemus* and related taxa) because of its biomedical importance. As decapod crustaceans and chelicerates respectively, they bracket both major lineages within the euarthropod crown group[27]. Trilobites are generally agreed to fall within this bracket[28]. The gut pH is usually measured indirectly through the optimum enzyme activity and depends on the stage of digestive processes. *Scylla*[22,29–31] and *Limulus*[15,32,33] both show similar gut pH values ranging from slightly acid to alkaline, with a nearly neutral optimum, allowing them to digest a wide range of food items. The compatibility of these data with the inferred gut pH of *Bohemolichas* suggests that a near-neutral pH may be ancestral for the euarthropod crown group as a whole.

Different stages of digestive tract infill by food and other particles were recorded in mud crabs, ranging from an empty to a gorged intestine[31], which can partly be a seasonal factor[30,31]. However, the moulting of arthropods is often preceded by a preparatory phase when the digestive tract is swollen and expanded to press on the internal organs, to push off the carapace. The digestive tract is often filled by air and/or by water (see summary by Ayali[34]). The studied trilobite specimen displays an anomalous position of the fifth and sixth thoracic segments, a slight vaulting of the central part of the thorax and partly loose librigenae, corresponding to an exuvial configuration documented in detail in other trilobite groups[35,36]. The combination of these features with the gorged digestive tract may be interpreted as a premoulting phase initiated by rupture inside the thorax through an expanded intestine. We suggest that the feeding behaviour of the trilobite may have resembled the corresponding life cycles of modern crustaceans: most of time, the intestine was empty or moderately filled, with occasional and swift overfeeding actions linked to specialized physiological requirements[37].

## Postmortem events

The exceptional preservation of the specimen reflects its unusual taphonomic history. As mentioned in the Introduction, the siliceous nodules from the Šárka Formation appear to have formed through very early diagenesis at a shallow depth in the sediment[38,39], resulting in this instance in the trilobite being encased in hardened sediment before its soft tissues fully decayed. The specimen was buried shortly before or after death, lying on its back. Minor postmortem deformation is manifested by the hypostome and gut displacements (Supplementary Information). The hypostome was originally connected to the antero-ventral side of the cephalon with a ligament. This tissue was stretched and torn due to the hypostome dislocation as clearly marked by a thin, discontinuous layer of 'crumble' (Fig. 4e,f and Extended Data Figs. 7–9).

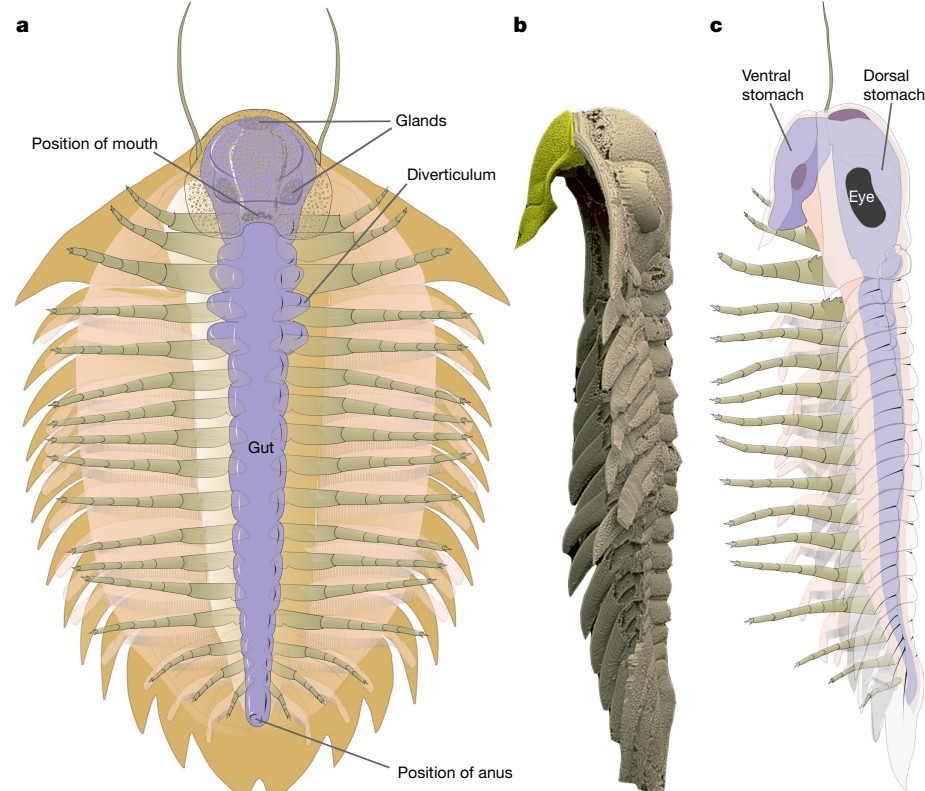

**Fig. 5 | Reconstruction of the digestive tract of *Bohemolichas incola*. a**, In ventral view. **b,c**, Left lateral view of scan model of exoskeleton with hypostome reconstructed in life position (**b**) and reconstruction of the digestive tract (**c**). Locomotory (including spines) and respiratory appendages suppressed for clarity. Hypostome in **a** and **c** is transparent; exoskeleton in **c** is transparent.

After burial in the sediment, but before the nodule formed, the trilobite was subjected to scavengers that produced slender vertical burrows extending down from the then-sediment surface to the carcase (Fig. 4 and Extended Data Figs. 7–9). The burrows are essentially identical in size and characteristics, pointing to identical producers. The distribution of burrows indicates not only the preferred targets but also areas that were strictly avoided (Fig. 4 and Extended Data Figs. 7–9). In other specimens from the Šárka Formation nodules, scavenging activity is often related to the 'crumble' representing decaying tissues[12], but that is not the case here. There was no feeding activity in the thorax or the pygidium. The only area of intensive feeding is indicated by a tangle of traces in the lobate anterior part of the glabella, in front of the cephalic food clusters representing the ventriculi (Fig. 4d and Extended Data Figs. 8e and 9). Here, an individual reaching the cephalic region found and systematically scoured a nutrient target. It is noteworthy that these burrows are situated anterior to the digestive system, outside the massive 'crumble', reflecting the decaying digestive tract. The cluster is situated inside the glabellar lobe cavity in the space between the stretched hypostomal ligament and the anterior part of the ventriculi, where it touches both structures but does not go through or inside them respectively (Fig. 4f and Extended Data Figs. 8e and 9a); it may have targeted glandular tissue. The digestive tract had a high potential to be consumed[12], but the trace makers entirely avoided the digestive system and targeted neither the intestine content nor the decaying tissue of the digestive tract (Extended Data Fig. 8a,b). This avoidance strongly suggests inhospitable conditions, possibly involving residual enzymatic activity throughout the entire tract. The distinct selectivity of targets by the scavengers and the lack of exit tracks directly support the model suggesting a rapid formation of the Šárka Formation nodules[39].

## Conclusion

The described specimen of *Bohemolichas* provides by far the most detailed source of information to date concerning the diet and the feeding mode of trilobites. This information includes indirect but robust evidence for a high-pH gut environment, aligning *Bohemolichas* with extant crustaceans and xiphosurans, and suggesting that such a digestive physiology may be primitive for the euarthropod crown group. It appears to have been an indiscriminate feeder on small, shelly, benthic invertebrates, most likely by scavenging rather than active hunting. *Bohemolichas* gives a unique glimpse of the role of lichid trilobites in an Ordovician marine ecosystem and provides evidence for the great antiquity of pH-neutral digestive physiology in arthropods.

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

# Article

## Methods

The specimen was scanned using PPC-SRμCT at the ID19 beamline of the ESRF in Grenoble, France, as a part of the proposal ES673 (for scan parameters see Extended Data Table 1). The reconstructed volumes were converted into stacks of 16-bit TIFF images. The scan data with voxel size of 11.35 μm were segmented in Mimics Research 19.0 (Materialise) software. Rendering was done in Blender 2.79b on black background which enhances the 3D quality of the images. The 3D pdf was created in Materialise 3-matic 16.0.

### Reporting summary

Further information on research design is available in the Nature Portfolio Reporting Summary linked to this article.

### Data availability

Synchrotron data are available at https://doi.org/10.5281/zenodo.8255969.

**Acknowledgements** The authors thank L Laibl and D. Snitting for consultation on scientific content and technical issues. The research team was financially supported by the Czech Science Foundation grant GA18-05935S (P.K., M.M.); Charles University, Prague, through the programme Cooperatio GEOL (P.K., O.F.); the P JAC project CZ.02.01.01/00/22_010/0002902 MSCA Fellowships CZ - UK (V.V.); the Centre for Geosphere Dynamics UNCE/SCI/006 (V.V.); institutional budget of the University of West Bohemia in Plzeň (M.M.); Czech Geological Survey, through the DKRVO 2023-2027 (P.B.); and a Wallenberg Scholarship from the Knut and Alice Wallenberg Foundation (P.E.A.). The Article is a contribution to the IGCP project 735. The fossil was scanned at the European Synchrotron Radiation Facility (ESRF) in Grenoble on beamline ID19 as part of the proposal ES673. We thank P. Tafforeau and V. Fernandez for their assistance during the scanning session. The access of V.V. to ESRF was made possible by the Ministerstvo Školství, Mládeže a Tělovýchovy (MŠMT) funding 33914/2017-1. 3D printing was performed at U-PRINT: Uppsala University's 3D-printing facility at the Disciplinary Domain of Medicine and Pharmacy and SciLifeLab Uppsala.

**Author contributions** P.K. designed and supervised the project. V.V., P.E.A. and P.K. conducted the synchrotron microtomography scans. V.V. segmented the microtomographic data. V.V. and M.M. produced the figures and tables. P.K. and V.V. compiled the primary interpretations and wrote the original draft. P.B., O.F. and P.E.A. reviewed and edited the manuscript. P.E.A. performed the language editing. All authors contributed to the interpretation of the results, discussion, manuscript writing and revision.

**Funding** Open access funding provided by Uppsala University.

**Competing interests** The authors declare no competing interests.

**Additional information**
**Correspondence and requests for materials** should be addressed to Per E. Ahlberg.

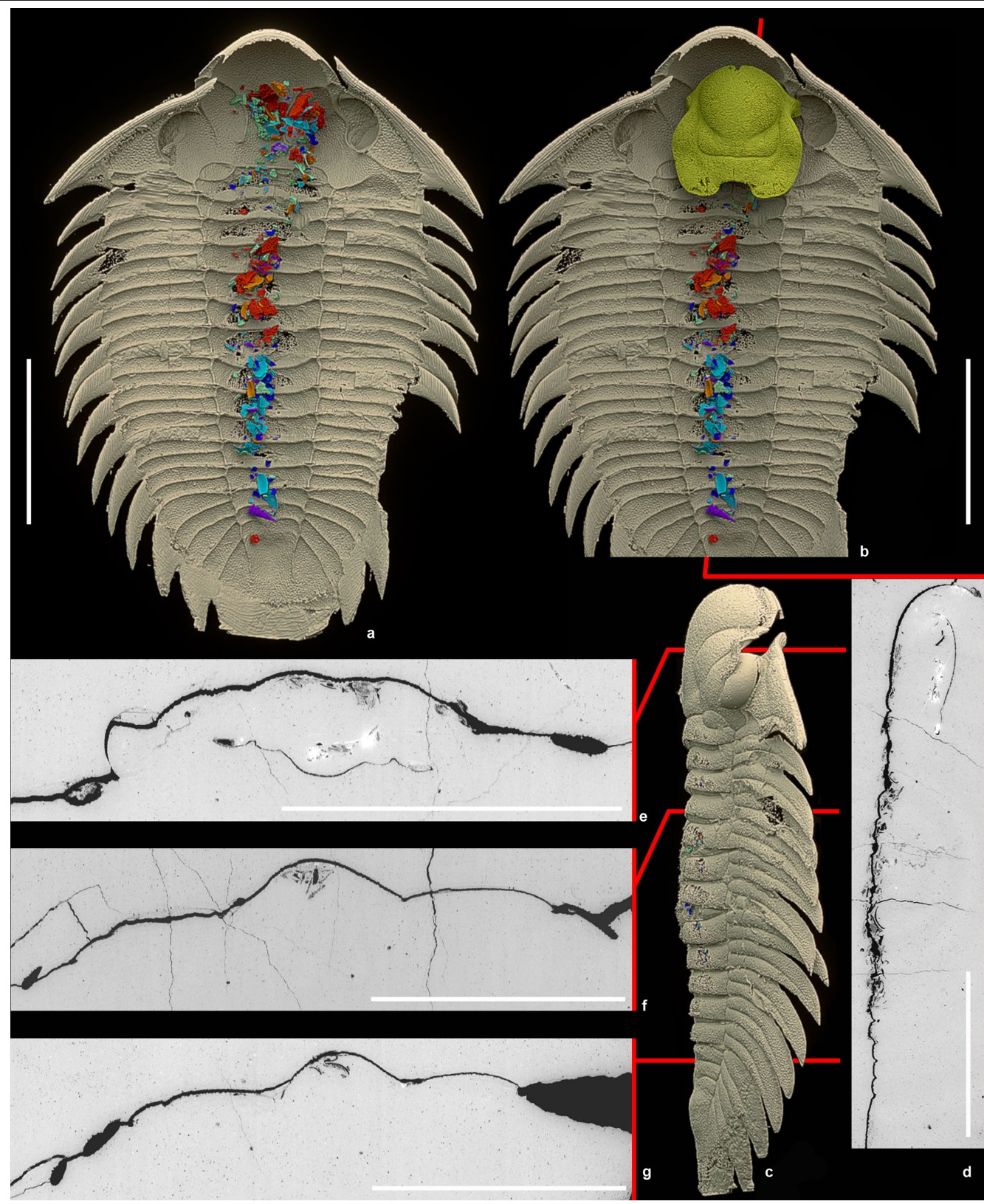

**Extended Data Fig. 1 | *Bohemolichas incola* (Barrande, 1872). a**, Scan model of exoskeleton in ventral view with digestive tract contents; **b**, and hypostome. **c**, Right lateral view. **d**, Sagittal section; **e**–**g**, transverse sections from the scan. Position of the sections indicated on the scan models by red lines. Colour coding as in Fig. 1. (applies for all Extended Data figures). Scale bars 10 mm.

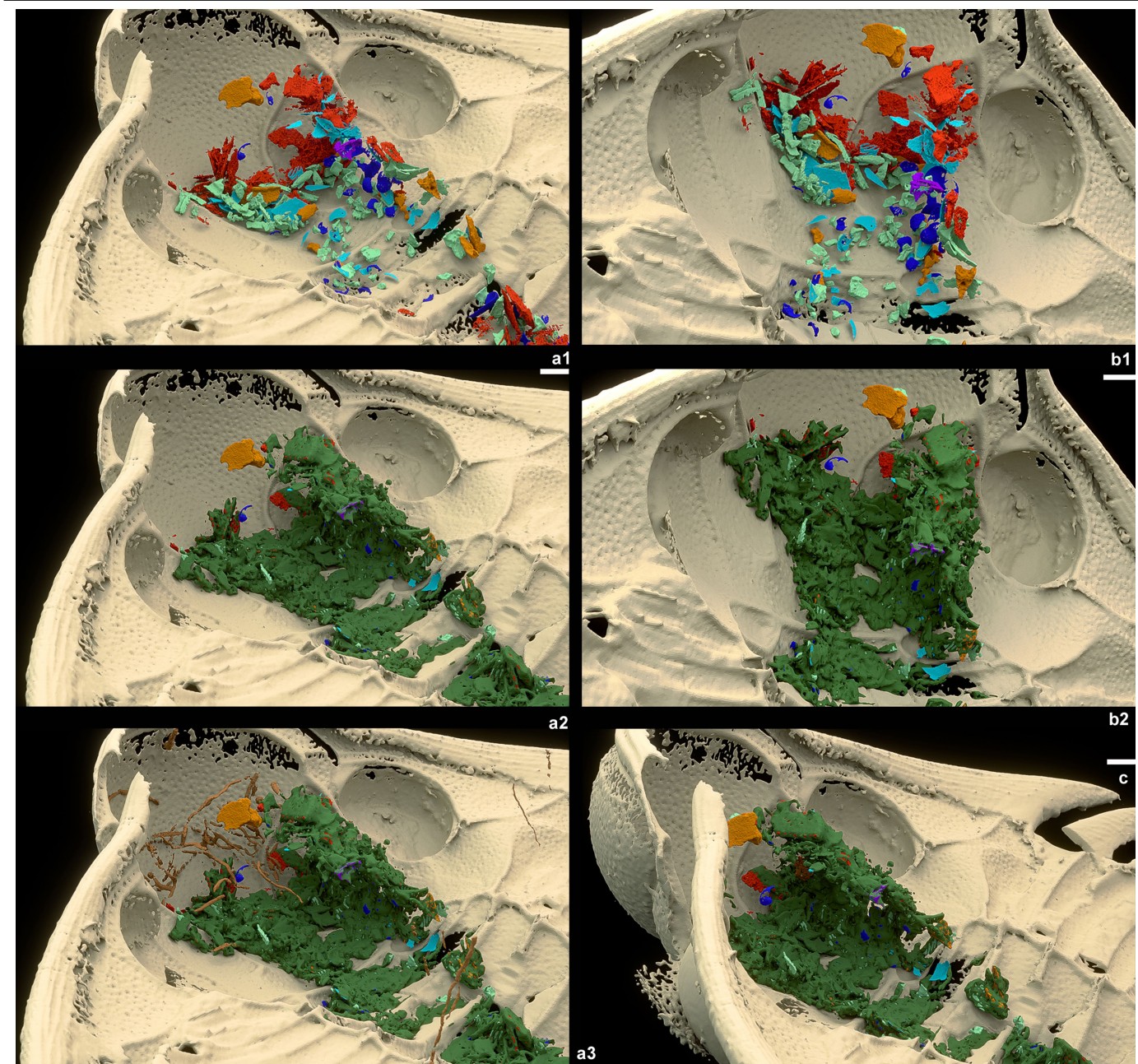

**Extended Data Fig. 2 | Scan model of the anterior part of the digestive tract infill with exoskeleton. a1**, oblique right lateroventral view; **a2**, with 'crumble'; **a3**, and associated trace fossils. **b1**, in oblique ventral view; **b2**, with 'crumble'. **c**, In right lateroventral view with 'crumble'. Scale bars 1 mm.

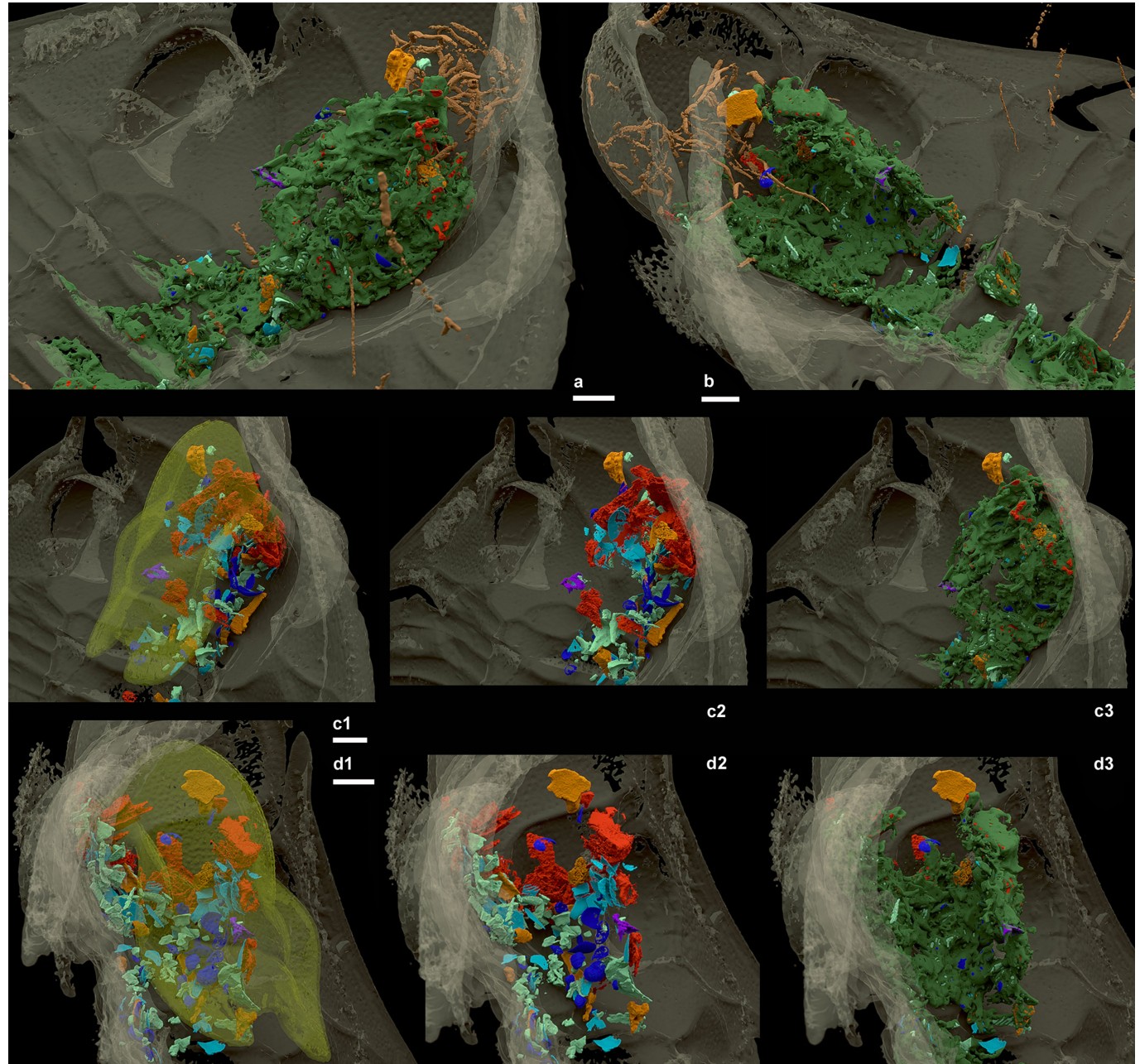

**Extended Data Fig. 3 | Scan model of the anterior part of the digestive tract infill on transparent exoskeleton. a**, 'Crumble' and trace fossils in oblique left ventrolateral view; **b**, in oblique right ventrolateral view. **c1**, In oblique left ventrolateral view with transparent hypostome; **c2**, with tract infill contents; **c3**, and 'crumble'. **d1**, In oblique right ventrolateral view with transparent hypostome; **d2**, with tract infill contents; **d3**, and 'crumble'. Scale bars 1 mm.

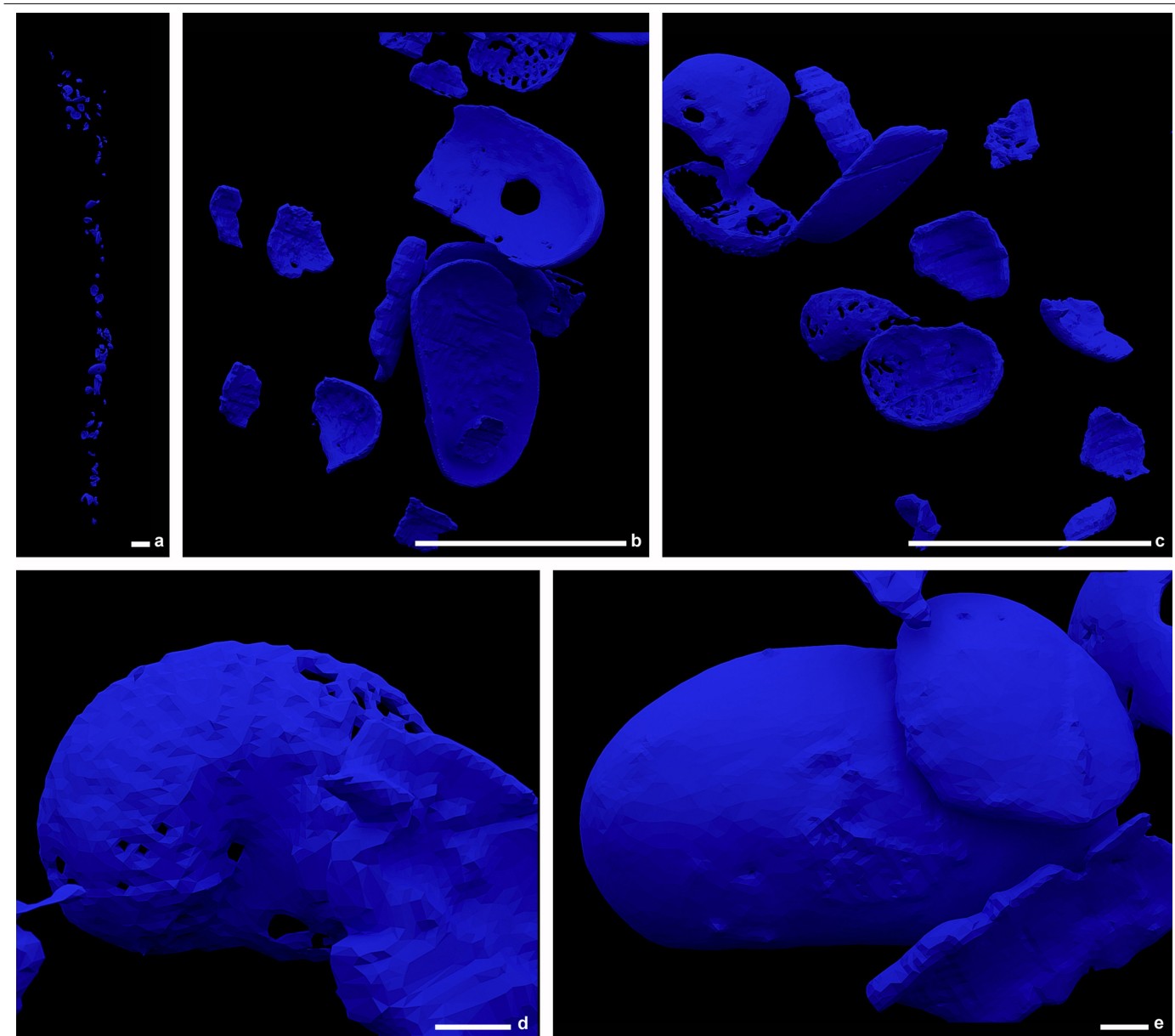

**Extended Data Fig. 4 | Fragments of ostracod shells. a**, within the whole digestive tract in left lateral view. **b**–**e**, details of ostracod shells. Scale bars in **a**–**c** 1 mm, in **d**–**e** 100 μm.

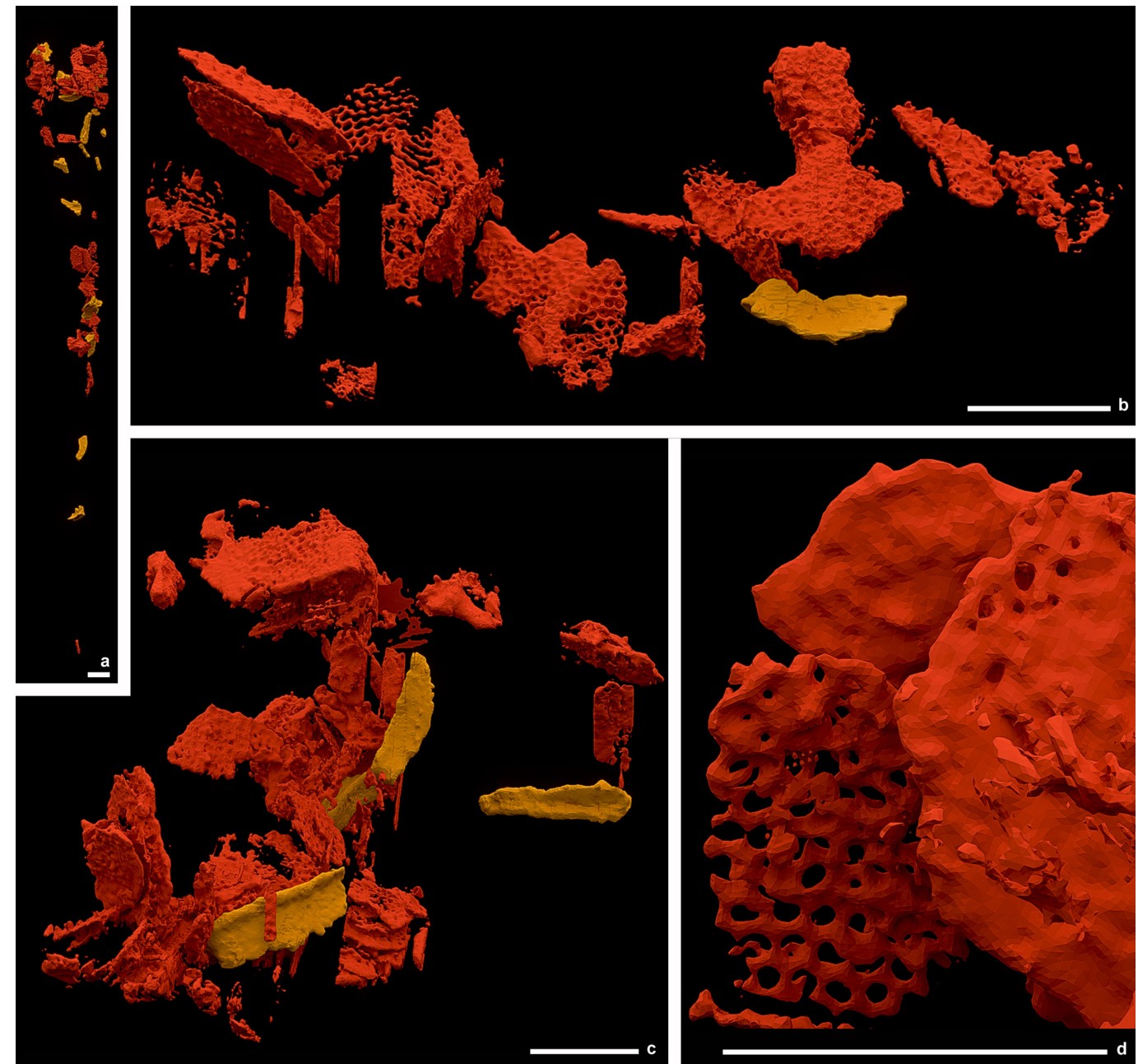

**Extended Data Fig. 5 | Fragments of stylophoran echinoderm plates.** (Red – central; orange – marginal) **a**, Within the whole digestive tract in left lateral view. **b**, Detail of central part of the tract in ventral view (anterior to the left). **c**, Detail of head area in ventral view (anterior to the left). **d**, Detail of central plates. Scale bars 1 mm.

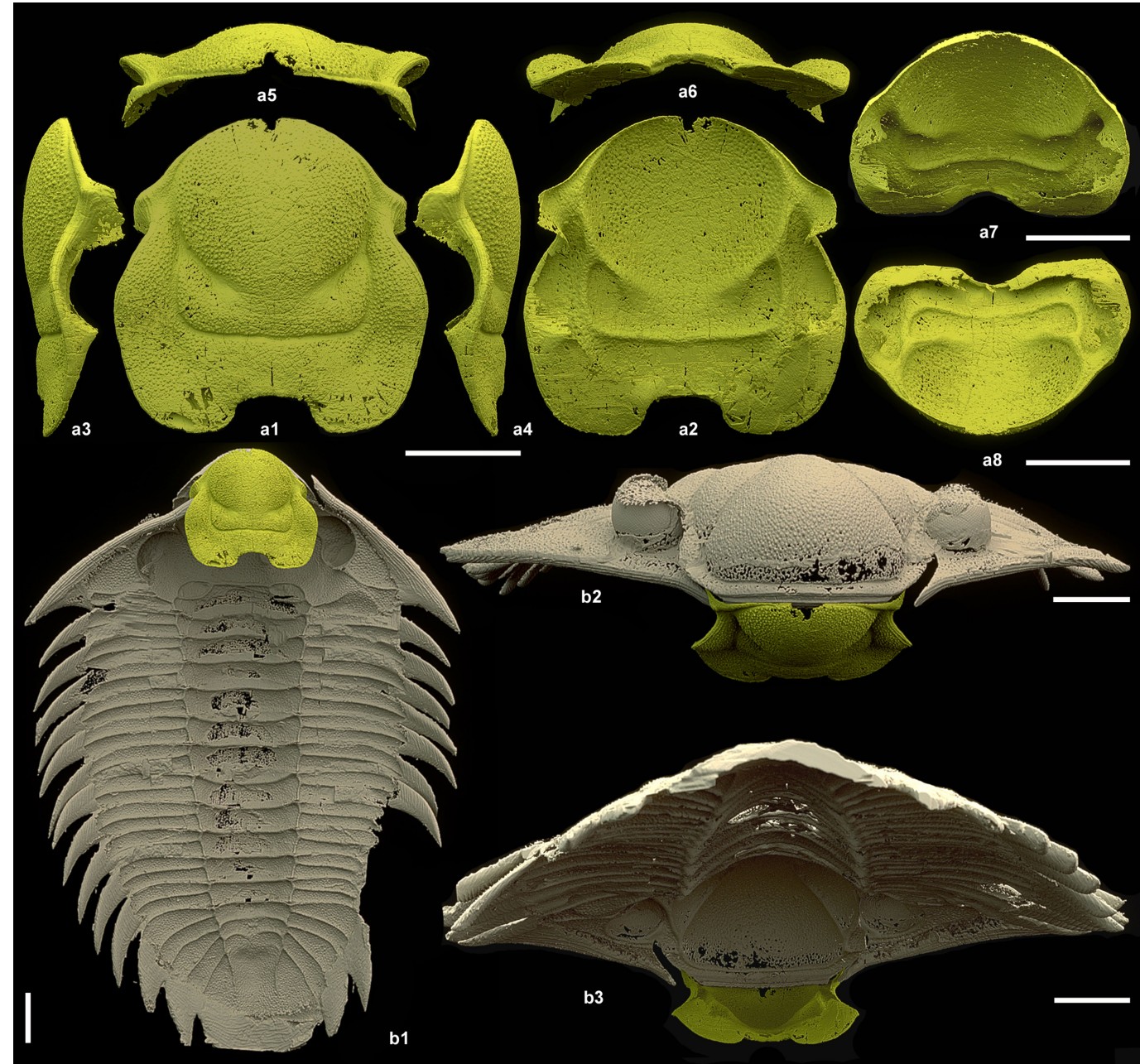

**Extended Data Fig. 6 | Scan model of hypostome. a1**, ventral; **a2**, dorsal; **a3**, left lateral; **a4**, right lateral; **a5**, anterior; **a6**, posterior; **a7**, oblique posterodorsal; **a8**, oblique anterodorsal views. **b1**, Compound reconstruction of the hypostome in life position (made by positioning the scan model of the hypostome on the exoskeleton in virtual space; verified using 3D printouts) in ventral; **b2**, anterior; **b3**, posterior views. Scale bars 3 mm.

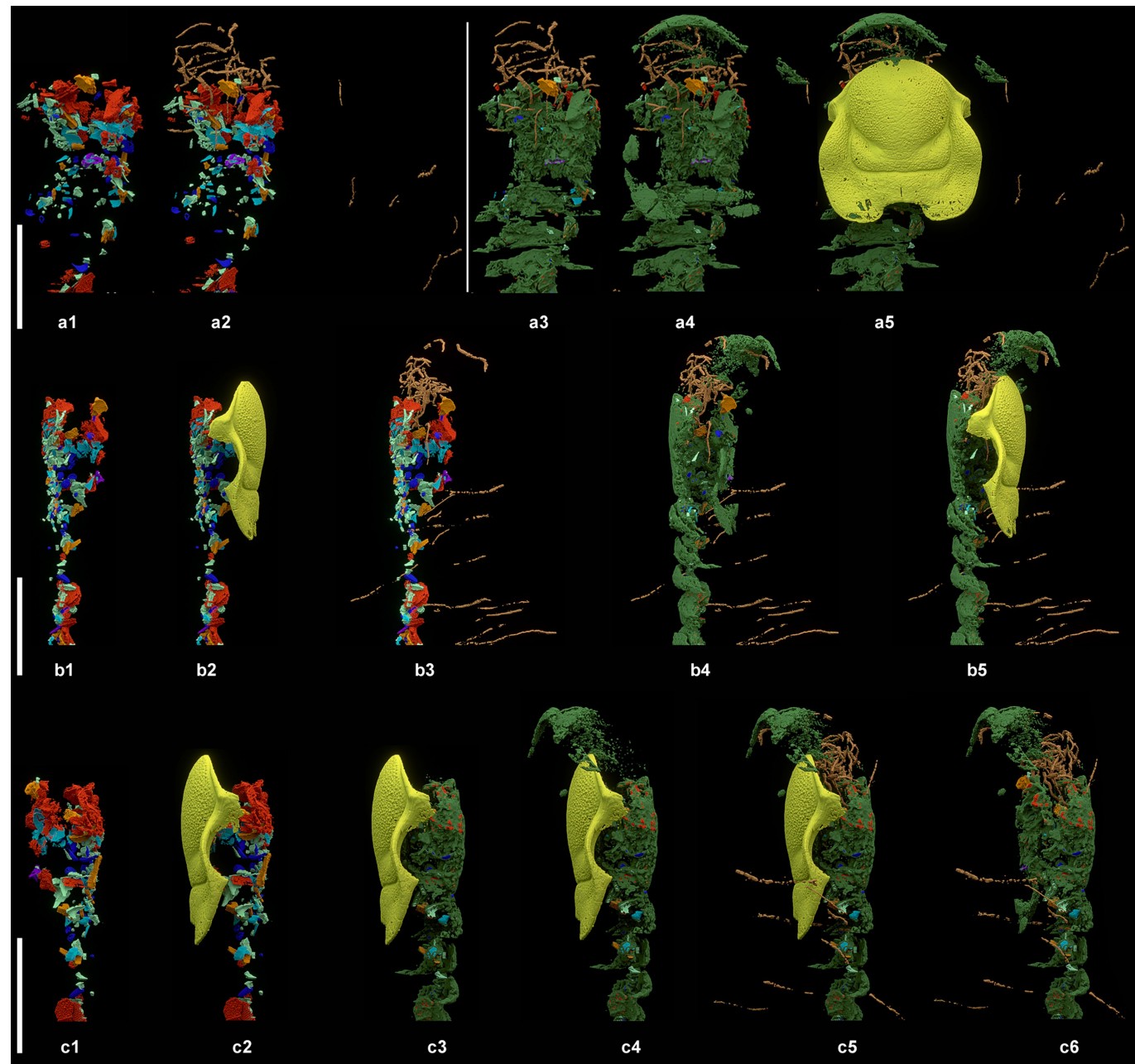

**Extended Data Fig. 7 | Scan model of the anterior part of the digestive tract infill. a1**, In ventral view; **a2**, with associated trace fossils; **a3**, and 'crumble' of gut; **a4**, and ligaments and 'crumble' within the hypostome; **a5**, and hypostome. **b1**, In right lateral view; **b2**, with hypostome; **b3**, with trace fossils; **b4**, and 'crumble'; **b5**, and hypostome. **c1**, In left lateral view; **c2**, with hypostome; **c3**, and 'crumble'; **c4**, and ligaments; **c5**, and associated trace fossils; **c6**, without hypostome. Scale bars 5 mm.

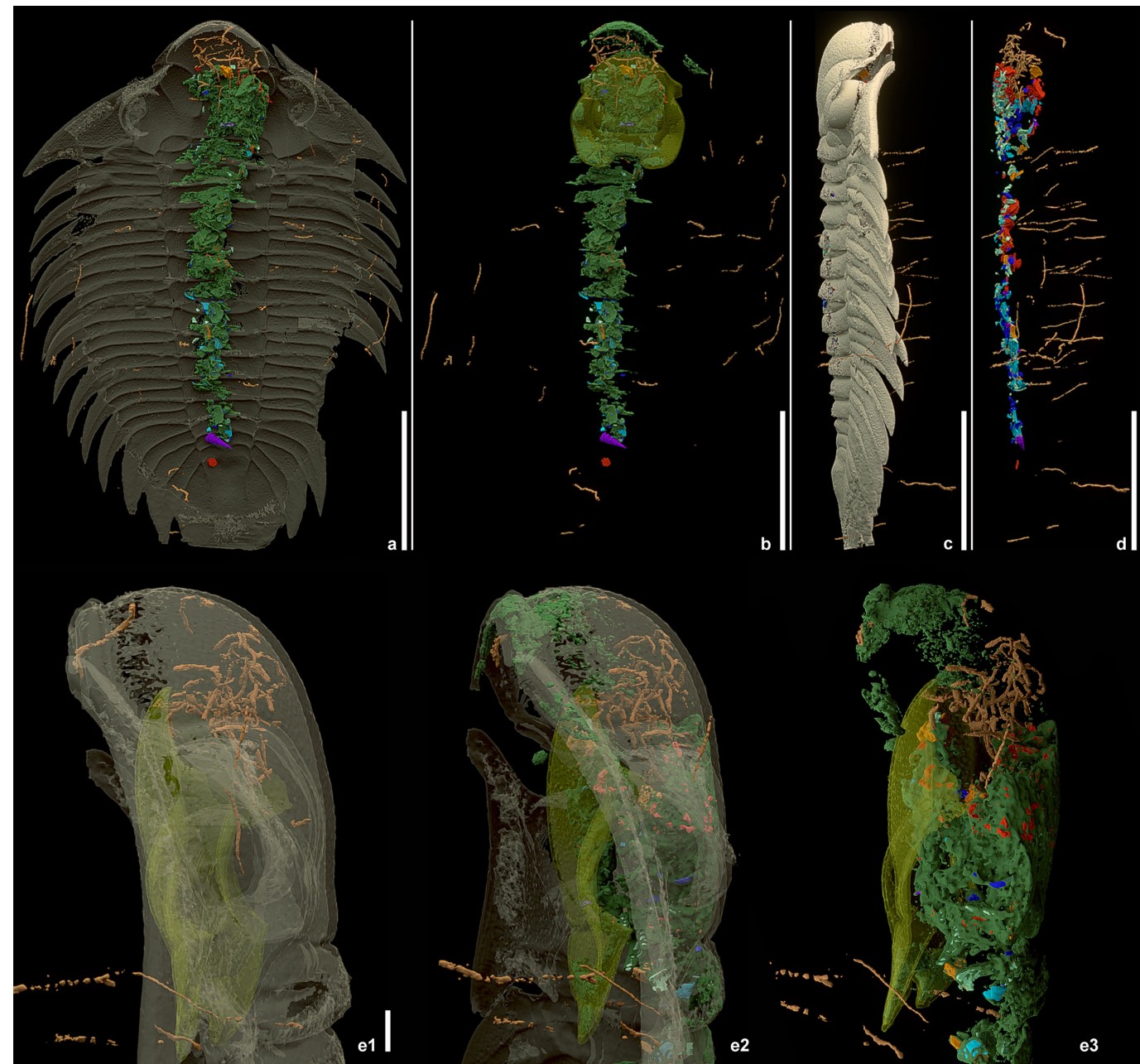

**Extended Data Fig. 8 | Associated trace fossils of scavengers. a**, In ventral view with digestive tract contents and 'crumble' on transparent exoskeleton; **b**, with ligament and transparent hypostome. **c**, In right lateral view with exoskeleton; **d**, without exoskeleton. **e1**, Head region in left lateral view, with transparent exoskeleton and hypostome; **e2**, and 'crumble' of gut and ligament; **e3**, without exoskeleton. Scale bars in **a**–**d** 10 mm, in **e1** 1 mm.

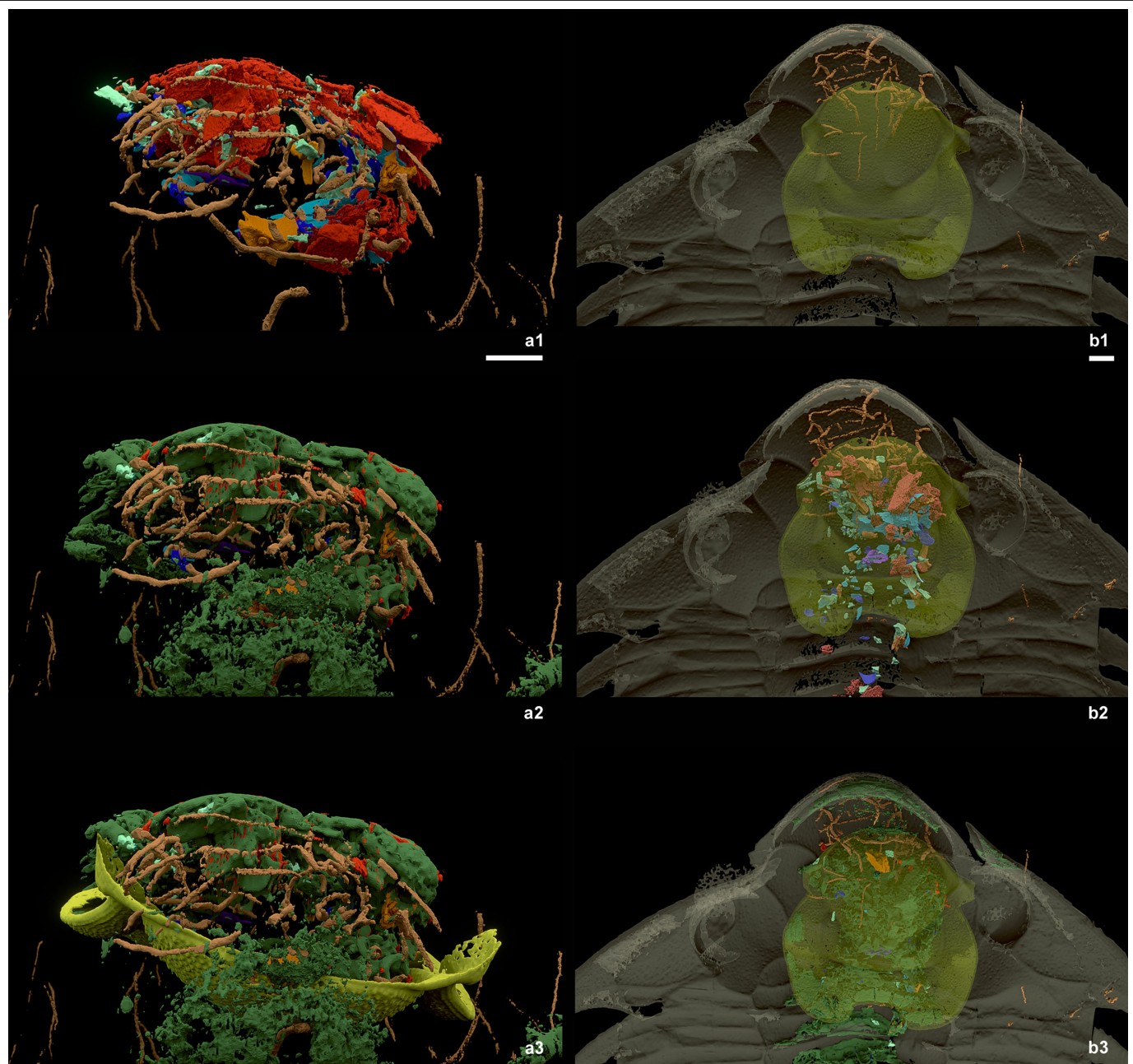

**Extended Data Fig. 9 | Associated trace fossils of scavengers in the head region. a1**, In anterior view (dorsal side up), with digestive tract contents; **a2**, and 'crumble' of gut and ligament; **a3**, and hypostome. **b1**, In ventral view, with transparent exoskeleton and hypostome; **b2**, and transparent digestive tract contents; **b3**, and 'crumble' of gut and ligament. Scale bars 1 mm.

**Extended Data Table 1 | Scan parameters of specimen inv. no. 8**

| | |
|---|---|
| voxel size (μm) | 11,35 |
| sample | *Bohemolichas incola* |
| optic | Lafip2 / Hasselblad 210 mm |
| date / proposal | 17/02/2018 es673 |
| Average energy (keV) | ~177 |
| Filters (mm) | Cu 5 |
| | W 0.7 |
| propagation distance (mm) | 11000 |
| sensor | sCMOS PCO edge 4.2 |
| Scintillator | LuAG:Ce 2000 |
| insertion device | W150 |
| ID Gap (mm) | 26,5 |
| machine filling mode | 7/8 multibunch 200 mA |
| projection number | 5000 |
| scan geometry | 360°, half-acquisition 800 pixels, vertical series 4.5 mm, double scan, three lateral columns |
| subframe time (s) | 0,01 |
| exposure time (s) | 0,05 |
| accumulation level | 5 |
| time per scan (min) | 4,90 |
| number of scans | 168 |
| total time (h) | 14 |
| reconstruction | single distance phase retrieval, ring artefact correction, vertical concatenation, 16 bits conversion, lateral concatenation, binning 2 |
| refs / dark | single ref/dark |

# Reporting Summary

## Statistics

For all statistical analyses, confirm that the following items are present in the figure legend, table legend, main text, or Methods section.

| n/a | Confirmed | |
|---|---|---|
| ☐ | ☒ | The exact sample size (*n*) for each experimental group/condition, given as a discrete number and unit of measurement |
| ☐ | ☒ | A statement on whether measurements were taken from distinct samples or whether the same sample was measured repeatedly |
| ☐ | ☒ | The statistical test(s) used AND whether they are one- or two-sided<br>*Only common tests should be described solely by name; describe more complex techniques in the Methods section.* |
| ☒ | ☐ | A description of all covariates tested |
| ☒ | ☐ | A description of any assumptions or corrections, such as tests of normality and adjustment for multiple comparisons |
| ☒ | ☐ | A full description of the statistical parameters including central tendency (e.g. means) or other basic estimates (e.g. regression coefficient) AND variation (e.g. standard deviation) or associated estimates of uncertainty (e.g. confidence intervals) |
| ☐ | ☒ | For null hypothesis testing, the test statistic (e.g. *F*, *t*, *r*) with confidence intervals, effect sizes, degrees of freedom and *P* value noted<br>*Give P values as exact values whenever suitable.* |
| ☒ | ☐ | For Bayesian analysis, information on the choice of priors and Markov chain Monte Carlo settings |
| ☒ | ☐ | For hierarchical and complex designs, identification of the appropriate level for tests and full reporting of outcomes |
| ☒ | ☐ | Estimates of effect sizes (e.g. Cohen's *d*, Pearson's *r*), indicating how they were calculated |

*Our web collection on statistics for biologists contains articles on many of the points above.*

## Software and code

Policy information about availability of computer code

| Data collection | No software was used to collect the data. |
|---|---|
| Data analysis | The scan data with voxel size of 11.35 μm were segmented in Mimics Research 19.0 (Materialise) software. Rendering was done in Blender 2.79b. |

For manuscripts utilizing custom algorithms or software that are central to the research but not yet described in published literature, software must be made available to editors and reviewers. We strongly encourage code deposition in a community repository (e.g. GitHub). See the Nature Portfolio guidelines for submitting code & software for further information.

## Data

Policy information about availability of data

All manuscripts must include a data availability statement. This statement should provide the following information, where applicable:

- Accession codes, unique identifiers, or web links for publicly available datasets
- A description of any restrictions on data availability
- For clinical datasets or third party data, please ensure that the statement adheres to our policy

Synchrotron data are available at https://doi.org/10.5281/zenodo.8255969

# Research involving human participants, their data, or biological material

Policy information about studies with human participants or human data. See also policy information about sex, gender (identity/presentation), and sexual orientation and race, ethnicity and racism.

| | |
|---|---|
| Reporting on sex and gender | N/A |
| Reporting on race, ethnicity, or other socially relevant groupings | N/A |
| Population characteristics | N/A |
| Recruitment | N/A |
| Ethics oversight | N/A |

Note that full information on the approval of the study protocol must also be provided in the manuscript.

# Field-specific reporting

Please select the one below that is the best fit for your research. If you are not sure, read the appropriate sections before making your selection.

☐ Life sciences  ☐ Behavioural & social sciences  ☒ Ecological, evolutionary & environmental sciences

For a reference copy of the document with all sections, see nature.com/documents/nr-reporting-summary-flat.pdf

# Ecological, evolutionary & environmental sciences study design

All studies must disclose on these points even when the disclosure is negative.

| | |
|---|---|
| Study description | *Briefly describe the study. For quantitative data include treatment factors and interactions, design structure (e.g. factorial, nested, hierarchical), nature and number of experimental units and replicates.* |
| Research sample | Synchrotron scan of a fossil trilobite enclosed in a siliceous nodule. |
| Sampling strategy | N/A |
| Data collection | The specimen was scanned using propagation phase-contrast synchrotron microtomography (PPC-SRμCT) at the ID19 beamline of the European Synchrotron Radiation Facility (ESRF) in Grenoble, France, as a part of the proposal ES 673. The reconstructed volumes were converted into stacks of 16 bits TIFF images. The scan data with voxel size of 11.35 μm were segmented in Mimics Research 19.0 (Materialise) software. Rendering was done in Blender. |
| Timing and spatial scale | 17.2.2018 |
| Data exclusions | No data were excluded. |
| Reproducibility | N/A |
| Randomization | N/A |
| Blinding | N/A |

Did the study involve field work?  ☐ Yes  ☒ No

# Reporting for specific materials, systems and methods

We require information from authors about some types of materials, experimental systems and methods used in many studies. Here, indicate whether each material, system or method listed is relevant to your study. If you are not sure if a list item applies to your research, read the appropriate section before selecting a response.

## Materials & experimental systems

| n/a | Involved in the study |
|-----|----------------------|
| ☒ | ☐ Antibodies |
| ☒ | ☐ Eukaryotic cell lines |
| ☐ | ☒ Palaeontology and archaeology |
| ☒ | ☐ Animals and other organisms |
| ☒ | ☐ Clinical data |
| ☒ | ☐ Dual use research of concern |
| ☒ | ☐ Plants |

## Methods

| n/a | Involved in the study |
|-----|----------------------|
| ☒ | ☐ ChIP-seq |
| ☒ | ☐ Flow cytometry |
| ☒ | ☐ MRI-based neuroimaging |

## Palaeontology and Archaeology

Specimen provenance | No permits needed, specimen was collected a century ago.

Specimen deposition | Museum of Dr. B. Horák, Rokycany, Czech Republic (inventory number 8)

Dating methods | N/A

☐ Tick this box to confirm that the raw and calibrated dates are available in the paper or in Supplementary Information.

Ethics oversight | No ethical approval was requested, specimen is housed in a public repository.

Note that full information on the approval of the study protocol must also be provided in the manuscript.

