## [Peer Review File · Nature]

Manuscript Title: Uniquely preserved gut contents illuminate trilobite palaeophysiology

Reviewer Comments & Author Rebuttals

Reviewer Reports on the Initial Version:

Referees' comments:

Referee #1 (Remarks to the Author):

Uniquely preserved trilobite gut contents reflecting a life cycle snapshot By Petr Kraft, Valéria Vaškaninová, Michal Mergl, Petr Budil, Oldřich Fatka, Per E. Ahlberg

This MS presents magnificent images of the gut contents of an Ordovician trilobite, that were obtained by using synchrotron microtomography (propagation phase-contrast synchrotron microtomography). Recently this powerful technique was successfully applied to various fossil specimens (e.g. Cambrian *Saccorhytus* published in Nature by Liu et al. 2022 or Huldtgren et al. 2011 in Science). These findings are however not new. A paper published in 2016 (Zacai et al. Arthropod Structure and Development) describes in details the gut contents (at species level) and the diet of *Sidneyia*, a ca 50-million-year older arthropod from the Cambrian Burgess Shale, closely related to trilobites (Artiopoda). This important paper (and other related ones) is simply ignored throughout the MS, which casts some doubt on the scientific seriousness of the authors. Many other key references are also missing. The MS also contains a great deal of speculation (except the descriptive part) and very poorly supported statements (see list of remarks that concern almost every sentence of the MS, below). The authors also embark into unnecessary discussions on molting, life cycle, Ph-conditions that are clearly out of the scope of the main target- i.e.- the diet of trilobites and the role of the group in ancient food chains. The whole paragraph concerning supposed scavengers is particularly obscure, full of speculation and not very credible, especially because of the lack of discussion concerning their biological nature (what kind of scavenger may have drilled into the carapace?). Numerous sentences are not understandable at all especially in the discussion part. What the engineers from Grenoble (ESRF) have done with this fossil specimen is a technical feat. Unfortunately, none of them seems to have been invited to join the present authors. How the authors used this remarkable set of data is unfortunately very disappointing. Although the images presented in this MS are spectacular and of top-quality and have good chance to attract readers, I will not recommend this MS for publication in Nature because of major scientific weaknesses throughout the MS. In contrast, the paper by Liu et al. 2022 in Nature that also used synchrotron microtomography was scientifically very well supported. Not the present MS. I would recommend the authors either to focus on mainly technical aspects (e.g. how synchrotron techniques reveal trilobite gut contents) and submit their MS to a specialized journal, or rewrite their MS on a more solid basis (adequate refs, less speculation) and publish it in a good palaeontology journal.

TITLE Uniquely preserved trilobite gut contents reflecting a life cycle snapshot I don't think it is a good title. The most significant results here chiefly concern the diet of trilobites and the role of these arthropods in Palaeozoic marine food webs. I would rather suggest something like "Direct evidence of trilobites feeding mode and diet?"

ABSTRACT

Line 19- That's true but gut contents were already described in closely related arthropodan arthropods (Sidneyia; Zacaï et al. XXXX). Please mention it.

Line 23- What do you mean by voracious? These trilobites may have fed on decaying carcasses (scavenger vs predation).

Line 23-25- Very unclear to readers. Why are you talking about the life cycle here? What you have here is a snapshot of the feeding mode of the animal i.e. what he had in its gut just before it was entombed and died. Please rewrite or simply delete.

Line 25-26 -The lack of dissolution of the shells implies a neutral or alkaline environment along the length of the intestine
It sounds a bit odd here. The trilobite exoskeleton is also mineralized as were the shell fragments in the gut. I see no evidence of a neutral or alkaline microenvironment within the gut in particular. Please rewrite and explain.

Line 27, 28- supporting the digestive enzymes comparable to those in modern crustaceans or chelicerates.

Nothing really supports enzymatic activity here. I see no digestive glands preserved in your specimen. In contrast Sidneyia (another arthropodan arthropod close to trilobite) from the Cambrian Burgess Shale (Zacaï et al. XXXX) displays digestive glands and also appendages with well-developed gnathobases, both providing details on how food was processed by the animal. No such key information is given in your MS simply because neither internal organs, nor appendages are preserved.

Line 28-30- Scavengers burrowing into the trilobite carcass targeted soft tissues below the glabella but avoided the gut, suggesting noxious conditions and apparently continuing digestive activity after death.

It is too speculative. What kind of scavengers? I see no evidence of scavenging activity on your fossil specimen. Why do you suppose that the gut was filled with noxious chemicals? Carcasses of relatively large animals such as trilobites probably served as an important food source to other animals (See Vannier 2012. In-situ Ottoia around the carcass of Sidneyia). Carcasses were also colonized by bacteria, including the digestive tract (see preserved bacteria in the gut content of a Cambrian arthropod in Zhu et al. XXXX). I see no such evidence in your MS. Autolysis occurs immediately after death and makes all organs and biochemical reactions ineffective. The digestive function stops. In contrast microbial activity takes over and consumes virtually everything. That enzymatic activity continues after death is purely speculative.

I would recommend the authors to re-focus on their main results: the diet of trilobites.

MAIN TEXT

Line 47- Small fossils and fragments of shells are densely distributed exclusively along the exoskeletal axial lobe
Please provide evidence that no other concentration of shell fragments occurs elsewhere within this trilobite.

Line 64- Please provide images showing this ostracod species in the rocks where the trilobite with gut contents was found. Line 65- the genus *Elegantilites* Same request here

Line 69- These fragments certainly belong to echinoderms (typical stereo pattern). However, please provide evidence that they do belong to stylophorans. Do you have stylophorans in the horizon where the trilobite with gut contents comes from. If so, provide images of a specimen and details of plates.

Line 77- implying that the trilobite devoured at least a half of the digestive tract content in one place and time, and thus showed a fast-occasional feeding behaviour.

This sentence is very unclear. Nothing proves that the trilobite “devoured” stylophorans. Most likely, it fed on decaying carcasses (e.g. stylophorans with already dislocated plates). Food was presumably digested in the frontal part of the gut (cephalon; that’s how it worked in *Sidneyia*) then undigested fragments conveyed through the gut tract by possible peristaltic contractions.

Line 79- Why do you think these fragments belong to bivalves? See remarks above.

Line 85- The accumulations of the devoured shells in the cephalic region form two sack-like ventriculi positioned dorsoventrally above each other between the hypostome and glabella (Fig. 3,

I see no distinct clusters of shell fragments that would support the presence of two ventriculi. In trilobite there is a unique spacious pouch within the cephalon that receives ingested food. The foregut of all arthropods is a tubular and muscular structure that served to suck up food. It has no role in concentrating food.

Line 87-93- Speculation here, not supported by any fossil evidence.

Line 97- The ring of the third thoracic segment indicating the beginning of the hindgut
No. In arthropods, the hindgut is the terminal part of the gut tract, close to the anus. It cannot be there.

Line 103- The locomotory appendages (endopods) of the trilobite must have been long and sturdy enough to raise the vaulted ventral side of the body off the substrate while moving. It seems to be quite evident. Appendages are well preserved in *Sidneyia* an arthropod close to trilobites (see again Zhai et al. 2016)

The presence such a grinding apparatus in *B. incola* was tested by comparing the relative size of fragments in both ventriculi and the gut.

In the extant *Limulus*, shells (e.g. muscles) are crushed and broken up by the strong gnathobases that occur at the base of their appendages (add recent refs). Comparable gnathobases are found in *Sidneyia* (Zhai et al. 2016). I see no evidence of an internal grinding apparatus in the trilobite described here.

Line 114-116- The ventral ventricular chamber of *B. incola* could represent a collecting space for ingested material prior to its passage into the dorsal chamber and gut for final digestion without subsequent grinding.

Quite speculative considering that there is no evidence of two distinct pouches.

Line 124- Digestive glands are well-preserved (including internal diverticles) in *Sidneyia*, an arthropod close to trilobites, where gut contents occur. Curiously this key paper is not cited in this MS.

Line 129- The almost entirely filled digestive tract combined with the ‘crumble’ reflects its simple J-like shape and allows the tract to be reconstructed (Fig. 5).

This sentence is unclear + please don’t use “crumble”; choose another term.

Line 136- taxa have no size. Size applied to individuals.

Line 139- fragments of large, thin-walled shells (probably bivalves), Why do you suppose they are large? You don't even know that the fragments belong to bivalves.

Line 139-142 The diet composition indicates a limited ability of the trilobite to break firm or robust inorganic shells, based on the absence of a concentrated assemblage of devoured fragments in the intestine, which could be identified as pieces of a complete crushed shell. Odd here. The gut of the trilobite does contain a high concentration of fragments apparently! The assumed bivalves seem to be broken into pieces.

Line 142-143- This accords with the non-biomineralized nature of trilobite gnathobases that were used for food processing/mastication
Problem here too. *Limulus* has strong non-biomineralized gnathobases with strongly sclerotized teeth that are able to break muscles (shell) into pieces. Same with *Sidneyia*. The gnathobases of *Sidneyia* are non-biomineralized elements that were able to break small trilobites into pieces (their remains are found in the gut). Again, this paper and other related one are not cited!!

Line 143-146- The larger valves reflect a clear preference for fragile elements which can be easily broken. Complete vertebral like marginal plates of a stylophoran indicate small size preference of solid elements (Fig. 2b, Extended Data Fig. 6)
These sentences are very unclear. Please rewrite.

Line 148- Durophagy is primarily an ability to crush inorganic parts of organisms with an aim to reach the soft, digestible tissues
Not necessarily inorganic. Durophagy refers to animals that have mouth parts or other appendages adapted to break hard skeletons (biomineralized or not)

Line 149- the ingestion of shell fragments is a side effect of this behaviour rather than intention
Delete this and avoid intention.
There is no side-effect here. Potential prey or carcasses were detected via chemical or visual cues. Small items were ingested whole (e.g. ostracods) others are broken into pieces and swallowed.

Line 150- However, the large accumulation of hard particles in the intestine of *B. incola* reflects a different feeding strategy.
Why different, I don't understand. Different from what?

Line 151-154- The studied trilobite was polyphagous with a low degree of food particle preference in their positive frequency dependent proportion¹, which means that the food selection was based rather on size and shell resistance than on the taxonomical composition. The way this sentence is written is odd. Moreover, it does make much sense. The size of ingested items is controlled by the diameter of the mouth opening and oesophagus diameter. Shells are broken into small pieces if the gnathobases and associated muscles are powerful enough to crush them. The animal is unable to detect which shell is less resistant than others. The process is purely mechanical.
Instead you should consider that this trilobite was an opportunistic feeder (predator or scavenger or more likely both) as was *Sidneyia* (Zacai et al. 2016) or *Ottoia* (Vannier 2012). Both *Sidneyia* and *Ottoia* have well-preserved gut contents that make it possible to characterize their diet.

Line 154- Thus, it contradicts a typical durophagous strategy, which is usually specialized on a limited variety of shelly animals.

Absolutely not true. The definition of durophagy does not imply that ingested food is not diverse!
Please delete.

Line 156- Because *B. incola* was a non-selective feeder it can be inferred that it was a scavenger. If predatory behaviour was present it was only occasional because predatory arthropods are usually also selective in their diet preferences.

Yes, it is correct that the trilobite in question may have been a scavenger. However how would you explain that broken shells of assumed bivalves occur in its stomach.

Line 158- It can be considered as a light crusher – a durophagous chance feeder hoovering dead or living organisms, adequately sized or thin shelled.

Unclear sentence. Please rewrite

Line 159- adequately sized or thin shelled.

Trilobites had no capacity to detect whether an invertebrate was thick or thin shelled

Line 163- Even though lacking the strongly calcified appendages typical for durophagous decapod crustaceans, which enabled this malacostracan clade to spread the feeding strategy widely since the Upper Cretaceous, trilobites occupied this niche some 250 166 million years earlier.

The example of *Sidneyia* (Zacai et al. 2016) shows that durophagy existed much earlier. Please mention this important point.

Line 169- Digestion of food composed of organic tissues along with the associated inorganic shells is highly specialized and requires deep adaptations.

No, exoskeletal elements are not digested. Only soft tissues can be digested. Please rewrite or delete. I don't see why deep adaptations are required.

Line 170 - On the other hand, it offers a rich food supply with limited competition.

This sentence does not make much sense.

Line 179 -the two-chambered ventriculus associated with a strong musculature while covering and protecting the foregut

Ventriculi not evidenced (see remarks above). Any evidence of muscles in your specimen.

Line 182- adaptation allowing the passage of large undigested organic particles through the anal opening, including necessary space for large extensor muscles.

Speculation here- Please go back to your specimen and try to determine the diameter of the most posterior part of the gut, whether it was capable to expel small or larger exoskeletal fragments.

Line 183. An increased enzymatic support, e.g. a hepatopancreas and other digestive glands¹⁷, is required for maximum energy exploitation from a diet rich in indigestible bits scarcely covered by organic films or decaying tissues.

The importance of digestive glands in the development of carnivory is discussed in Vannier et al. 2014 (Nat Comm). Please add this ref and discuss in more details. Again, enzymatic reaction has nothing to do with indigestible exoskeletons. Enzyme breakdown molecules present in soft tissues.

What do you mean by scarcely covered by organic films or decaying tissues? Please rewrite. Nobody will understand.

Line 186

This sentence sounds odd. Please rewrite There are many better refs than 17!!

Line 189- In addition, robust external organs for manipulation and mechanical grinding of inorganic shelly material must have been present.

If you had read Zacai et al. 2016 and other related papers (Bricknell, etc..) you would know that they were present.

The occlusion of the hypostome with the unpreserved gnathobases is indicated by its robust structure and a wide, double-walled posterior margin.

Very unclear sentence

Line 194- The hypostome could have served as a splitter or even a greater utilizing its prominent rough sculpture

Highly speculative. The hypostome is above all a protective element in the head region. Again, food processing (crushing) is due to gnathobases that have opposed crushing margins within the midline of the animal (see extant *Limulus*).

Line 197- functional analogy to the gastric mill or gizzard of extant arthropods) although a necessary limb specialization has not been fully demonstrated²

Vague statement (gastric mill). What do you mean here?

There is absolutely no need of specialized appendages. Look at *Limulus* or *Sidneyia*. Food is processed by the gnathobases of a series of homonomous appendages.

Line 199 +-The clusters of shell remains below the axis are distributed in 199 indistinct faecal pellets.

Unconvincing. I see no pellet in any image show in this MS. What you have here are the gut contents only.

Line 203- also for their chaotic huddles.
??????

Line 204 A disordered orientation of the shells or fragments of anisotropic shapes indicates peristaltic movements but

Pure speculation. No evidence at all of peristaltic contractions !!! such as for example beads in certain fossil and extant arthropods (crustaceans). Delete.

Line 205- it also clearly points to the presence and essential role of a protective cover of the large pellets inside the gut, such as a peritrophic membrane.

Since there is no convincing support for pellets, the hypothesis of peritrophic membrane does not make any sense

Line 207- You forget that the whole trilobite biomineralized exoskeleton is preserved. There is absolutely no reason to suppose that special PH-conditions prevailed within the gut. Please delete.

Line 209- an acid gut environment the organism would face an ion imbalance and a dangerously high concentration of extracellular Ca⁺ (hypercalcemia).

Very obscure- Delete or rearrange the discussion with examples taken from detailed studies on extant arthropod gut contents. Why “dangerous”. Avoid these terms.

Line 211. That’s absolutely not true! Please read Zacaï et al. and other articles cited in this paper. For example, the gut of modern limulids stores microspherules of Ca and P. (see figures in Zacai et al.). It is the reason why many Cambrian arthropods have their digestive system preserved in apatite. The source for Ca and P comes from inside the animal.

Line 212-215

This sentence is particularly unclear!

Line 215 +The studied specimen illustrates the potential of fossils to reconstruct life processes of extinct invertebrates and, on the other hand, points to an insufficient knowledge about many aspects of modern marine arthropods.

This sentence means nothing. Delete.

Line 217- The digestive tract of recent arthropods has been studied in detail with respect to its morphology, anatomy, food content, enzymatic processes and other features (e.g. for decapod crustaceans)

What kind of relevant information do you want to provide here? It would be far more interesting to discuss the several cases of preserved gut contents and digestive systems in early Palaeozoic arthropods (see Vannier, Lerosee-Aubril and collaborators).

Line 219- However, there are surprisingly few data on the pH values and dynamics inside their digestive tract, even though they directly influence all the mentioned characteristics. The discussion therefore has to be limited to a few model organisms.

Do you really expect that the chemical composition of mineralized guts of Ordovician trilobites will tell us about the Ph-conditions that prevailed in their digestive systems? The last sentence is obscure.

Line 225 + As decapod crustaceans and chelicerates respectively, they bracket both major lineages within the euarthropod crown group²³. Trilobites are generally agreed to fall within this bracket²⁴ Very unclear again here. Where do you place trilobites? According to the most recent phylogenies (e.g. Aria 2022), trilobites belong to Artiopoda. Artiopoda are placed somewhere between chelicerates and mandibulates (see tree in Aria's paper)

Line 230. In digestion processes, it is the enzymes that matter!

Line 232. That's pure speculation based on no detailed evidence. Delete

Line 233. Different stages of digestive tract infill by food and other particles were recorded in mud crabs, ranging from an empty to a gorged intestine²⁷, which can partly be a seasonal factor.

What do you mean here by seasonal? Yes, food composition may vary according to the season (as in birds). But quantity of ingested food is another matter. When food is abundant, crabs eat a lot and store food. When food is scarce, they stomach is often empty. Rewrite.

Line 235- The authors don't seem to know about the complexity of the moulting process.

Line 204- I don't believe this. Vaulting may have nothing to do with moulting. Preservation of arthropods in the act of moulting is extremely rare (see Garcia Bellido for Marella). Vaulting is more likely to contractions just before death. Please don't take this hypothesis for granted and be more critical here.

Line 241- The combination of these features with the gorged digestive tract suggests that the studied specimen of *B. incola* represents a premoulting phase initiated by rupture inside the thorax through an expanded intestine.

It is very weakly supported and speculative. Please refocus your paper on fossil evidence and delete all that speculative stuff that ruins the strength of the MS.

Line 242- When an arthropod is about to molt (ecdysis) the old and new cuticle start to split off. I see nothing suggesting this stage in the trilobite presented here. Your molting hypothesis is not convincing.

Line 245- indicated by a limit between two faecal pellets, reflecting different food sources and feeding periods but also different feeding intensity.

It does not make much sense since no evidence of pellets exist. Again, such poorly supported statements weaken your MS.

Line 246- We infer that the feeding behaviour of the trilobite resembled the corresponding life cycles of modern crustaceans: most of time the intestine was empty or moderately filled, with occasional and swift overfeeding actions linked to specialized physiological requirements.

Extremely speculative and based on no extant analogues. The only thing we know is that trilobites grew in size by molting in a manner similar with that of modern crustaceans. The amount of ingested food within gut was certainly highly variable and depended on the amount of a food available on the sea floor. Nothing suggests that food intake is related to molting.

Line 249- The animal was vulnerable during the premoulting and moulting phases, which is connected to a high mortality rate also in modern ecosystems and attempts to reduce it in aquacultures.

Please delete. Everybody knows that arthropods that have just discarded their old carapace are relatively soft and for this reason vulnerable. What has aquaculture to do with the topic of this MS?

Line 251- The gorged specimen represents a snapshot directly documenting this specific life period.

No. Not supported. Pure speculation.

Line 254. Life after life

You believe in reincarnation? Avoid these terms.

Line 259- This tissue was stretched and torn due to the hypostome dislocation as clearly marked by a thin, discontinuous layer of 'crumble

????? What makes you believe that this kind of supposed tissues occurred here

Line 261- Scavenging activity is generally related to the 'crumble' as a demonstration of decaying tissues¹², unlike in the studied specimen.

This sentence makes no sense at all. What do you mean by "crumble"???

Line 262- The distributional pattern of a number of thin vertical burrows indicates not only the preferred targets but also areas that were strictly avoided

Do you really think that these features were made by scavengers? If so, what kind of scavengers? Borers? These features could be simply cracks with no biological origin. This paragraph is hard to follow especially because the nature of these assumed microscavengers is not discussed. If this trilobite was a carcass exposed to scavengers why did they spend energy drilling through a thick biomineralized cuticle (glabella) whereas obtaining food was far easier among the decaying tissues that surrounded the animal.

Line 275- An analogous organ is highly probable in lichid trilobites

Also naraoiids have such well-developed diverticles (check refs)

it also represents a nutrient target that would not have to decay in order to be fed upon by a weak mouth apparatus, as opposed to, for example, tough muscle tissue, tendons or epidermis.

This sentence is not understandable

Line 280- What do you mean by dangerous? Think that the cephalon does not house only a digestive pouch but a variety of other tissues such as muscle, digestive system, haemolymph network. The problem here is the nature of your supposed scavengers. What are they? Is there anything similar in extant animals, other trilobites (check)?

Line 282 The distribution of traces suggests that the digestion processes throughout the entire tract continued until the nodule was formed in the shallow sediment and only ceased during the diagenesis. No. Totally unconvincing. Supported by no evidence. Delete. It is pure speculation

Line 284-285- The trace makers had to avoid the digestive tract to evade digestion by the dead trilobite. The behaviour of scavengers directly supports the model suggesting a rapid formation of these nodules³⁵.

Idem here- It is a nice story but nobody will believe it I am afraid. Please stick to facts and observations!

Line 287- The described specimen of Bohemolichas provides by far the most detailed source of information to date concerning the diet and the feeding mode of trilobites.

I agree that this MS tells us about the diet of the trilobite Bohemolichas but much less concerning its feeding mode (no appendage, no soft parts). As I wrote before, the diet of Sidneyia an arthropod closely related to trilobites was described in great details in 2016 (Zacai, Vannier et al). Comparisons with Sidneyia would have been extremely interesting. Unfortunately, and curiously this paper was ignored by the authors...

Line 289 It captures two interconnected but discrete moments, life and after life, frozen in one nodule. This sentence is obscure. "After life" sounds very odd. Delete

Line 290- One points to the complex digestive physiology related to a specific short moulting stage in the life cycle of the trilobite. The other documents a post mortem endurance of metabolic processes reflected through a snapshot of constrained targeting of scavengers feeding on its corpse. Not understandable. Rewrite or delete.

Referee #2 (Remarks to the Author):

Kraft et al present an exceptionally well-preserved trilobite that was synchrotron scanned. Overall, the paper is written very well, it hits a lot of key points that are assumed but lacking in the fossil record and overall, I applaud the authors for their work, this is spectacular. My only main comment is: I would very much like to see this as a 3D PDF, or ideally STLs. This kind of data are ideal for subsequent work on 3D analyses and being able to replicate what is done, or ideally use the data produced here, would allow for this work to have even more longer-term impact. I appreciate the authors may not want to share this data, but I ask that they consider doing so.

Originality and significance: This is exceptionally novel and deserves publication

Data & methodology: All sound, but as I mentioned above, the 3D data should be made available

Appropriate use of statistics and treatment of uncertainties: These were limited, but seem appropriate.

Conclusions: Very strong, some minor tweaks needed

Suggested improvements: experiments, data for possible revision: see below.

References: appropriate credit to previous work: more than fine.

Clarity and context: very well written.

I have a few minor comments below that I think could be addressed quickly. A few (such as the last one) are more philosophical based on my own interest in the studied groups, so if the authors would like to keep such terms as 'living fossil' in the text, it is their choice.

Yours,

Dr Russell Bicknell.

19: "not a single trilobite gut content has been described". I know what is implied here, but it might be better to say that "no single trilobite specimen with internal gut contents has been described" as gut contents have been noted (see Gut content fossilization and evidence for detritus feeding habits in an enrolled trilobite from the Cambrian of China, Zhu et al 2014, Lethia)

135-137 Ostracods are all not benthic, so are these possibly remains?

142-143: Bicknell et al looked at different trilobites in this paper, one of which was proposed to be a shell-crusher based on re-enforced gnathobasic spines. This may need to be reworded to reflect that, but it is very much the case for Olenoides.

156-158: "If predatory behaviour was present it was only occasional because predatory arthropods are usually also selective in their diet preferences¹⁴." I like this. However, The issue with this paper is the examples are mostly terrestrial hexapods. So I think another example here is needed, or possibly some more unpacking of the situation as this comparison is not the most appropriate when looking at trilobites.

161-163: this comparison with decapods is maybe not the best. Better to look at xiphosurids that feed very similarly.

191-195: the lack of gnathobases here does not immediately imply that the hypostome would be used in this situation. I think the lack gnathobases here reflects the lack of appendages in the scan. This should be tweaked a bit.

225: “and perceived status as a 'living fossil’”. Personally, I think this is incorrect. Living fossil is an oxymoron in this context as there are not fossil *Limulus polyphemus*.

Author Rebuttals to Initial Comments:

Response to referee #1

Remark: This MS presents magnificent images of the gut contents of an Ordovician trilobite, that were obtained by using synchrotron microtomography (propagation phase-contrast synchrotron microtomography). Recently this powerful technique was successfully applied to various fossil specimens (e.g. Cambrian *Saccorhytus* published in Nature by Liu et al. 2022 or Hultgren et al. 2011 in Science).

Response: This technique has been successfully applied also to many other fossils, including vertebrates. The novelty of the paper lies not in the technique, except to the extent that this is the first time it has been used to image trilobite gut contents, but in the results.

Remark: These findings are however not new. A paper published in 2016 (Zacai et al. Arthropod Structure and Development) describes in details the gut contents (at species level) and the diet of *Sidneyia*, a ca 50-million-year older arthropod from the Cambrian Burgess Shale, closely related to trilobites (Artiopoda).

Response: We disagree. *Sidneyia* and *Bohemolichas* are both members of the Artiopoda, but *Bohemolichas* is a trilobitomorph (and trilobite) whereas *Sidneyia* belongs most probably to the Vicissicaudata. They are thus only distantly related and the fact that gut contents have already been described in *Sidneyia* does not impact the novelty value of our data from *Bohemolichas*. Furthermore, *Bohemolichas* is the oldest fossil arthropod with the digestive system and its contents preserved and visualized in full 3D (comp. Zacai et al 2016, page 203 “most anatomical and exoskeletal features ... are strongly compressed”). Referee 2 considers these same data “exceptionally novel”.

Remark: This important paper (and other related ones) is simply ignored throughout the MS, which casts some doubt on the scientific seriousness of the authors.

Response: They were cited in the supplementary discussion, but have now move the references into the main text.

Remark: Many other key references are also missing.

Response: This remark is difficult to address, as the missing references are not specified. Referee 2 is content with our reference list. We would also stress that this is a primary descriptive paper, not a review paper.

Remark: The MS also contains a great deal of speculation (except the descriptive part) and very poorly supported statements (see list of remarks that concern almost every sentence of the MS, below).

Response: We don't entirely agree with this assessment, and feel that the referee sometimes conflates speculation with (legitimate) indirect inference. However, we have deleted or modified all statements which could be considered speculative.

Remark: The authors also embark into unnecessary discussions on molting, life cycle, Ph-conditions that are clearly out of the scope of the main target- i.e.- the diet of trilobites and the role of the group in ancient food chains.

Response: We disagree with this assessment; the topics in question are well within the scope of the paper and highly relevant to our understanding of trilobite biology. Referee 2 is enthusiastic about our approach.

Remark: The whole paragraph concerning supposed scavengers is particularly obscure, full of speculation and not very credible, especially because of the lack of discussion concerning their biological nature (what kind of scavenger may have drilled into the carapace?).

Response: We state clearly in the manuscript and supplementary information that the trilobite is lying on its back, and this should also be obvious to an observant reader from the way the intestinal contents and the hypostome have settled into the carapace. The burrows do not penetrate the carapace but reach down from the then-sediment surface towards the ventral surface of the trilobite.

Remark: Numerous sentences are not understandable at all especially in the discussion part.

Response: This is surprising, because referee 2 encountered no such problems. We refer this query back to the Editor, who will form his own opinion about the comprehensibility of the text and advise us accordingly.

Remark: What the engineers from Grenoble (ESRF) have done with this fossil specimen is a technical feat.

Response: Presenting tomographic data is not just about scanning the specimens. One of the co-authors (VV) spent several months of full-time work segmenting the data: without this the data would have been uninterpretable and there would have been nothing to present.

[This has been redacted]

Remark: How the authors used this remarkable set of data is unfortunately very disappointing. Although the images presented in this MS are spectacular and of top-quality and have good chance to attract readers, I will not recommend this MS for publication in Nature because of major scientific weaknesses throughout the MS. In contrast, the paper by Liu et al. 2022 in Nature that also used synchrotron microtomography was scientifically very well supported. Not the present MS.

Response: The referee has not in fact succeeded in identifying any 'major weaknesses', as should be clear from our responses to the other remarks.

Remark: I would recommend the authors either to focus on mainly technical aspects (e.g. how synchrotron techniques reveal trilobite gut contents) and submit their MS to a specialized journal, or rewrite their MS on a more solid basis (adequate refs, less speculation) and publish it in a good palaeontology journal.

Response: We disagree; in any case, this is a question for the Editor.

Remark: TITLE Uniquely preserved trilobite gut contents reflecting a life cycle snapshot I don't think it is a good title. The most significant results here chiefly concern the diet of trilobites and the role of these arthropods in Palaeozoic marine food webs. I would rather suggest something like "Direct evidence of trilobites feeding mode and diet?"

Response: The title was modified.

ABSTRACT

Remark: Line 19- That's true but gut contents were already described in closely related arthropods (Sidneyia; Zacaï et al. XXXX). Please mention it.

Response: See previous responses on this topic.

Remark: Line 23- What do you mean by voracious? These trilobites may have fed on decaying carcasses (scavenger vs predation).

Response: We do not understand this remark. The referee may be confusing two aspects – feeding strategy and feeding mode. Even decaying carcasses can be eaten hastily.

Remark: Line 23-25- Very unclear to readers. Why are you talking about the life cycle here? What you have here is a snapshot of the feeding mode of the animal i.e. what he had in its gut just before it was entombed and died. Please rewrite or simply delete.

Response: The sentence was deleted from the abstract.

Remark: Line 25-26 -The lack of dissolution of the shells implies a neutral or alkaline environment along the length of the intestine

It sounds a bit odd here. The trilobite exoskeleton is also mineralized as were the shell fragments in the gut. I see no evidence of a neutral or alkaline microenvironment within the gut in particular. Please rewrite and explain.

Response: The argumentation is straightforward. The exoskeleton of the trilobite is in contact with the surrounding sea water, which we would of course expect to be slightly alkaline. Gut fluids are a different matter: they can range from neutral or alkaline (for example in modern decapod crustaceans, or *Limulus*) all the way to strongly acidic (for example in our own stomach). Acidic gut fluid would certainly cause etching or dissolution of calcium carbonate shell fragments, so the fact that the shell fragments are equally well preserved throughout the gut demonstrates that the gut fluids were not strongly acidic. Post mortem, all shelly elements including the exoskeleton of the trilobite itself have been dissolved by circulating pore waters, leaving voids that replicate their shapes.

Remark: Line 27, 28- supporting the digestive enzymes comparable to those in modern crustaceans or chelicerates.

Nothing really supports enzymatic activity here. I see no digestive glands preserved in your specimen. In contrast Sidneyia (another arthropod close to trilobite) from the Cambrian Burgess Shale (Zacaï et al. XXXX) displays digestive glands and also appendages with well-developed gnathobases, both providing details on how food was processed by the animal. No such key information is given in your MS simply because neither internal organs, nor appendages are preserved.

Response: Palaeontology often relies on indirect evidence; this is a typical example. We can safely assume that trilobites had digestive enzymes, because all living arthropods have them and they are functionally necessary. Even though no digestive glands are preserved in the cephalic area, we can suppose that trilobites had them – for comparison, see e.g. Chatterton et al. (1994); Lerosey-Aubril et al. (2011, 2012), Fatka et al. (2013), Zhu et al. (2014). We can even (indirectly) locate their positions, thanks to a peculiar morphology of the glabella, dissimilar from other trilobite groups, and estimate their topology and size by comparison to modern relatives. The “digestive glands” the referee mentions in *Sidneyia* (Zacaï et al. 2016, Fig. 4) have the same shape and occupy the same position as the “diverticulae” of *Bohemolichas*. In *Sidneyia* they display some internal features in the form of striations (Zacaï et al. 2016, Fig. 5), but the authors are cautious about their determination (page 203 “these

assumed digestive glands”) and function (abstract, page 200 “their primary function was most likely to digest and assimilate food”). The gut pH of *Bohemolichas* can be estimated from the condition of the calcareous shell fragments (see previous response). Enzymes are highly sensitive to pH; in fact, in extant animals the gut pH is measured indirectly through optimal enzyme activity, because of methodological difficulties connected to pH value instability throughout the digestive tract and its variability in time depending on the feeding stages. Thus, a natural approach is to study enzymes and the conditions of their efficiency in vitro. Many papers about this methodological approach exist (see references in manuscript). We can thus indirectly but easily estimate the enzymatic activity due to the inferred environment inside the digestive tract during digestion, as we did in the case of our fossil specimen.

Remark: Line 28-30- Scavengers burrowing into the trilobite carcass targeted soft tissues below the glabella but avoided the gut, suggesting noxious conditions and apparently continuing digestive activity after death.

It is too speculative. What kind of scavengers? I see no evidence of scavenging activity on your fossil specimen. Why do you suppose that the gut was filled with noxious chemicals? Carcasses of relatively large animals such as trilobites probably served as an important food source to other animals (See Vannier 2012. In-situ *Ottoia* around the carcass of *Sidneyia*). Carcasses were also colonized by bacteria, including the digestive tract (see preserved bacteria in the gut content of a Cambrian arthropod in Zhu et al. XXXX). I see no such evidence in your MS. Autolysis occurs immediately after death and make all organs and biochemical reactions ineffective. The digestive function stops. In contrast microbial activity takes over and consumes virtually everything. That enzymatic activity continues after death is purely speculative. I would recommend the authors to re-focus on their main results: the diet of trilobites.

Response: This interpretation is based on unambiguous ichnofossils, in the form of small-diameter burrows reaching down towards the trilobite. For the other questions see Kraft et al. 2020 and replies below.

The comment on autolysis makes sense. However, contrary to the referee's claim, autolysis is a self-destruction of the necrotic tissue cells by the residual enzymatic activity of the body itself. It is obvious from the distribution of the scavenger burrows that they avoid the digestive tract, as discussed in the manuscript. Since the enzymatic activity that runs the process of autolysis starts in the tissues where enzymes are present already in life, the digestive tract may have been a dangerous place because of ongoing autolysis. Simply put, the digestive system was digesting itself. We modified the text accordingly.

Remark: MAIN TEXT

Line 47- Small fossils and fragments of shells are densely distributed exclusively along the exoskeletal axial lobe

Please provide evidence that no other concentration of shell fragments occurs elsewhere within this trilobite.

Response: All the scan data were available to the referees and will be freely available to the readers once published, but in addition we have added cross-sections to Extended Data Fig. 1.

Remark: Line 64- Please provide images showing this ostracod species in the rocks where the trilobite with gut contents was found.

Response: The supplementary discussion explains that the origin of the specimen was allochthonous, i.e. it lived in a different environment to where it was buried, so this and the next two comments are not entirely relevant. However, ostracods are very common and occur

continually throughout the formation (up to 300 m in thickness) – see Lajblová and Kraft (2014).

Remark: Line 65- the genus *Elegantilites*

Same request here

Response: See above. It is one of the most common hyoliths in the formation. Hundreds of specimens can be collected at many localities – see Marek (1967).

Remark: Line 69- These fragments certainly belong to echinoderms (typical stereo pattern). However, please provide evidence that they do belong to stylophorans. Do you have stylophorans in the horizon where the trilobite with gut contents comes from. If so, provide images of a specimen and details of plates.

Response: Stylophorans are also very common in levels where our specimen comes from in the entire Šárka Formation (for review, see Lefebvre et al. 2007). They were discussed and figured in many papers (see e.g. Budil et al. 2007 and Fatka et al. 2015). Why the fragments are identified as stylophorans rather than some other kind of echinoderm is explained in the text.

Remark: Line 77-implying that the trilobite devoured at least a half of the digestive tract content in one place and time, and thus showed a fast-occasional feeding behaviour.

This sentence is very unclear. Nothing proves that the trilobite “devoured” stylophorans. Most likely, it fed on decaying carcasses (e.g. stylophorans with already dislocated plates).

Response: The word “devour” can be applied to dead animals and other foodstuffs as well as live prey. The Oxford English Dictionary offers the following relevant definitions:

1: To swallow or eat up voraciously, as a beast of prey; to make a prey of, to prey upon.

2: To eat greedily, eat up, consume or make away with, as food.

However, the sentence has been modified.

Remark: Food was presumably digested in the frontal part of the gut (cephalon; that’s how it worked in *Sidneyia*) then undigested fragments conveyed through the gut tract by possible peristaltic contractions.

Response: We are not saying anything that contradicts this comment.

Remark: Line 79- Why do you think these fragments belong to bivalves? See remarks above.

Response: Rephrased to “most probably” bivalves. Because of comparison with the fossil assemblage in the Šárka Formation (see Polechová 2013). Fragments of different fossil shells occur in the nodules in huge numbers so there is enough comparable material. Of course, they can be left unidentified, but calling them “probably bivalves” distinguishes those remains - similar to each other in form and character (e.g. thickness of the shell, mode of fragmentation) - from the incoherent group of “unidentified” fragments occurring throughout the gut.

Remark: Line 85- The accumulations of the devoured shells in the cephalic region form two sack-like ventriculi positioned dorsoventrally above each other between the hypostome and glabella (Fig. 3,

I see no distinct clusters of shell fragments that would support the presence of two ventriculi. In trilobite there is a unique spacious pouch within the cephalon that receives ingested food.

Response: We are describing what we see, not what is typical for trilobites. We see two distinct clusters of food remains in the cephalic region, one addressed to the internal cavity of

the hypostome, the other to the internal side of glabella side of the cephalon. We can discuss the interpretation, but the two separate clusters are clearly there and there is no way to explain their separation by taphonomy - the decayed trilobite was covered by sediment while lying on its back, so if gravity would affect the food remains within a single pouch, all fragments would be pressed against the ventral side of the exoskeleton, as is the case for the intestine infill.

Remark: The foregut of all arthropods is a tubular and muscular structure that served to suck up food. It has no role in concentrating food.

Response: This applies only to the oesophagus; as a general statement about the foregut it is incorrect. As we discuss in the manuscript, and support by references, many extant arthropods (e.g. decapods or horseshoe crabs) have a two-chambered ventriculus positioned between the oesophagus and the midgut – see, for example, Watling (2015) or Newman et al. (2021); for good examples in other arthropods, see especially Chapman et al. (2013), p. 38-40. Likewise, the proventriculus (crop) of the horseshoe crab serves only for storing food items (explicitly stated by Newman et al. 2021, p. 949). See also Zacaï et al 2016, page 215 “pear-shaped anterior part of the midgut that probably functioned as a temporary storage pocket” in *Sidneyia*.

Remark: Line 87-93- Speculation here, not supported by any fossil evidence.

Response: Fossil evidence is what we see in the scans, where the separation is obvious. We may discuss the interpretation but the observations here are clear.

Remark: Line 97- The ring of the third thoracic segment indicating the beginning of the hindgut

No. In arthropods, the hindgut is the terminal part of the gut tract, close to the anus. It cannot be there.

Response: Not correct, see for example Strus et al. 2019. However, the sentence was slightly modified.

Remark: Line 103- The locomotory appendages (endopods) of the trilobite must have been long and sturdy enough to raise the vaulted ventral side of the body off the substrate while moving.

It seems to be quite evident. Appendages are well preserved in *Sidneyia* an arthropod close to trilobites (see again Zacaï et al. 2016)

Response: The referee seems to misunderstand our claim. Of course, the trilobite had locomotory appendages. What is striking is the reconstructed shape of the anterior part of the body, which was unusually highly vaulted compared to other known well-preserved trilobite specimens. Therefore, we can safely infer that the appendages of *Bohemolichas* were unusually long and sturdy compared to other trilobites.

Remark: The presence such a grinding apparatus in *B. incola* was tested by comparing the relative size of fragments in both ventriculi and the gut.

In the extant *Limulus*, shells (e.g. muscles) are crushed and broken up by the strong gnathobases that occur at the base of their appendages (add recent refs). Comparable gnathobases are found in *Sidneyia* (Zacaï et al. 2016). I see no evidence of an internal grinding apparatus in the trilobite described here.

Response: We don't entirely understand why this point is being raised, as we are in fact in agreement with the referee.

Remark: Line 114-116- The ventral ventricular chamber of *B. incola* could represent a collecting space for ingested material prior to its passage into the dorsal chamber and gut for final digestion without subsequent grinding.

Quite speculative considering that there is no evidence of two distinct pouches.

Response: See above, for discussion on the collecting space in the proventriculus of *Limulus*, see especially Newman et al. 2021, p. 949. See also Zacaï et al 2016, page 215 “pear-shaped anterior part of the midgut that probably functioned as a temporary storage pocket” in *Sidneyia*.

Remark: Line 124- Digestive glands are well-preserved (including internal diverticles) in *Sidneyia*, an arthropod close to trilobites, where gut contents occur. Curiously this key paper is not cited in this MS.

Response: See above, Zacaï et al. (2016) is now cited in all relevant points in the manuscript. There is a number of papers dealing directly with the digestive system of trilobites and some of them are cited in our manuscript. All of these animals are much more closely related to *Bohemolichas* than to *Sidneyia*.

Remark: Line 129- The almost entirely filled digestive tract combined with the ‘crumble’ reflects its simple J-like shape and allows the tract to be reconstructed (Fig. 5).

This sentence is unclear + please don’t use “crumble”; choose another term.

Response: We don’t see what is unclear. One of the co-authors (P.K) was the first to describe this phenomenon occurring in the Šárka Formation, see Kraft et al. (2020, p. 18): “Many specimens with feeding traces preserved as internal moulds are also associated with a unique taphonomic feature. It is represented by sponge-like aggregates fully or partly filling shell cavities, i.e. occurring in part of a mould or substituting the whole mould (seen in some parts of specimens in Figs. 8C, 13D and 14F–G). The preliminary study of it (in prep.) shows that its topology, as well as feeding traces reflect remains of decaying tissues.” We only use a new term that better reflects its appearance.

The previous paragraph of the manuscript was modified for greater clarity.

Remark: Line 136- taxa have no size. Size applied to individuals.

Response: Untrue, taxa usually have characteristic sizes. For example, *Brachiosaurus* is a larger taxon than *Compsognathus* and we are smaller than elephants. By the way, see Zacaï et al 2016, p. 217: “*Sidneyia* was a relatively large ... arthropod”

Remark: Line 139- fragments of large, thin-walled shells (probably bivalves), Why do you suppose they are large? You don’t even know that the fragments belong to bivalves.

Response: The word “large” was deleted.

Remark: Line 139-142 The diet composition indicates a limited ability of the trilobite to break firm or robust inorganic shells, based on the absence of a concentrated assemblage of devoured fragments in the intestine, which could be identified as pieces of a complete crushed shell.

Odd here. The gut of the trilobite does contain a high concentration of fragments apparently! The assumed bivalves seem to be broken into pieces.

Response: The referee has misunderstood the text. We are talking about a verifiable individual broken into pieces. What is preserved in the gut infill are indeed fragments. They could have been collected already broken and originating from many diverse and distant

individuals, or have originated from a few thin-shelled individuals. The sentence was modified for greater clarity.

Remark: Line 142-143- This accords with the non-biomineralized nature of trilobite gnathobases that were used for food processing/mastication

Problem here too. *Limulus* has strong non-biomineralized gnathobases with strongly sclerotized teeth that are able to break muscles (shell) into pieces. Same with *Sidneyia*. The gnathobases of *Sidneyia* are non-biomineralized elements that were able to break small trilobites into pieces (their remains are found in the gut). Again, this paper and other related one are not cited!!

Response: We don't see any contradiction between the referee's claim and ours. However, concerning the "strong" gnathobases, see Zacaï et al 2016, page 209 "majority of ingested trilobites belong to juvenile stages"; page 211-213 "The selectivity towards feeding on immature individuals might have been related to ... their gnathobases ... not powerful and/or resistant enough to break the heavily mineralized cuticle of adult trilobites"; page 215 "the grinding power of gnathobases was however limited since no thick sclerotized or mineralised fragments are found within the gut contents" and page 217 "its indiscriminate feeding mode ... must have been strongly constrained by ... the grinding power of ventral appendages".

Remark: Line 143-146- The larger valves reflect a clear preference for fragile elements which can be easily broken. Complete vertebral like marginal plates of a stylophoran indicate small size preference of solid elements (Fig. 2b, Extended Data Fig. 6)

These sentences are very unclear. Please rewrite.

Response: The sentences were modified.

Remark: Line 148- Durophagy is primarily an ability to crush inorganic parts of organisms with an aim to reach the soft, digestible tissues

Not necessarily inorganic. Durophagy refers to animals that have mouth parts or other appendages adapted to break hard skeletons (biomineralized or not)

Response: The paragraph was modified.

Remark: Line 149- the ingestion of shell fragments is a side effect of this behaviour rather than intention

Delete this and avoid intention. There is no side-effect here. Potential prey or carcasses were detected via chemical or visual cues. Small items were ingested whole (e.g. ostracods) others are broken into pieces and swallowed.

Response: This is only a logical conclusion of the statement above. The intention of a typical durophagous animal (e.g. an octopus opening a clam) is to avoid eating the hard parts. If hard parts are ingested, it is probably a side effect of food manipulation. In case of *Bohemolichas*, we can safely assume that ingesting the hard bits was also a side effect of its feeding mode, but it was apparently not as strictly avoided. If the ingestion of hard shells in such quantity was not a side effect of the feeding mode, what would be the reason for such a behaviour? The sentence was modified.

Remark: Line 150- However, the large accumulation of hard particles in the intestine of *B. incola* reflects a different feeding strategy.

Why different, I don't understand. Different from what?

Response: Different from *Sidneyia*, for example, in which the majority of the fragments is located in the so-called abdominal pocket (see Zacaï et al 2016, Table 1, page 209), not

continually throughout the whole digestive tract as is the case in *B. incola*. The sentence was modified and the reference was added.

Remark: Line 151-154- The studied trilobite was polyphagous with a low degree of food particle preference in their positive frequency dependent proportion¹, which means that the food selection was based rather on size and shell resistance than on the taxonomical composition.

The way this sentence is written is odd. Moreover, it does not make much sense.

Response: The beginning of the sentence is a direct citation, see reference, which is subsequently explained.

Remark: The size of ingested items is controlled by the diameter of the mouth opening and oesophagus diameter. Shells are broken into small pieces if the gnathobases and associated muscles are powerful enough to crush them. The animal is unable to detect which shell is less resistant than others. The process is purely mechanical. Instead you should consider that this trilobite was an opportunistic feeder (predator or scavenger or more likely both) as was *Sidneyia* (Zacai et al. 2016) or *Ottoia* (Vannier 2012). Both *Sidneyia* and *Ottoia* have well-preserved gut contents that make it possible to characterize their diet.

Response: We don't see how our claims contradict the referee. *Ottoia* is an unrelated animal.

Remark: Line 154-Thus, it contradicts a typical durophagous strategy, which is usually specialized on a limited variety of shelly animals.

Absolutely not true. The definition of durophagy does not imply that ingested food is not diverse! Please delete.

Response: Durophagy refers to a feeding strategy or ability to consume prey with hard shell or skeleton. It is a very specialized strategy with many morphological and ethological limits and constraints. The diet of a durophagous organism is usually oligophagous rather than polyphagous because of the adaptations that made durophagy possible - these also limit the adapted organism to some extent, representing a toll for this specialization (see also Langerhans et al. 2021 or Schweitzer & Feldman 2010 on molluscivory). However, we slightly modified the statement.

Remark: Line 156- Because *B. incola* was a non-selective feeder it can be inferred that it was a scavenger. If predatory behaviour was present it was only occasional because predatory arthropods are usually also selective in their diet preferences.

Yes, it is correct that the trilobite in question may have been a scavenger. However how would you explain that broken shells of assumed bivalves occur in its stomach.

Response: Because it was an occasional predator. The sentence was modified.

Remark: Line 158- It can be considered as a light crusher – a durophagous chance feeder hoovering dead or living organisms, adequately sized or thin shelled.

Unclear sentence. Please rewrite

Response: The sentence was modified and moved to another place in the text.

Remark: Line 159- adequately sized or thin shelled.

Trilobites had no capacity to detect whether an invertebrate was thick or thin shelled

Response: Yes, they had, in the sense that they only ate whatever fitted their mouth opening or what they were able to diminish, which is a "capacity". For discussion on such a capacity and diet preferences of *Limulus* see Botton 1984.

Remark: Line 163- Even though lacking the strongly calcified appendages typical for durophagous decapod crustaceans, which enabled this malacostracan clade to spread the feeding strategy widely since the Upper Cretaceous, trilobites occupied this niche some 250 million years earlier.

The example of *Sidneyia* (Zacai et al. 2016) shows that durophagy existed much earlier. Please mention this important point.

Response: We are not saying that it was the first occurrence, we are pointing to *Bohemolichas* occupying a comparable niche to crustaceans. Naturally, durophagy existed much earlier (see for example Fatka et al. 2015 and Bicknell et al. 2021), but the gut contents of *Sidneyia* demonstrate that its feeding mode and processing of digested food was not comparable to modern crustaceans. On the contrary, the authors compare its gut morphology to modern arachnids and related forms (Zacai et al. 2016, page 205 “so called stercoral pocket”).

The paragraph was modified.

Remark: Line 169- Digestion of food composed of organic tissues along with the associated inorganic shells is highly specialized and requires deep adaptations.

No, exoskeletal elements are not digested. Only soft tissues can be digested. Please rewrite or delete. I don't see why deep adaptations are required.

Response: See above, our response about calcium carbonate shells in an acid environment. The exoskeletal elements cannot be digested, but they can be dissolved by digestive fluids. The sentence was modified.

Remark: Line 170 - On the other hand, it offers a rich food supply with limited competition. This sentence does not make much sense.

Response: We don't understand this comment. If an organism is able to feed on hard shelled fragments and scraps ignored by other scavengers, this is certainly an advantage. We can say that the competition in this niche is low. An obvious modern example is the lammergeier, *Gypaetus barbatus*, which is the only vulture specialised for eating bones.

Remark: Line 179 -the two-chambered ventriculus associated with a strong musculature while covering and protecting the foregut

Ventriculi not evidenced (see remarks above). Any evidence of muscles in your specimen.

Response: See above, the presence of muscles can be logically assumed.

Remark: Line 182- adaptation allowing the passage of large undigested organic particles through the anal opening, including necessary space for large extensor muscles.

Speculation here- Please go back to your specimen and try to determine the diameter of the most posterior part of the gut, whether it was capable to expel small or larger exoskeletal fragments.

Response: We are talking about exoskeletal features here. This is not a speculation but an attempt to explain why the morphology of the pygidial axis is so peculiar in lichid trilobites. Now, when we know more about their diet, the blunt end of the pygidial axis in lichid trilobites makes more sense. If the trilobite was not able to expel fragments of the size present in its intestine, it would overflow and die. We do not record any blockage in our specimen, so we can safely assume that overeating was not the cause of death, but a physiological behaviour. The morphology of its exoskeleton perfectly matches our assumptions on the feeding strategy.

Remark: Line 183. An increased enzymatic support, e.g. a hepatopancreas and other digestive glands¹⁷, is required for maximum energy exploitation from a diet rich in indigestible bits scarcely covered by organic films or decaying tissues. The importance of digestive glands in the development of carnivory is discussed in Vannier et al. 2014 (Nat Comm). Please add this ref and discuss in more details.

Response: Requested reference added.

Remark: Again, enzymatic reaction has nothing to do with indigestible exoskeletons. Enzyme breakdown molecules present in soft tissues.

Response: Yes, it does, the presence of large quantities of indigestible bits points to a specific range of pH values inside the gut, which is directly connected to enzymatic activity, see above.

Remark: What do you mean by scarcely covered by organic films or decaying tissues? Please rewrite. Nobody will understand.

Response: The sentence was modified.

Remark: Line 186

This sentence sounds odd. Please rewrite There are many better refs than 17!!

Response: Reference 17 (now 20; Ceccaldi 1989) is a complex and comprehensive overview that includes morphology and physiology. Although the reference may be considered rather old, its content is valid and the data are discussed in context.

Remark: Line 189- In addition, robust external organs for manipulation and mechanical grinding of inorganic shelly material must have been present.

If you had read Zacai et al. 2016 and other related papers (Bricknell, etc..) you would know that they were present.

Response: They are not preserved in our specimen; we can infer their presence, but we cannot say with certainty that they were present.

Remark: The occlusion of the hypostome with the unpreserved gnathobases is indicated by its robust structure and a wide, double-walled posterior margin.

Very unclear sentence

Response: The paragraph was modified.

Remark: Line 194- The hypostome could have served as a splitter or even a greater utilizing its prominent rough sculpture

Highly speculative. The hypostome is above all a protective element in the head region.

Again, food processing (crushing) is due to gnathobases that have opposed crushing margins within the midline of the animal (see extant *Limulus*).

Response: The posterior margin of the hypostome as a site for further grinding of food after its processing by gnathobases has been widely discussed (Chatterton & Perry 1983, Bruton & Haas 1997, 2003, Budil et al. 2008 and Hegna 2009) in diverse trilobites. Clusters of spines or coarse terrace lines allowing grinding of the food have been observed. Similar structures have been observed also in lichenid trilobites (Budil, unpublished observations, e.g. genus *Acanthopyge*). All these structures are compatible with the grinding function, suggesting a possible limb differentiation to complement this structure. The forked structure of the

posterior part of the hypostome suggests a similar function (for a discussion of analogous, almost identical-looking structures in hypostomes of asaphid trilobites, see Hegna 2009).

Remark: Line 197- functional analogy to the gastric mill or gizzard of extant arthropods) although a necessary limb specialization has not been fully demonstrated

Vague statement (gastric mill). What do you mean here?

Response: See manuscript, lines 107-108.

Remark: There is absolutely no need of specialized appendages. Look at *Limulus* or *Sidneyia*. Food is processed by the gnathobases of a series of homonomous appendages.

Response: See reference (Hegna 2010), the sentence was modified.

Remark: Line 199 +-The clusters of shell remains below the axis are distributed in 199 indistinct faecal pellets.

Unconvincing. I see no pellet in any image show in this MS. What you have here are the gut contents only.

Response: That's why we say "indistinct" but in the almost continuous infill there are three narrow but clearly empty disjunctions. (Note also the influence of the post mortem flattening of the gut discussed in the text.) Thus, the individual clusters can be classified as faecal pellets.

Remark: Line 203- also for their chaotic huddles.

??????

Response: This part of the sentence was deleted.

Remark: Line 204 A disordered orientation of the shells or fragments of anisotropic shapes indicates peristaltic movements but

Pure speculation. No evidence at all of peristaltic contractions !!! such as for example beads in certain fossil and extant arthropods (crustaceans). Delete.

Response: The paragraph was modified.

Remark: Line 205- it also clearly points to the presence and essential role of a protective cover of the large pellets inside the gut, such as a peritrophic membrane.

Since there is no convincing support for pellets, the hypothesis of peritrophic membrane does not make any sense

Response: See above.

Remark: Line 207- You forget that the whole trilobite biomineralized exoskeleton is preserved. There is absolutely no reason to suppose that special PH-conditions prevailed within the gut. Please delete.

Response: We disagree. The exoskeleton of the trilobite is exposed to the pH of the surrounding sea water; the shelly gut contents are exposed to the pH inside the gut. These are completely different things, and in a sense the pH of the gut is always 'special' in that it is not a mere reflection of the environment – even when, in fact, it is close to the environmental value.

Remark: Line 209- an acid gut environment the organism would face an ion imbalance and a dangerously high concentration of extracellular Ca⁺ (hypercalcemia).

Very obscure- Delete or rearrange the discussion with examples taken from detailed studies on extant arthropod gut contents. Why “dangerous”. Avoid these terms.

Response: Because hypercalcemia is dangerous for the organism affected. The sentence was modified.

Remark: Line 211. That’s absolutely not true! Please read Zacaï et al. and other articles cited in this paper. For example, the gut of modern limulids stores microspherules of Ca and P. (see figures in Zacaï et al.). It is the reason why many Cambrian arthropods have their digestive system preserved in apatite. The source for Ca and P comes from inside the animal.

Response: Yes, but our claims are not in contradiction. The presence of microspherules in cells (Zacaï et al 2016, p. 217) and the extracellular calcium are two completely different phenomena. The capacity of the midgut cells to store calcium in form of microspherules is much lower than the amount of calcium that would be released by dissolution, if the environment inside the digestive tract was acidic. In that a case, swallowing such an amount of calcium carbonate shells would lead to an extensive presence of calcium ions polluting the inner environment of the body and causing irreversible damage to the vital functions of organs.

Remark: Line 212-215

This sentence is particularly unclear!

Response: We disagree.

Remark: Line 215 +The studied specimen illustrates the potential of fossils to reconstruct life processes of extinct invertebrates and, on the other hand, points to an insufficient knowledge about many aspects of modern marine arthropods.

This sentence means nothing. Delete.

Response: Its meaning is unambiguous and important.

Remark: Line 217- The digestive tract of recent arthropods has been studied in detail with respect to its morphology, anatomy, food content, enzymatic processes and other features (e.g. for decapod crustaceans)

What kind of relevant information do you want to provide here? It would be far more interesting to discuss the several cases of preserved gut contents and digestive systems in early Palaeozoic arthropods (see Vannier, Lerosey-Aubril and collaborators).

Response: We disagree. Our study is about trilobites, which we compare with extant related forms that can tell us more about functional morphology and physiology. We see no reason to compare trilobites with distantly related arthropods preserved only as fossils. On the contrary, discussion of recent analogues is essential because it is the only way to avoid the circular argumentation of fossil interpretations built on other fossil interpretations, without any connection back to real data from living organisms.

Remark: Line 219- However, there are surprisingly few data on the pH values and dynamics inside their digestive tract, even though they directly influence all the mentioned characteristics. The discussion therefore has to be limited to a few model organisms.

Do you really expect that the chemical composition of mineralized guts of Ordovician trilobites will tell us about the Ph-conditions that prevailed in their digestive systems? The last sentence is obscure.

Response: We are talking about a striking lack of knowledge about certain aspects of physiology of extant arthropods, which is an interesting observation to note. One would think

that a comparison with recent forms would be much easier, considering they are still around. But no, surprisingly few records on the gut pH exist. The reader should be informed about that. Hence, we can only compare with two model organisms. We do not understand why the referee finds the sentence obscure. We have explained above the basis for our estimation of pH values in *Bohemolichas*. The condition of the calcium carbonate shell fragments within the gut (which is not itself 'mineralised') is certainly informative about the gut pH of the living trilobite.

Remark: Line 225 + As decapod crustaceans and chelicerates respectively, they bracket both major lineages within the euarthropod crown group²³. Trilobites are generally agreed to fall within this bracket²⁴

Very unclear again here. Where do you place trilobites? According to the most recent phylogenies (e.g. Aria 2022), trilobites belong to Artiopoda. Artiopoda are placed somewhere between chelicerates and mandibulates (see tree in Aria's paper)

Response: Our phylogenetic statement is exactly equivalent to that made by the referee.

Beyond that, we don't pursue the phylogenetic placement of *Bohemolichas* and trilobites generally, because this is not a phylogenetics paper; what matters in the present instance is that trilobites fall within the extant phylogenetic bracket of [crustaceans + chelicerates], meaning that physiological characters shared by crustaceans and chelicerates can, with caution, also be attributed to trilobites.

Remark: Line 230. In digestion processes, it is the enzymes that matter!

Response: And enzymes have highly specific pH requirements. See above.

Remark: Line 232. That's pure speculation based on no detailed evidence. Delete

Response: It is a logical conclusion; see above.

Remark: Line 233. Different stages of digestive tract infill by food and other particles were recorded in mud crabs, ranging from an empty to a gorged intestine²⁷, which can partly be a seasonal factor.

What do you mean here by seasonal? Yes, food composition may vary according to the season (as in birds). But quantity of ingested food is another matter. When food is abundant, crabs eat a lot and store food. When food is scarce, they stomach is often empty. Rewrite.

Response: See reference.

Remark: Line 235- The authors don't seem to know about the complexity of the moulting process.

Response: One of the co-authors (PB) has studied the phenomenon of trilobite moulting for over twenty years (Bruthansová and Budil 2003, Budil and Bruthansová 2003, 2005, Drage et al. 2018 etc.), all authors are aware of the complexity of moulting. The second referee is an experienced trilobite specialist and has no reservations about this interpretation.

Remark: Line 204- I don't believe this. Vaulting may have nothing to do with moulting. Preservation of arthropods in the act of moulting is extremely rare (see Garcia Bellido for Marella). Vaulting is more likely to contractions just before death. Please don't take this hypothesis for granted and be more critical here.

Response: See our references. Also, we are taking about the premoulting phase, not the act of moulting. The paragraph was modified.

Remark: Line 241- The combination of these features with the gorged digestive tract suggests that the studied specimen of *B. incola* represents a premoulting phase initiated by rupture inside the thorax through an expanded intestine.

It is very weakly supported and speculative. Please refocus your paper on fossil evidence and delete all that speculative stuff that ruins the strength of the MS.

Response: The sentence was modified.

Remark: Line 242- When an arthropod is about to molt (ecdysis) the old and new cuticle start to split off. I see nothing suggesting this stage in the trilobite presented here. Your molting hypothesis is not convincing.

Response: We don't understand this comment. Our argument, as usual in trilobites, is based on the configuration of the individual parts of the exoskeleton. You cannot expect to observe the process of forming new cuticle because taphonomical effects will almost certainly prevent its preservation. Interpretations of fossils are, in this respect, completely different from examining extant animals.

Remark: Line 245- indicated by a limit between two faecal pellets, reflecting different food sources and feeding periods but also different feeding intensity.

It does not make much sense since no evidence of pellets exist. Again, such poorly supported statements weaken your MS.

Response: The sentence was deleted from the main text, modified and moved to the supplementary discussion.

Remark: Line 246- We infer that the feeding behaviour of the trilobite resembled the corresponding life cycles of modern crustaceans: most of time the intestine was empty or moderately filled, with occasional and swift overfeeding actions linked to specialized physiological requirements.

Extremely speculative and based on no extant analogues. The only thing we know is that trilobites grew in size by molting in a manner similar with that of modern crustaceans. The amount of ingested food within gut was certainly highly variable and depended on the amount of a food available on the sea floor. Nothing suggests that food intake is related to molting.

Response: No, it is based on a reference, which we added.

Remark: Line 249- The animal was vulnerable during the premoulting and moulting phases, which is connected to a high mortality rate also in modern ecosystems and attempts to reduce it in aquacultures.

Please delete. Everybody knows that arthropods that have just discarded their old carapace are relatively soft and for this reason vulnerable. What has aquaculture to do with the topic of this MS?

Response: The sentence was deleted.

Remark: Line 251- The gorged specimen represents a snapshot directly documenting this specific life period.

No. Not supported. Pure speculation.

Response: The sentence was deleted.

Remark: Line 254. Life after life

You believe in reincarnation? Avoid these terms.

Response: We are unable to respond to this comment.

Remark: Line 259- This tissue was stretched and torn due to the hypostome dislocation as clearly marked by a thin, discontinuous layer of ‘crumble

????? What makes you believe that this kind of supposed tissues occurred here

Response: We describe what we see. It is not a question of belief.

Remark: Line 261- Scavenging activity is generally related to the ‘crumble’ as a demonstration of decaying tissues¹², unlike in the studied specimen.

This sentence makes no sense at all. What do you mean by “crumble”???

Response: The sentence was modified and the paragraph was extended.

Remark: Line 262- The distributional pattern of a number of thin vertical burrows indicates not only the preferred targets but also areas that were strictly avoided

Do you really think that these features were made by scavengers? If so, what kind of scavengers? Borers? These features could be simply cracks with no biological origin. This paragraph is hard to follow especially because the nature of these assumed microscavengers is not discussed. If this trilobite was a carcass exposed to scavengers why did they spend energy drilling through a thick biomineralized cuticle (glabella) whereas obtaining food was far easier among the decaying tissues that surrounded the animal.

Response: See Kraft et al. (2020) and above. Cracks in the sediment (and also in nodules) look completely different. The producers of ichnofossils are often unknown because it is complicated to determine the producer of a track based solely on the morphology of the track. Here, it is likely to be a representative of the ichnogenus *Pilichnus* (for discussion on the occurrence of this genus in the nodules of the Šárka Formation, see also Mikuláš 2003). The paragraph was modified.

Remark: Line 275- An analogous organ is highly probable in lichid trilobites

Also naraoids have such well-developed diverticles (check refs)

Response: We prefer to focus on comparisons with extant animals, where the anatomy and function can be studied directly.

Remark: it also represents a nutrient target that would not have to decay in order to be fed upon by a weak mouth apparatus, as opposed to, for example, tough muscle tissue, tendons or epidermis.

This sentence is not understandable

Response: The sentence was modified.

Remark: Line 280- What do you mean by dangerous? Think that the cephalon does not house only a digestive pouch but a variety of other tissues such as muscle, digestive system, haemolymph network. The problem here is the nature of your supposed scavengers. What are they? Is there anything similar in extant animals, other trilobites (check)?

Response: See Kraft et al. (2020) with numerous examples; on the nature of ichnofossil producers, see discussion above (Line 262). Polychaetes are regarded as being the producers of *Pilichnus*, *Arachnostega* and other similar ichnofossils (see Rodríguez-Tovar et al. 2020, Bertling 1992). The sentence was edited to make it easier to understand.

Remark: Line 282 The distribution of traces suggests that the digestion processes throughout the entire tract continued until the nodule was formed in the shallow sediment and only ceased during the diagenesis.

No. Totally unconvincing. Supported by no evidence. Delete. It is pure speculation

Response: It is evidenced by described observations. See also references dealing with the origin of nodules. However, we slightly modified the text.

Remark: Line 284-285- The trace makers had to avoid the digestive tract to evade digestion by the dead trilobite. The behaviour of scavengers directly supports the model suggesting a rapid formation of these nodules³⁵.

Idem here- It is a nice story but nobody will believe it I am afraid. Please stick to facts and observations!

Response: See above, especially the discussions in Kraft et al. (2020) and Mikuláš (2003).

Remark: Line 287- The described specimen of *Bohemolichas* provides by far the most detailed source of information to date concerning the diet and the feeding mode of trilobites. I agree that this MS tells us about the diet of the trilobite *Bohemolichas* but much less concerning its feeding mode (no appendage, no soft parts). As I wrote before, the diet of *Sidneyia* an arthropod closely related to trilobites was described in great details in 2016 (Zacai, Vannier et al). Comparisons with *Sidneyia* would have been extremely interesting. Unfortunately, and curiously this paper was ignored by the authors...

Response: See above. *Sidneyia* is not a trilobite. It belongs most probably to the vicissicaudata, *Bohemolichas* is a trilobitomorph. Although they are both arthropods and artiopods they represent different lineages. *Bohemolichas* is the oldest fossil arthropod with the digestive system including its contents preserved and visualized in full 3D (comp. Zacai et al 2016, page 203 “most anatomical and exoskeletal features ... are strongly compressed”).

Remark: Line 289 It captures two interconnected but discrete moments, life and after life, frozen in one nodule.

This sentence is obscure. “After life” sounds very odd. Delete

Response: The sentence was modified.

Remark: Line 290- One points to the complex digestive physiology related to a specific short moulting stage in the life cycle of the trilobite. The other documents a post mortem endurance of metabolic processes reflected through a snapshot of constrained targeting of scavengers feeding on its corpse.

Not understandable. Rewrite or delete.

Response: The sentence was modified.

Response to referee #2

Remark: Kraft et al present an exceptionally well-preserved trilobite that was synchrotron scanned. Overall, the paper is written very well, it hits a lot of key points that are assumed but lacking in the fossil record and overall, I applaud the authors for their work, this is spectacular. My only main comment is: I would very much like to see this as a 3D PDF, or ideally STLs. This kind of data are ideal for subsequent work on 3D analyses and being able to replicate what is done, or ideally use the data produced here, would allow for this work to have even more longer-term impact. I appreciate the authors may not want to share this data, but I ask that they consider doing so.

Response: Two 3D PDFs and all STLs along with the raw tomographic data were uploaded to dropbox (<https://uppsala.box.com/s/n71426jvkbmg7ljaxh47nnl252e1ob2l>). The link was included in the cover letter and should have been provided to the reviewers. If the manuscript is accepted all the data will be made public, as stated in the section Data availability.

Remark: Originality and significance: This is exceptionally novel and deserves publication
Data & methodology: All sound, but as I mentioned above, the 3D data should be made available

Response: See above.

Remark: Appropriate use of statistics and treatment of uncertainties: These were limited, but seem appropriate.

Conclusions: Very strong, some minor tweaks needed

Suggested improvements: experiments, data for possible revision: see below.

References: appropriate credit to previous work: more than fine.

Clarity and context: very well written.

I have a few minor comments below that I think could be addressed quickly. A few (such as the last one) are more philosophical based on my own interest in the studied groups, so if the authors would like to keep such terms as 'living fossil' in the text, it is their choice.

Yours,

Dr Russell Bicknell.

Response: Thank you for your fair and qualified review!

Remark: 19: "not a single trilobite gut content has been described". I know what is implied here, but it might be better to say that "no single trilobite specimen with internal gut contents has been described" as gut contents have been noted (see Gut content fossilization and evidence for detritus feeding habits in an enrolled trilobite from the Cambrian of China, Zhu et al 2014, Lethia)

Response: The sentence was modified.

Remark: 135-137 Ostracods are all not benthic, so are these possibly remains?

Response: All ostracod remains in the gut belong to a single species, which is considered benthic. See e.g. HENNINGSMOEN, G. 1965. On certain features of palaeocope ostracodes. Geologiska Föreningens i Stockholm Förhandlingar 86, 329–394.

SIVETER, D. 1984. Habitats and mode of life of Silurian ostracodes. Special Papers in Palaeontology 32, 71–85.

TINN, O., MEIDLA, T. & SOHAR, K. 2010. Intraspecific variation and polymorphism in the ostracode *Conchoprimitia socialis* (Brøgger, 1882) from the early Middle Ordovician Baltoscandian Palaeobasin. *Bulletin of Geosciences* 85(4), 603–616.

Remark: 142-143: Bicknell et al looked at different trilobites in this paper, one of which was proposed to be a shell-crusher based on re-enforced gnathobasic spines. This may need to be reworded to reflect that, but it is very much the case for *Olenoides*.

Response: The information was added.

Remark: 156-158: “If predatory behaviour was present it was only occasional because predatory arthropods are usually also selective in their diet preferences¹⁴.” I like this. However, the issue with this paper is the examples are mostly terrestrial hexapods. So I think another example here is needed, or possibly some more unpacking of the situation as this comparison is not the most appropriate when looking at trilobites.

Response: Additional reference was included.

Remark: 161-163: this comparison with decapods is maybe not the best. Better to look at xiphosurids that feed very similarly.

Response: The paragraph was modified and references to Botton (1984) and Newman and Smith (2021, p. 949) were added. We are discussing the occupation of similar niches here, not the feeding mode sensu stricto.

Remark: 191-195: the lack of gnathobases here does not immediately imply that the hypostome would be used in this situation. I think the lack gnathobases here reflects the lack of appendages in the scan. This should be tweaked a bit.

Response: Here we were discussing a possible functional morphology of the hypostome. Of course, the gnathobases should be expected in the specimen, but they were not preserved, so we can only assume them. We state that in the manuscript. We softened the statement to indicate that this is only a possible explanation not a certainty.

Remark: 225: “and perceived status as a 'living fossil’”. Personally, I think this is incorrect. Living fossil is an oxymoron in this context as there are not fossil *Limulus polyphemus*.

Response: Suggestion accepted, the phrase was deleted.

Reviewer Reports on the First Revision:

Referees' comments:

Referee #1 (Remarks to the Author):

In my first review I spent time suggesting things that may improve the scientific content of this MS (typically less speculation, open the debate beyond trilobites, focusing on the diet of trilobites and their role in food chains). The authors ignored the majority of my remarks. There are many points throughout the revised version of this MS (except description) that need to be improved and rewritten. Please see below:

TITLE

These well-preserved gut contents tell us about the diet of trilobites not their physiology. Physiology is the study of the functioning of living organisms, especially their constituent tissues and cells. The present MS is based on fossil evidence and does not illuminate any mechanism at tissue or cell level. "Palaeophysiology" is not correct in this context. Moreover, this term does not exist. Please change the title.

ABSTRACT

"fragmented calcareous shells indicating a voracious feeding activity"

As I wrote in my first review, "voracious" is not the best term. To me, trilobites simply scanned their environment and ingested all edible items they came across, including food from decaying carcasses.

Line 56

Better write that this concentration of undigested elements roughly corresponds with the outline the digestive tract (for general information on morphology see e.g. Leroosey-Aubril's papers)

Line 87-88

"consumed at least a half of the digestive tract content" does not really make sense. Please rewrite. I see no evidence that the trilobite consumed everything in one go. A more realistic view is that trilobites scanned the bottom sediment and randomly fed on everything they came across. Extant ostracods when present are generally very abundant at the water-sediment interval. No wonder that they occur in large number in the trilobite's gut. Trilobites also probably fed of decaying organisms (e.g. echinoderms). The view of trilobites being voracious predators is probably not correct. Please add details and rewrite.

Line 92

Why "most probably" here ? Is there any doubt or alternative option that would be worth being mentioned ? If so, please give details.

Line 94

Are the gut contents homogenous in composition (i.e- elements of various origin randomly

distributed and all mixed up) or do you see any local concentrations of echinoderm/ostracods/shell remains. It is an important point that is not addressed here. The way how elements are distributed within the gut may provide key information on the feeding strategy of the animal. Please add information.

Line 97

I would rather say that the gut was extensible. For comparison, the gut of *Sidneyia* (an arthropod closely related to trilobites) is often very narrow when not filled with food (see Zacaï et al.). Please improve here.

Line 101-105

Please add refs and information on how the digestive system of trilobites is usually reconstructed by the specialists of the group. Key refs are missing here. Then write that the distribution of the gut contents is consistent with what we know about the anatomy of the digestive system. Please rewrite this part.

Line 108

You are inferring fine details of the anatomy of the digestive system from the spatial distribution of undigested fragments. Please discuss the limits of this kind of inference.

Line 119

Here again, the first thing to do is to give basic information on the appendages of trilobites or related taxa. Refs exist on the appendage morphology and their possible role in feeding (see also *Sidneyia*, an arthropod closely related to trilobites).

Line 120

Trilobites are crawling not creeping animals. Again refs are available in the literature on locomotion in trilobites.

Line 126

Zacaï et al. provided images of a longitudinal section through *Limulus*. Please refer to this paper here. Comparisons with limulids are much more relevant than those with derived decapods. Another recent paper on *Sidneyia* (Bicknell et al. 2018) also provides key information (and illustration) on the gut contents (muscle shells) of extant *Limulus*.

Line 131

There is not enough discussing on the possible grinding mechanism by ventral appendages (gnathobases). It is surprising that the authors of this MS do not refer in details to key papers on the feeding behaviour and diet of Cambrian arthropods (Bicknell et al. 2018, Zacaï et al., etc.). This problem has already been addressed when I first reviewed the MS. The MS would have a much better impact if enriched with discussions based on previous work. The authors argue that *Sidneyia* belongs to a separate group. They are perfectly right. However, trilobites and *Sidneyia* both belong to Artiopoda (see Aria's most recent phylogenies in 2022) and share important features (digestive system, appendage morphology, etc.).

Line 132-133

Concerning the diet (direct evidence from gut contents) of Cambrian priapulid worms please add Vannier 2012. "Ecdysozoan" is too vague.

"Thus, food may have been ingested at speed, but digested at leisure, as documented in other fossil ecdysozoans¹¹"

You have no strong evidence to say that (i.e. how fast trilobites processed and ingested food, how fast food was broken down by enzymes). Please delete. It would be more realistic to write that the animal probably kept on ingesting food until its stomach became full without any consideration of speed. Please delete sentences that suggest a kind of "intentional" behaviour.

Line 134-138

These sentences are very unclear and not correct (language problems). Please rewrite. It is clear that shapeless aggregates frequently occur in rocks throughout the Ordovician of Bohemia. As far as I know there is no Burgess Shale type Lagerstätte in Bohemia. Why do you think that some of the aggregates represent in situ decaying organisms? Most of these skeletal elements were most likely accumulated by currents. Please rewrite these sentences accordingly.

Line 139 +

"In the thorax of our specimen, the 'crumble' is limited to the axial part reflecting the decaying tissue of the gut"

This trilobite specimens clearly show undigested elements throughout the gut but no remain of any soft tissue. Please clarify this important point. "Crumble" sounds odd here. Does the "crumble" represents mineralized tissues? If so, what kind of mineralization (please indicate composition).

IMPORTANT: another thing concerns the chemical composition of these undigested elements. Please provide elemental maps or at least an EDS-analysis of some of the elements. The digestive system of numerous Cambrian arthropods is preserved in phosphate (glands, tract) including gut contents when present (see Zacaï et al, Zhu and Vannier on phosphatized gut contents from Chengjiang, Biology Letters; papers by Lerosey-Aubril). It is absolutely essential to know about the chemical composition of the trilobite gut contents. This has implications on your interpretations.

Digestive glands in trilobites or related taxa do not occur throughout the digestive tract. Instead they are concentrated anteriorly. I doubt that your so-called "crumble" (light green in model) represents the remains of soft tissues such as digestive glands.

Line 163

"no parallel either in the coeval fossil communities".

Not really true. It is surprising that you keep on ignoring or minimizing previous work on the gut contents of Cambrian arthropods. You would make your MS much more interesting if you went beyond trilobites. Sorry for repeating it again and again, Zacaï et al. described well-preserved gut contents in *Sidneyia* (juvenile trilobites among other elements). These elements are as well preserved as the gut contents presented here. Please improve this part simply to be scientifically honest.

Line 164

These elements were originally calcareous but nothing proves that are mineralized in calcite. The

exoskeletal remains of trilobites inside the gut of Sidneyia are phosphatic although they were originally calcified. It is an important point to consider.

Line 168

“based on the absence of an assemblage of fragments in the intestine”

It is not very logical here. If the animal is unable to crush hard shell, then how do you explain the occurrence of shell fragments in its gut. Please change.

Line 169-170

Very unclear. Please rewrite (language problems)

Line 170

The gnathobases of Limulus are not biomineralized, only sclerotized. However, they are able to crush muscle shells into pieces (see figure in Bicknell et al. 2018). Similarly, Sidneyia’s gnathobases were only sclerotized and however able to crush trilobite juveniles. The ability of arthropods to crush mineralized organisms does not necessarily require mineralized appendages. Please rewrite here.

Line 173

Not correct here. Ostracodes are tiny thin-shelled crustaceans. Trilobites were relatively large animals and could swallow organisms such as ostracods or tiny epibenthic larvae. They also probably fed on decaying carcasses such as echinoderms and swallowed soft tissues along with disarticulated small exoskeletal elements. You should consider other options. This part needs to be improved.

Line 174-175

No. No “preference” here. The trilobite probably fed on dead echinoderms, swallowed the edible remains along with plates.

Line 192-193

“Durophagy requires special morphological adaptations in the mouth region or appendages that provide the organism with the necessary biting force to overcome the physical limitations of hard tissues”

Yes, please provide adequate refs here. At least, Zacaï et al + Bicknell et al. 2018 in which such things are clearly explained. Not “the mouth region”, instead “gnathobases.”

Line 193-194

“The primary aim is to crush open the shell or exoskeleton in order to reach the soft, digestible tissues”

It is not a question of aim. Scavenging is very frequent even among predators. A considerable amount of food could be obtained without spending energy in crushing. Please improve here.

Line 194

“The ingestion of shell fragments is undesirable and usually in small quantities”

This sentence does not make much sense. It is not a question of “desirable”.

Line 197

“polyphagous with a low degree of food particle preference in their positive frequency-dependent”
This sentence is unclear. Please rewrite.

Line 198

Yes, the trilobite fed on what he could swallow (see diameter of oesophagus) or manage to crush (gnathobases). As many extant arthropods, it was probably chemically attracted to food, especially dead carcasses (see example in Vannier et al. 2000, Marine Biology)

Line 205

OK

Line 208-210

This sentence looks odd (no logics between the two parts). Please delete. Decapods are highly derived arthropods with specialized digestive systems. Better concentrate on *Limulus*.

Line 262

Inorganic shell is not correct. Arthropod biomineralized exoskeletons also contain chitin

Line 263

No deep adaptation. Please delete

Line 262-264

On the other hand, it offers a rich food supply with limited competition
What do you mean here? Unclear and not necessary. Please delete

Line 262-275

This part has nothing to do with “digestion”. Please change sub-title

Line 276

An increased enzymatic support, e.g. a hepatopancreas and other digestive glands²⁰, is required for maximum energy exploitation from a diet rich in indigestible bits associated with organic tissues²¹.
Unclear- There is no relation between enzymatic action and indigestible parts. Enzymes digest soft tissues but have no action on biomineralized exoskeletal parts. That is why they are found in guts.
Please clarify here. Please refer to Zacaï et al. for digestive glands in *Sidneyia*. Delete “increased”.
We don't know whether trilobites produced abundant enzymes or not.

Line 284

morphology of the relatively robust hypostome suggests its possible participation in the biting action.

“Biting action” sounds odd. Perhaps the hypostome served as a hard element on which appendages could press food items ?

Line 316- The peritrophic membrane is not meant to protect the pellets but instead the gut epidermis (it is secreted by the gut to avoid tissue abrasion)

Line 318-

The presence of undissolved sharp-edged calcareous shells in all parts of the gut shows that the digestive tract had an almost pH-neutral inner environment.

Please see remarks above concerning the chemical composition of gut contents

Line 323

The problem is that the digestive tract of limulids (see Zacaï et al.) contains plenty of Ca and P spherules that are stored and recycled during molting. It was most probably the case in *Sidneyia* and that would explain why guts are always preserved in phosphate. Perhaps the same thing occurred in trilobites (see Lerosey-Aubril). I am surprised that you ignored these data. The fact that phosphate precipitated in guts also gives important information on the Ph conditions (see experimental taphonomy in Briggs et al.'s papers). Please change here and do not focus on crustaceans that are unrelated to trilobites.

Line 325

This trilobite mainly gives precise information on its diet not physiology.

unmatched by the usual morphological features included in character matrices for phylogenetic analyses of fossils.

Unclear and not useful. Delete

Line 331-350

There is a huge amount of papers on the physiology of extant crustaceans. Please admit you had no time to look for key refs. Better delete.

Line 354

What do you mean by "bracket". Unclear. Just say that trilobites are Artiopoda (see Aria 2022)

Line 359

Better not mix phylogeny and Ph values. This Ph-range most probably apply to numerous animal groups. It has no phylogenetic significance.

Line 369

I am not convinced by this. The premolting stage (just before edysis) is normally marked by the old cuticle splitting off from the newly secreted one. Nothing similar can be seen here. The vaulting shape is most probably due to animal's attitude when it died and the thoracic "anomaly" to post-mortem dislocation.

Line 371

Not enough evidence to infer it. Muscular contractions seem to be much more important in ecdysis (the process of extricating from the old cuticle). Please improve.

Line 393

This title sounds odd.

Line 398

This 'crumble' could be anything. I see no evidence that it may represent a ligament. Remember that the head of an arthropod contains many different types of tissues.

Line 401-402

Not at all. What sort of scavenging are you talking about ? Relatively large scavengers usually disturb carcasses and pull soft tissues apart. Microscopic scavengers dig into soft tissues and consume them. There is also an enormous microbial activity on and inside carcasses. Please consider this before embarking into speculative interpretations.

Line 405

What kind of potential scavengers ? To me these fibrous mineralized features are microbial filaments. Please find similar occurrence in literature. Add refs. What you see here probably results from microbial activity. Please test these two options

Line 431-432

Better delete this- It is too speculative. Trilobites and derived crustaceans such as decapods can hardly be compared

Line 433-434

It also represents a nutrient target that would not have to decay in order to be fed upon by a weak mouth apparatus (as opposed to, for example, tough muscle tissue, tendons or epidermis).

This sentence does not make much sense. Please rewrite. There is too much speculation here + you give no information on the possible nature of these assumed "scavengers". Why ?

weak mouth apparatus ??

Are you talking about the scavenger's mouth. It is 100% speculative.

Such speculations are unnecessary here. Perhaps just write that some filaments occur inside the glabella and may be due to microbial activity.

Line 436

The digestive tract had a high potential to be consumed,

Not necessarily. The gut contents mainly consist of undigested elements. Edible nutrients have already been assimilated. Based on a single specimen you can hardly conclude that "scavenging" preferentially occurred in a particular area of the head and not elsewhere and that "scavengers" avoided some parts of the dead body. Please delete or reduce this overspeculative part.

Line 438

This avoidance gives clear evidence of inhospitable, probably even dangerous conditions.

I don't believe this. Before you write that the Ph conditions were close to neutral ! Please do not speculate on "scavengers" you know nothing about.

Line 440

I find this quite unrealistic and not correct ! The nodules form over a relatively long time interval.

Enzymes are complex molecules that destroy very rapidly after death and lose their function. The digestive system of all animals houses numerous microbial organisms that play a role in digestion. This clearly shows that this microenvironment is not “dangerous” or “noxious”. Please consider this and improve this part.

Line 442

The behaviour of scavengers and the lack of exit tracks directly support the model suggesting a rapid formation of these nodules
Pure speculation. Please delete.

Line 446

Please add that it confirms other lines of evidence obtained from arthropods close to trilobites (Sidneyia). Refs

Line 445-450

This is not the conclusion that we really expect from a paper on the diet of trilobites. Please refocus your discussion on the important question of the diet of arthropods in the lower Palaeozoic and implications concerning food chains. By doing so, you would give much more strength to your MS

Referee #2 (Remarks to the Author):

Kraft et al present an updated version of an already polished text I had previously reviewed. I have the grand total of 11 incredibly minor comments. This is an exceptional piece of work and deserves publication in Nature. I had also considered reviews by Reviewer 1 and find many of the comments to be unnecessary, in cases could be considered personal and condescending, and have outlined my thoughts on these in after own comments. There are four comments marked with *** that relate to possible changes the authors may want to consider, the rest are more my thoughts on Reviewer 1's comments and can (and should) mostly be ignored as they have no bearing on this review other than to say I think Reviewer 1 was unnecessarily harsh and in places, I think incorrect.

Originality and significance: I maintain this is exceptionally novel and deserves publication.

Data & methodology: All sound and I am sorry I had missed the 3D data in the first review. I did not see the covering letter, but also, I may have missed that letter, so this is on me.

Appropriate use of statistics and treatment of uncertainties: These were limited, but seem appropriate.

Conclusions: Very strong, some minor tweaks needed

Suggested improvements: experiments, data for possible revision: see below.

References: appropriate credit to previous work: more than fine.

Clarity and context: very well written.

[No changes from my previous review].

I look forward to seeing this in press in Nature!

Yours,

Dr Russell Bicknell.

My comments

120: Provisionally called 'crumble' after its appearance, ◇ Provisionally called 'crumble', after its appearance,

127: a citation here is a good idea, as you are generalising across Lichidae

Not sure a new paragraph is needed at 130?

134: accords◇ aligns?

168-170: I think there is something missing here. Decapods in modern systems into trilobites therefore had similar functions. I think one more sentence here would make it clearer

187: "typical of lichid trilobites" also needs a citation. Likely the same as the one at line 127?

199-201: the which, because of the inward midline curvature, distracts flow a bit. Maybe reworking the sentence here?

205: clusters of shell remains ◇ clusters of shells remain?

207: gut was of a large diameter ◇ gut had a large diameter

Paragraph starting at 228: You shift from using the abbreviated genus and species to the genus.

Maybe stick to genus and species

265: what is they referring to here? I prefer to use the noun as, in this case, 'they' could refer to the burrows, the carcass, or the traces

Comments on Reviewer 1 review

I have considered the reviewer 1 comments and would like to highlight some points I have some agreement or disagreement with.

"These findings are however not new": I disagree with this point. As far as I know, this is the first time a trilobite with gut contents that are shelly have been documented.

"The authors also embark into unnecessary discussions on molting, life cycle, Ph conditions that are clearly out of the scope of the main target": I think this is unfair, as the exploration of other topics makes this a really interesting piece of work. I think these additions expand the scope and allow for more discussion.

"Numerous sentences are not understandable at all especially in the discussion part.": I do not agree here. There are some minor changes I suggest this time, but honestly, I find this an enjoyable read.

"I would recommend the authors either to focus on mainly technical aspects (e.g. how synchrotron techniques reveal trilobite gut contents) and submit their MS to a specialized journal, or rewrite their MS on a more solid basis (adequate refs, less speculation) and publish it in a good palaeontology journal.": I strongly disagree here. This is the kind of research that belongs in top-tier journals as it is innovative, it is exciting, and it opens up more avenues to explore virtual arthropod palaeontology.

"please don't use 'crumble'; choose another term": I personally think this is a fine enough word to use in this context, and understood what was intended in the text.

"Size applied to individuals": I have seen size used in taxonomic descriptions. Admittedly it IS a relative concept and should be limited in taxonomic use. But in this context, I think it is fine.

"Limulus has strong non-biomineralized gnathobases with strongly sclerotized teeth that are able to break muscles (shell) into pieces. Same with Sidneyia. The gnathobases of Sidneyia are non-biomineralized elements that were able to break small trilobites into pieces (their remains are found in the gut). Again, this paper and other related one are not cited!!": I am not sure what the reviewer has issue with here. The author state the trilobites have non-biomineralised (maybe sclerotised is better?) gnathobases, which is the same as Limulus and Sidneyia. I am not sure what this comment is trying to achieve.

"Not necessarily inorganic. Durophagy refers to animals that have mouth parts or other

appendages adapted to break hard skeletons (biomineralized or not)": This I do agree with and the authors have made changes to accommodate this point.

*** "Trilobites had no capacity to detect whether an invertebrate was thick or thin shelled": I do have to agree with Reviewer 1 here. I think what is implied is that they could not pick and choose thin from thick shells. Indeed, *Limulus* cannot really either. The limit is whether it can crush the shell, but that is dependant on the individual (I had fed *limulus* bivalves as an experiment once so have some hands-on experience here). I think maybe specifying that trilobites had no way of 'choosing prey' maybe be useful here.

"The example of *Sidneyia* (Zacai et al. 2016) shows that durophagy existed much earlier.

Please mention this important point": I do not think this is relevant. And if it were, then there is the Emu Bay Shale with many shelly coprolites, most likely from trilobites (Bicknell et al., 2022 *Palaeo3*), as the authors point out in the rebut.

"If you had read Zacai et al. 2016 and other related papers (Bicknell, etc..) you would know that they were present.": this reads rather condescendingly to the authors. I am sure they had read around the literature while writing this, and I find this kind of comment rather unfair on the authors.

"There is absolutely no need of specialized appendages. Look at *Limulus* or *Sidneyia*. Food is processed by the gnathobases of a series of homonomous appendages.": this is wrong. *Limulus* has 3 kinds of gnathobasic spine sets for processing prey, pushing legs being the most stout for the strongest crushing. (Bicknell et al, 2021 *Proc B*).

"You forget that the whole trilobite biomineralized exoskeleton is preserved. There is absolutely no reason to suppose that special PH-conditions prevailed within the gut.": I am not sure how the exoskeleton preservation relates to the gut here.

*** "Delete or rearrange the discussion with examples taken from detailed studies on extant arthropod gut contents. Why "dangerous". Avoid these terms": I agree here. Dangerous is a possibly odd choice of word and has a bit of a humanised aspect. Though the authors have modified to "inhospitable" which works very well. That being said "dangerous prey" is a term popularised by Vermeij, so maybe it is appropriate here.

"The authors don't seem to know about the complexity of the moulting process.": this is a very personal statement and was not needed in review.

*** "Everybody knows that arthropods that have just discarded their old carapace are relatively soft and for this reason vulnerable": 'Everyone' is a broad statement from this reviewer. I personally think the inclusion of the comment on moulting was fair. Though yes, aquacultures was a bit left field. I would consider adding this back in, if length permits

*** "You believe in reincarnation? Avoid these terms.": I do not think the reviewer needed to say this in such a way. Though they do raise a point: is there another way of saying this? "Post-mortem events" possibly? It is up to the authors though

Referee #3 (Remarks to the Author):

The authors show the gut contents that contain many exoskeletal fragments, being the first reported in trilobites. The beautiful CT reconstruction has helped to display the gut contents piece by piece. In

addition, many issues, such as ventriculi, premoulting stage, quick digestion, ligament, enzymatic activity, and trace fossils, are involved to discuss the functional process of the gut system. All these ideas working together are quite amazing, however, the connections among these discussions are not smooth but relatively robust.

Exoskeletal fragments preserved in gut systems have been reported in some early organisms such as arthropods and annelids. It is true that the report of fragments in trilobites is a unique case but this single point is not that rare in the early fossil record. The most fascinating part is that the authors have discussed more about the functional aspect of the gut system, however, the evidence present in this paper is not convincing. Some arguments in the paper contradict each other.

Line 49

Fig 3 appears here but Fig 2 is missing in the previous part. Adding Fig 2 to the correct place such as line 45.

Lines 55-56

“though three indistinct clusters can be differentiated in detail (Extended Data Fig. 1b,c).” Should it be Extended Fig. 1a,b?

Lines 70-71

The marginal plates do not have a figure citation, so it is better to cite “Fig. 2b” at the end of this sentence for pinpointing the marginal plates.

Lines 81-90

The authors have suggested “two sack-like ventriculi” and described them with a connection by a short, flat, and bent segment, which is secondarily dislocated to the left. However, I don’t see the “short, flat and bent segment” in the figures and can’t catch this important information. If authors make a line drawing of the outline of the segment and ventriculi in the figures, it would be very helpful.

Authors are stating that the ventral ventriculus (line 90) is connected to the dorsal ventriculus by a short segment, then the dorsal ventriculus is connected to the midgut (line 91-92), and finally the hindgut (line 94). Based on this sequence, the morphology of the gut system is comparable to the modern examples (lines 105-109). This is one of the major arguments in the paper, but I can’t see the evidence with confidence. As the exoskeletal fragments are close to each other, the boundary between clusters, which authors have suggested, is not that clear. That is why I hope the authors can provide a line drawing of ventriculi.

How is the boundary between the ventriculi (authors have argued) and the posterior gut tract different from the boundaries of clusters occurred in the posterior gut tract (Fig. 2a; a boundary in front of the cluster containing the last four blue pieces of fragments that are located at four

corners)? Based on Fig. 2a2, the suggested ventriculi are on the same level and connected transversely. The separation between dorsal and ventral ventriculi is likely the taphonomic artifact because the gut is dislocated.

Fig. 3

The dorsal (glabellar) and ventral (hypostome) ventriculi are suggested. By looking at the fig 3, the ventral ventriculus has a curvature by following the inner concave surface of the hypostome (Fig. 3c), and the dorsal ventriculus has a curvature that follows the inner convex surface of the glabellar (Fig. 3b). I use “concave and convex” for the fig 3 to recognize the ventriculi’ morphology. As ventriculi are restricted by the inner surfaces of the glabella and hypostome, it is hard to make an argument that the dorsal and ventral ventriculi represent their original morphology. Authors need to distinguish why it is not part of taphonomy.

Lines 111-116

“However, the results of the analysis show a statistically insignificant difference (Line 111-112)”. This insignificant difference is interpreted as “Thus, food may have been ingested at speed, but digested at leisure, as documented in other fossil ecdysozoans (lines 115-116)”. This has been used to argue why the function between the suggested two ventriculi is not that efficient. This issue gets back to the comment directly above. If the suggested ventriculi did not exist, the inefficient digestion becomes more reasonable. How do authors distinguish the recognized detail of ventriculi from the taphonomic artifact?

Lines 120-123

“Crumble” is limited to the axial part reflecting the decaying tissue of the gut. Authors have agreed it represents the decaying tissue of the gut, then it is understandable for the question below. The collapse of the decaying tissue would become closer to the trunk segments and likely cast on the inner surface of the trunk segments. Do authors agree on this?

Line 125-129

Digestive caecae are introduced. Three pairs are located below the occipital ring and two thoracic segments. They are shown in the reconstructions (Fig. 5a). However the less extensive diverticulae in the posterior part of the trunk are not reconstructed. Making a distinction between the anterior and posterior diverticulae is likely needed.

In Fig. 4c, the suggested digestive caecae or diverticulae are following the morphology of articulating half rings of the trunk segments. How do authors distinguish the diverticulae from taphonomy because of the collapse of the decaying gut? In 2D fossils, we can recognize the ink-like impressions to recognize the diverticulae. This method is clearly not working in 3D fossils studied herein. If authors can show the diverticulae lifted away from the trunk segment, this evidence would be more convincing. Otherwise, it is likely part of the collapse of the gut tissues but not diverticulae.

Lines 211-212

Undissolved sharp-edged shells suggest an almost pH-neutral inner environment. However, the authors suggested quick digestion in line 115. "The inferred pH of the digestiveis a unique direct evidence of palaeophysiology" in lines 217-218.

I can't catch the theory behind these arguments. Both quick digestion and neutral pH condition can result in slow digestion of the shells, which makes the neutral pH not the unique evidence.

Lines 243-247

Based on the slightly vaulted trunk and partly loosen librigenae, authors suggest it is a premoult phase. This is hard to follow. As the head is distinctly convex and leaves the librigenae in a bedding plane lacking the support of the cranidium, the lateral force performed during the preservation would likely give a shear force existed between the librigenae and cranidium, resulting in the displacement of the librigenae. This isn't evidence for moulting or premoult stage especially when the soft tissues are preserved. Do authors see any part of the exoskeleton that shows double exoskeletal thickness? If it is in a stage that is so close to the moulting, the exoskeleton should be thicker because a new exoskeleton is formed below the old exoskeleton. The vaulting trunk is still the taphonomic artifact or it may represent that animals react to the burial event (I forget the original paper where this has been said, so I just mention this can be the case).

Some well-preserved or enrolled specimens had similar patterns of dislocated librigenae in Whittington, 1971.

Whittington, 1971. Silurian calymenid trilobites from united states, Norway, and Sweden. *Palaeontology*, 14, 455-477.

Line 255

"Minor post-mortem deformation is manifested by the hypostome and gut displacements". Authors have stated the post-mortem deformation. How does the "partly loose librigenae" in line 244 correlate with the moulting? Why is the deformation only affecting the hypostome but not the librigenae? This statement contradicts the premoult theory.

Lines 256-257

A "ligament" is suggested between the hypostome and anterior-ventral side of the cephalon. However, the evidence provided by the authors does not support the "ligament" hypothesis. Check the list of major issues at the end of the comments.

Forty (1990) had described three types of hypostome attachment: natant, conterminant, and impendent. The current one can be natant or conterminant. I don't insist authors use these terms as they are too specialized. Both papers below are helpful for understanding the feeding issues.

Fortey, 1990. Ontogeny, hypostome attachment and trilobite classification. *Palaeontology*, 33, 529-576.

Fortye and Owens 1999. Feeding habits in trilobites. *Palaeontology*, 42, 429-465.

Lines 267-272

Burrows are only restricted to the lobate anterior part of the glabella, in front of the ventriculi. Authors then argue that the burrows are located in front of the digestive system and thus avoid the decaying digestive tract. I understand that authors have used “crumble” to describe the digestive tract, but I find there may be alternative explanations about the relationship between “ligament” and the “crumble”. Please check the possible explanations at the end of the comments.

Line 271

Antennal glands are suggested. We don't have any evidence for the position of antennal glands in any early arthropod. Even if modern arthropods can have the antennal glands there, we can't simply apply this in trilobites. In trilobites, we can use the muscle scars to discuss the position of muscles because muscle scars are observable. There is nothing left of the antennal glands in the fossil record, so it should be cautious about this. If this is purely for the discussion of enzymatic activity, it is overinterpreted by discussing the position and size of the glands.

Lines 283-286

This section suggests evidence of an inhospitable condition, which is then connected with the enzymatic activity. Authors have seen the trace fossils present dominantly in the frontal lobe of the glabella and other trunk pleural regions. There is an alternative explanation about this, please check the list of major issues below.

Many Cambrian arthropods or annelids have gut contents preserved with animal fragments. I list a few here. I don't deny the importance of the work done by authors. If the authors can provide confident evidence for the issues discussed in the paper, it is definitely an innovative thing for early arthropod studies. However, there are some issues that authors need to think about. Please check the list of major issues below.

I here list a few more papers that have discussed the gut contents.

1) Zhu et al. 2004. Direct evidence for predation on trilobites in the Cambrian. *Biology Letters*, 271, S277-S280.

2) Bruton 1981. The arthropod *Sidneyia inexpectans*, middle Cambrian, burgess shale, british Columbia. *Phil Trans R Soc Lond*, 295, 619-653.

3) Vannier 2012. Gut contents as direct indicators for trophic relationships in the Cambrian marine ecosystem. *Plos One*, 7, e52200.

I list a few major issues that authors may need to consider. As there is only one specimen for the study, I think the authors need to provide more convincing evidence.

(I) Dorsal and ventral ventriculi

The connection between the dorsal and ventral ventriculi is an important character but I don't see this evidence clearly in the figures. Authors have mentioned both ventriculi are close to either the dorsal glabella or the ventral hypostome, thus the lobe-like shapes of the ventriculi are affected by the rounded inner surfaces of both glabella and hypostome.

(II) "Ligament" and "crumble"

The authors have used the "ligament" and "crumble" to describe the two separately located structures. There is another possibility that these two structures are continuous and connected with each other in life.

The evidence to challenge the "ligament" hypothesis is shown here. (1) Many small branches or filamentous extensions, like cotton with filaments connecting with each other, are likely not the component of a ligament that should be relatively regular in shape. (2) The coarse appearance and the high thickness are all against the "ligament" hypothesis, especially considering the "ligament" is stretched (authors mentioned it in line 257). (3) The lateral fragment of the "ligament" (Extended Data Fig. 8b), separated from the centrally located "ligament", is likely extended into the lateral border of the left librigena (Extended Data Fig. 8e2). If the ligament is only connecting the hypostome and anterior border of the cephalon, why the ligament extends into the lateral border? (5) "Ligament" perfectly extends into the anterior and lateral borders of the cephalon, indicating a soft tissue with high flexibility. Does a ligament bear such high flexibility? (6) "Ligament" has a perfect shape that matches the inner surface of the frontal lobe of the glabella (Extended Data Fig. 8e2 and 8e3). Again, soft tissue is favored.

The "crumble" is the decaying tissue of the gut. I agree with this. However, the anterior portion of the crumble has many small branches or filamentous extensions, which are extremely similar to those in the posterior portion of the suggested "ligament". I think both "ligament" and "crumble" are connected with each other because they share a similar pattern of filamentous extensions, indicating a complete gut tract.

Authors have mentioned that the hypostome is dislocated from the anterior position to the posterior or the ventral position, the hypostome likely squeezes or cuts off the structure that is located between the hypostome and glabella. As the force is performed by the anterior margin of the hypostome, the gut tract is then cut off rather than two different structures.

In short, the downward displacement of the hypostome has forced and cut the gut tract into two pieces, the anterior "ligament" and the posterior "crumble".

If this is the case, the gut tract will have an expanded frontal crop (expanded anterior portion of the gut tract). By following this possible explanation, the suggested dorsal and ventral ventriculi do not exist as they are inside the wide crop. That is why the connecting segment between the two

ventriculi is not clearly displayed.

(III) Why are traces clustered near the crop or foregut?

As the gut tube is broken, the nutrients released from the crop would supply the surrounding microbes and allow the organisms to feed near the broken region. This is nothing related to the enzymatic activities of the gut tract. Suggesting a favorable way for trace makers is relatively easy compared to suggesting an unfavorable way.

The favorable way involves one hypothesis: nutrients are there. The unfavorable way involves two hypotheses: toxic condition and its producer: enzymes. If we follow the method of phylogenetic studies, one hypothesis is likely better.

Author Rebuttals to First Revision:

Referee #1:

Remark: In my first review I spent time suggesting things that may improve the scientific content of this MS (typically less speculation, open the debate beyond trilobites, focusing on the diet of trilobites and their role in food chains). The authors ignored the majority of my remarks. There are many points throughout the revised version of this MS (except description) that need to be improved and rewritten.

Please see below:

Response: We face an unfortunate and vexing problem when responding to Referee #1 this time: the line numbers given by the referee do not match up with those in our manuscript. We don't know what's gone wrong, but the result is that it is in many places difficult to guess what exactly the remark refers to, especially when only the line number is given without a citation of our text. Wherever we could guess what the comment was referring to, we have tried our best to respond, but in many cases we simply had to pass over the remark.

Remark: TITLE

These well-preserved gut contents tell us about the diet of trilobites not their physiology.

Physiology is the study of the functioning of living organisms, especially their constituent tissues and cells. The present MS is based on fossil evidence and does not illuminate any mechanism at tissue or cell level. "Palaeophysiology" is not correct in this context. Moreover, this term does not exist. Please change the title.

Response: We do not agree. The term "palaeophysiology" is not new, it appears in literature (e.g. Ruben 1989, Cubo & Huttenlocker 2020), though it is not yet commonly used. The idea that aspects of the physiology of a fossil organism can be inferred from preserved attributes has a long pedigree; the most obvious example is the decades-long debate about dinosaur endothermy. We feel it is time that this research field had an established name, and believe that Nature is the perfect platform to highlight the term. Palaeophysiology represents a research direction that attempts to reconstruct life processes of extinct organisms based on indirect evidence. In the same manner as palaeogeography cannot study real existing continents and palaeoecology cannot directly observe natural environments, yet both remain universally accepted as valid scientific disciplines, so palaeophysiology does not require the study of physiological processes in vivo. Its validity will depend on the robustness and falsifiability of the inferences drawn from the data.

Remark: ABSTRACT

"fragmented calcareous shells indicating a voracious feeding activity"

As I wrote in my first review, "voracious" is not the best term. To me, trilobites simply scanned their environment and ingested all edible items they came across, including food from decaying carcasses.

Response: The sentence was modified using terminology common in modern arthropod research.

Remark: Line 56

Better write that this concentration of undigested elements roughly corresponds with

the outline the digestive tract (for general information on morphology see e.g. Lerosey-Aubril's papers)

Response: Unfortunately, we do not know which part of the manuscript this remark refers to. The line number does not match the content of the remark.

Remark: Line 87-88

“consumed at least a half of the digestive tract content” does not really make sense. Please rewrite.

I see no evidence that the trilobite consumed everything in one go. A more realistic view is that trilobites scanned the bottom sediment and randomly fed on everything they came across. Extant ostracods when present are generally very abundant at the water-sediment interval. No wonder that they occur in large number in the trilobite's gut. Trilobites also probably fed of decaying organisms (e.g. echinoderms). The view of trilobites being voracious predators is probably not correct. Please add details and rewrite.

Response: We are not implying that the trilobite consumed everything in one go. We infer that it consumed at least half of the contents of its digestive tract in one place because fragments of a single echinoderm specimen, presumably found in one place, are spread through half of the digestive tract.

Remark: Line 92

Why “most probably” here ? Is there any doubt or alternative option that would be worth being mentioned ? If so, please give details.

Response: We do not know which part of the manuscript this remark refers to. The line number does not match the content of the remark and the piece of text cited doesn't help to specify what the reviewer had in mind.

Remark: Line 94

Are the gut contents homogenous in composition (i.e- elements of various origin randomly distributed and all mixed up) or do you see any local concentrations of echinoderm/ostracods/shell remains. It is an important point that is not addressed here. The way how elements are distributed within the gut may provide key information on the feeding strategy of the animal. Please add information.

Response: The distribution of contents is clearly visible in figures and sufficiently discussed in the text (lines 74-77). We do not know how to expand the discussion without repeating ourselves.

Remark: Line 97

I would rather say that the gut was extensible. For comparison, the gut of *Sidneyia* (an arthropod closely related to trilobites) is often very narrow when not filled with food (see Zacaï et al.). Please improve here.

Response: This remark was used to modify the sentence in line 199.

Remark: Line 101-105

Please add refs and information on how the digestive system of trilobites is usually reconstructed by the specialists of the group. Key refs are missing here. Then write that the distribution of the gut contents is consistent with what we know about the anatomy of the digestive system. Please rewrite this part.

Response: We do not know which part of the manuscript this remark refers to. The line number does not match the content of the remark.

Remark: Line 108

You are inferring fine details of the anatomy of the digestive system from the spatial distribution of undigested fragments. Please discuss the limits of this kind of inference.

Response: We do not know which part of the manuscript this remark refers to. The line number does not match the content of the remark.

Remark: Line 119

Here again, the first thing to do is to give basic information on the appendages of trilobites or related taxa. Refs exists on the appendage morphology and their possible role in feeding (see also *Sidneyia*, an arthropod closely related to trilobites).

Response: We do not know which part of the manuscript this remark refers to. The line number does not match the content of the remark.

Remark: Line 120

Trilobites are crawling not creeping animals. Again refs are available in the literature on locomotion in trilobites.

Response: We do not know which part of the manuscript this remark refers to. The line number does not match the content of the remark. We are surprised by this remark because the words “crawling” and “creeping” do not occur in the text at all.

Remark: Line 126

Zacai et al. provided images of a longitudinal section through *Limulus*. Please refer to this paper here. Comparisons with limulids are much more relevant than those with derived decapods. Another recent paper on *Sidneyia* (Bicknell et al. 2018) also provides key information (and illustration) on the gut contents (muscle shells) of extant *Limulus*.

Response: We do not know which part of the manuscript this remark refers to. The line number does not match the content of the remark.

Remark: Line 131

There is not enough discussing on the possible grinding mechanism by ventral appendages (gnathobases). It is surprising that the authors of this MS do not refer in details to key papers on the feeding behaviour and diet of Cambrian arthropods (Bicknell et al. 2018, Zacai et al., etc..). This problem has already been addressed when I first reviewed the MS. The MS would have a much better impact if enriched with discussions based on previous work. The authors argue that *Sidneyia* belongs to a separate group. They are perfectly right. However, trilobites and *Sidneyia* both belong to Artiopoda (see Aria's most recent phylogenies in 2022) and share important features (digestive system, appendage morphology, etc..).

Response: The details of the grinding mechanism are not the subject of this study. In addition, more discussing would lead to speculations, considering that the gnathobases are not preserved. It is sufficiently discussed and referred in lines 189-196.

Remark: Line 132-133

Concerning the diet (direct evidence from gut contents) of Cambrian priapulid worms please add Vannier 2012. “Ecdysozoan” is too vague.

“Thus, food may have been ingested at speed, but digested at leisure, as

documented in other fossil ecdysozoans¹¹”

You have no strong evidence to say that (i.e. how fast trilobites processed and ingested food, how fast food was broken down by enzymes). Please delete. It would be more realistic to write that the animal probably kept on ingesting food until its stomach became full without any consideration of speed. Please delete sentences that suggest a kind of “intentional” behaviour.

Response: The sentence was deleted.

Remark: Line 134-138

These sentences are very unclear and not correct (language problems). Please rewrite. It is clear that shapeless aggregates frequently occur in rocks throughout the Ordovician of Bohemia. As far as I know there is no Burgess Shale type Lagerstätte in Bohemia. Why do you think that some of the aggregates represent in situ decaying organisms ? Most of these skeletal elements were most likely accumulated by currents. Please rewrite these sentences accordingly.

Response: We do not know which part of the manuscript this remark refers to. The line number does not match the content of the remark and the remark does not specify what the reviewer had in mind.

Remark: Line 139 +

“In the thorax of our specimen, the ‘crumble’ is limited to the axial part reflecting the decaying tissue of the gut”

This trilobite specimens clearly show undigested elements throughout the gut but no remain of any soft tissue. Please clarify this important point. “Crumble” sounds odd here. Does the “crumble” represents mineralized tissues ? If so, what kind of mineralization (please indicate composition).

Response: Referee no. 2, a native English speaker, doesn't have a problem with the term “crumble”. It represents a specific fill of cavities after the decayed tissue (see Kraft et al. 2020). We expanded the chapter Taphonomy to clarify the specifics of “crumble” preservation.

Remark: IMPORTANT: another thing concerns the chemical composition of these undigested elements. Please provide elemental maps or at least an EDS-analysis of some of the elements. The digestive system of numerous Cambrian arthropods is preserved in phosphate (glands, tract) including gut contents when present (see Zacaï et al, Zhu and Vannier on phosphatized gut contents from Chengjiang, Biology Letters; papers by Lerosey-Aubril). It is absolutely essential to know about the chemical composition of the trilobite gut contents. This has implications on your interpretations.

Digestive glands in trilobites or related taxa do not occur throughout the digestive tract. Instead they are concentrated anteriorly. I doubt that your so-called “crumble” (light green in model) represents the remains of soft tissues such as digestive glands.

Response: As mentioned several times in the manuscript, all hard parts of the specimen, as well as undigested gut contents, are preserved as negatives, that is, empty spaces that perfectly represent the original shape. Their chemical composition is inferred from the phylogenetic affinities of the organisms they represent. The most important in this respect are the echinoderm plates, identifiable from their characteristic stereom mesh; echinoderm skeletons are always formed from calcite

and their survival in an uncorroded state demonstrates that the gut pH was not significantly acidic.

We thought about the problem of digestive glands from several points of view. We considered that these structures may represent taphonomical artefacts due to the gut compression. However, this mechanism would only transversally wrinkle the dorsal surface of the gut. Any other deformation would be reflected in the gut diameter and form artificial segmentation. In addition, the deformation would be rather heterogenous and not uniform along the gut as it is in our specimen. A homogenous deformation is very hard to imagine with respect to gut content and tissues which surrounded it.

Remark: Line 163

“no parallel either in the coeval fossil communities”.

Not really true. It is surprising that you keep on ignoring or minimizing previous work on the gut contents of Cambrian arthropods. You would make your MS much more interesting if you went beyond trilobites. Sorry for repeating it again and again, Zacaï et al. described well-preserved gut contents in *Sidneyia* (juvenile trilobites among other elements). These elements are as well preserved as the gut contents presented here. Please improve this part simply to be scientifically honest.

Response: The trilobite is Ordovician, therefore Cambrian arthropods are not coeval. For the rest of the remark, see comments by Referee no. 2.

Remark: Line 164

These elements were originally calcareous but nothing proves that are mineralized in calcite. The exoskeletal remains of trilobites inside the gut of *Sidneyia* are phosphatic although they were originally calcified. It is an important point to consider.

Response: We do not know which part of the manuscript this remark refers to. The line number is wrong. For “chemical composition” see above.

Remark: Line 168

“based on the absence of an assemblage of fragments in the intestine”

It is not very logical here. If the animal is unable to crush hard shell, then how do you explain the occurrence of shell fragments in its gut. Please change.

Response: The mechanism is described in detail in the chapter Feeding strategy.

Remark: Line 169-170

Very unclear. Please rewrite (language problems)

Response: We do not know which part of the manuscript this remark refers to. The line number does not match the content of the remark. Referee no. 2 is a native English speaker and he did not encounter any serious language problems in our manuscript.

Remark: Line 170

The gnathobases of *Limulus* are not biomineralized, only sclerotized. However, they are able to crush muscle shells into pieces (see figure in Bicknell et al. 2018).

Similarly, *Sidneyia*'s gnathobases were only sclerotized and however able to crush trilobite juveniles. The ability of arthropods to crush mineralized organisms does not necessarily require mineralized appendages. Please rewrite here.

Response: We do not know exactly which part of the manuscript this remark refers to. The line number does not match the content of the remark. However, we infer

that this remark refers to the same point as the remark by Referee no. 2. If it is so, then we have changed the word “non-biomineralized” to “sclerotised”.

Remark: Line 173

Not correct here. Ostracodes are tiny thin-shelled crustaceans. Trilobites were relatively large animals and could swallow organisms such as ostracods or tiny epibenthic larvae. They also probably fed on decaying carcasses such as echinoderms and swallowed soft tissues along with disarticulated small exoskeletal elements. You should consider other options. This part needs to be improved.

Response: We do not know which part of the manuscript this remark refers to. The line number does not match the content of the remark.

Remark: Line 174-175

No. No “preference” here. The trilobite probably fed on dead echinoderms, swallowed the edible remains along with plates.

Response: We do not know which part of the manuscript this remark refers to. The line number does not match the content of the remark.

Remark: Line 192-193

“Durophagy requires special morphological adaptations in the mouth region or appendages that provide the organism with the necessary biting force to overcome the physical limitations of hard tissues”

Yes, please provide adequate refs here. At least, Zacaï et al + Bicknell et al. 2018 in which such things are clearly explained. Not “the mouth region”, instead “gnathobases.”

Response: We disagree, “mouth region” must be used in this sentence, because it refers to the morphology of a general durophagous animal, not just animals with gnathobases. One of requested references was added.

Remark: Line 193-194

“The primary aim is to crush open the shell or exoskeleton in order to reach the soft, digestible tissues”

It is not a question of aim. Scavenging is very frequent even among predators. A considerable amount of food could be obtained without spending energy in crushing. Please improve here.

Response: We do not understand what the referee requires from us here. How does the argument relate to the characteristics of durophagy?

Remark: Line 194

“The ingestion of shell fragments is undesirable and usually in small quantities”
This sentence does not make much sense. It is not a question of “desirable”.

Response: Well, most durophagous animals avoid consuming the shells if possible. We don't know how else to phrase it.

Remark: Line 197

“polyphagous with a low degree of food particle preference in their positive frequency-dependent”

This sentence is unclear. Please rewrite.

Response: It is a direct transcription from the cited reference.

Remark: Line 198

Yes, the trilobite fed on what he could swallow (see diameter of oesophagus) or manage to crush (gnathobases). As many extant arthropods, it was probably chemically attracted to food, especially dead carcasses (see example in Vannier et al. 2000, Marine Biology)

Response: We do not know which part of the manuscript this remark refers to. The line number does not match the content of the remark.

Remark: Line 205

OK

Response: We don't know what this remark refers to.

Remark: Line 208-210

This sentence looks odd (no logics between the two parts). Please delete. Decapods are highly derived arthropods with specialized digestive systems. Better concentrate on Limulus.

Response: We do not know which part of the manuscript this remark refers to. The line number does not match the content of the remark.

Remark: Line 262

Inorganic shell is not correct. Arthropod biomineralized exoskeletons also contain chitin

Response: We do not know which part of the manuscript this remark refers to. The line number does not match the content of the remark.

Remark: Line 263

No deep adaptation. Please delete

Response: We do not know which part of the manuscript this remark refers to. The line number does not match the content of the remark.

Remark: Line 262-264

On the other hand, it offers a rich food supply with limited competition

What do you mean here? Unclear and not necessary. Please delete

Response: We disagree, the sentence is essential. A feeding mode that requires special adaptations has many disadvantages, but on the other hand, as not many other animals share these adaptations, the competition in such an ecological niche is usually low. If an animal is able to gather nutrients from a diet that is abundant but avoided by most, the advantages outweigh the disadvantages. The sentence was slightly modified.

Remark: Line 262-275

This part has nothing to do with "digestion". Please change sub-title

Response: We do not know which part of the manuscript this remark refers to. The line number does not match the content of the remark. However, we changed most of the chapter titles.

Remark: Line 276

An increased enzymatic support, e.g. a hepatopancreas and other digestive glands²⁰, is required for maximum energy exploitation from a diet rich in indigestible bits associated with organic tissues²¹.

Unclear- There is no relation between enzymatic action and indigestible parts. Enzyme digest soft tissues but have no action on biomineralized exoskeletal parts. That is why they are found in guts. Please clarify here. Please refer to Zacaï et al. for digestive glands in Sidneyia. Delete "increased". We don't know whether trilobites produced abundant enzymes or not.

Response: The sentence was modified.

Remark: Line 284

morphology of the relatively robust hypostome suggests its possible participation in the biting action.

"Biting action" sounds odd. Perhaps the hypostome served as a hard element on which appendages could press food items ?

Response: The sentence was modified.

Remark: Line 316- The peritrophic membrane is not meant to protect the pellets but instead the gut epidermis (it is secreted by the gut to avoid tissue abrasion)

Response: Yes, the sentence was misleading and was modified.

Remark: Line 318-

The presence of undissolved sharp-edged calcareous shells in all parts of the gut shows that the digestive tract had an almost pH-neutral inner environment.

Please see remarks above concerning the chemical composition of gut contents

Response: See above.

Remark: Line 323

The problem is that the digestive tract of limulids (see Zacaï et al.) contains plenty of Ca and P spherules that are stored and recycled during molting. It was most probably the case in Sidneyia and that would explain why guts are always preserved in phosphate. Perhaps the same thing occurred in trilobites (see Lerosey-Aubril). I am surprised that you ignored these data. The fact that phosphate precipitated in guts also gives important information on the Ph conditions (see experimental taphonomy in Briggs et al.'s papers). Please change here and do not focus on crustaceans that are unrelated to trilobites.

Response: We already answered this one in the previous round of reviews. We can describe and discuss only what we observe or what we are able to infer from the evidence. As the nodule is siliceous, the 'crumble' (based on unpublished analyses, in prep.) is siliceous and shells are dissolved we cannot comment on the role of phosphate and the taphonomical aspects of different modes of preservation.

Remark: Line 325

This trilobite mainly gives precise information on its diet not physiology.

Response: We do not know which part of the manuscript this remark refers to. The line number does not match the content of the remark.

Remark: unmatched by the usual morphological features included in character matrices for phylogenetic analyses of fossils.

Unclear and not useful. Delete

Response: We disagree. Having at least indirect evidence of life processes of extinct organisms is a valuable data source for comparative analyses not only with other fossil organisms but also with extant forms. The sentence was rewritten.

Remark: Line 331-350

There is a huge amount of papers on the physiology of extant crustaceans. Please admit you had no time to look for key refs. Better delete.

Response: We do not know which part of the manuscript this remark refers to. The line number is wrong.

We spent a considerable amount of time and effort studying all the key and also many related publications, and carefully deciding which of them were relevant to our argumentation.

Remark: Line 354

What do you mean by “bracket”. Unclear. Just say that trilobites are Artiopoda (see Aria 2022)

Response: It is a standard term used in phylogeny.

Remark: Line 359

Better not mix phylogeny and Ph values. This Ph-range most probably apply to numerous animal groups. It has no phylogenetic significance.

Response: Line number 359 points to the reference list of our manuscript. We do not know which part of the manuscript the referee had in mind. In any case there is no reason to assume that gut pH lacks phylogenetic significance; it is an important physiological characteristic and thus likely to exhibit considerable evolutionary stability.

Remark: Line 369

I am not convinced by this. The premolting stage (just before edysis) is normally marked by the old cuticle splitting off from the newly secreted one. Nothing similar can be seen here. The vaulting shape is most probably due to animal’s attitude when it died and the thoracic “anomaly” to post-mortem dislocation.

Response: Line number 369 points to the reference list of our manuscript. We do not know which part of the manuscript the referee had in mind. Anyway, we responded on the presence of cuticle in fossil material in the first round of reviews.

Remark: Line 371

Not enough evidence to infer it. Muscular contractions seem to be much more important in ecdysis (the process of extricating from the old cuticle). Please improve.

Response: Line number 371 points to the reference list of our manuscript. We do not know which part of the manuscript the referee had in mind.

Remark: Line 393

This title sounds odd.

Response: Line number 393 points to the reference list of our manuscript. We do not know which part of the manuscript the referee had in mind.

Remark: Line 398

This ‘crumble’ could be anything. I see no evidence that it may represent a ligament. Remember that the head of an arthropod contains many different types of tissues.

Response: Line number 398 points to the reference list of our manuscript. We do not know which part of the manuscript the referee had in mind.

Remark: Line 401-402

Not at all. What sort of scavenging are you talking about ? Relatively large scavengers usually disturb carcasses and pull soft tissues apart. Microscopic scavengers dig into soft tissues and consume them. There is also an enormous microbial activity on and inside carcasses. Please consider this before embarking into speculative interpretations.

Response: Line numbers 401-402 point to the figures in our manuscript. We do not know which part of the manuscript the referee had in mind. However, this is a typical case of misunderstanding ichnology, the biology of infauna combined with sedimentological aspects. It is significant that neither of the other referees objected against this trivial ichnological interpretation (not speculation!).

Remark: Line 405

What kind of potential scavengers ? To me these fibrous mineralized features are microbial filaments. Please find similar occurrence in literature. Add refs. What you see here probably results from microbial activity. Please test these two options

Response: Untrue, what we see here are ichnofossils. See publications dealing with ichnofossils in the nodules of the Šárka Formation, as we stated in our previous reply (!), such as Mikuláš 2003 and Kraft et al. 2020. The other two referees did not object to our interpretation.

Remark: Line 431-432

Better delete this- It is too speculative. Trilobites and derived crustaceans such as decapods can hardly be compared

Response: Line numbers 431-432 point to the figures in our manuscript. We do not know which part of the manuscript the referee had in mind.

Remark: Line 433-434

It also represents a nutrient target that would not have to decay in order to be fed upon by a weak mouth apparatus (as opposed to, for example, tough muscle tissue, tendons or epidermis).

This sentence does not make much sense. Please rewrite. There is too much speculation here + you give no information on the possible nature of these assumed "scavengers". Why ?

Response: Please see Kraft et al. 2020 and everything will become clear.

Remark: weak mouth apparatus ??

Are you talking about the scavenger's mouth. It is 100% speculative.

Such speculations are unnecessary here. Perhaps just write that some filaments occur inside the glabella and may be due to microbial activity.

Response: See above.

Remark: Line 436

The digestive tract had a high potential to be consumed,
Not necessarily. The gut contents mainly consist of undigested elements. Edible nutrients have already been assimilated. Based on a single specimen you can hardly conclude that "scavenging" preferentially occurred in a particular area of the head and not elsewhere and that "scavengers" avoided some parts of the dead body. Please delete or reduce this overspeculative part.

Response: We are talking about the entire digestive tract, i.e. including the gut tissue, which is usually one of the first targets by scavengers (see Kraft et al. 2020). Also, we are not extrapolating the general behaviour of scavengers, we are only discussing what we see in this single specimen and what it may imply.

Remark: Line 438

This avoidance gives clear evidence of inhospitable, probably even dangerous conditions.

I don't believe this. Before you write that the Ph conditions were close to neutral ! Please do not speculate on "scavengers" you know nothing about.

Response: This is not a question of belief. In addition, the referee is confusing arguments here. The pH condition controls enzymatic activity. The enzymes and their activity are dangerous to any creature that is affected by them. In addition, there can be other (speculative in this case) processes that make this environment inhospitable. What is speculative about the presence of scavengers based on trace fossils, which are common in the Šárka Formation nodules and well described in the literature? The other two referees did not object.

Remark: Line 440

I find this quite unrealistic and not correct ! The nodules form over a relatively long time interval. Enzymes are complex molecules that destroy very rapidly after death and lose their function. The digestive system of all animals houses numerous microbial organisms that play a role in digestion. This clearly shows that this microenvironment is not "dangerous" or "noxious". Please consider this and improve this part.

Response: Line number 440 points to the figures in our manuscript. Anyway, this remark is unfair. The referee keeps repeating that we have no data for our interpretations, but on what grounds do they think that the nodules formed during a long interval? We provide clear evidence that it was the opposite.

Remark: Line 442

The behaviour of scavengers and the lack of exit tracks directly support the model suggesting a rapid formation of these nodules
Pure speculation. Please delete.

Response: This remark is based on the referee's feelings and beliefs that the tracks are not ichnofossils. We have scientific evidence that they are.

Remark: Line 446

Please add that it confirms other lines of evidence obtained from arthropods close to trilobites (Sidneyia). Refs

Response: Line number 446 points to the figures in our manuscript. We do not know which part of the manuscript the referee had in mind.

Remark: Line 445-450

This is not the conclusion that we really expect from a paper on the diet of trilobites. Please refocus your discussion on the important question of the diet of arthropods in the lower Palaeozoic and implications concerning food chains. By doing so, you would give much more strength to your MS

Response: Line numbers 445-450 point to the figures in our manuscript. We do not know which part of the manuscript the referee had in mind. However, the conclusion

chapter was modified.

Referee #2

Kraft et al present an updated version of an already polished text I had previously reviewed. I have the grand total of 11 incredibly minor comments. This is an exceptional piece of work and deserves publication in Nature. I had also considered reviews by Reviewer 1 and find many of the comments to be unnecessary, in cases could be considered personal and condescending, and have outlined my thoughts on these in after own comments. There are four comments marked with *** that relate to possible changes the authors may want to consider, the rest are more my thoughts on Reviewer 1's comments and can (and should) mostly be ignored as they have no bearing on this review other than to say I think Reviewer 1 was unnecessarily harsh and in places, I think incorrect.

Originality and significance: I maintain this is exceptionally novel and deserves publication.

Data & methodology: All sound and I am sorry I had missed the 3D data in the first review. I did not see the covering letter, but also, I may have missed that letter, so this is on me.

Appropriate use of statistics and treatment of uncertainties: These were limited, but seem appropriate.

Conclusions: Very strong, some minor tweaks needed

Suggested improvements: experiments, data for possible revision: see below.

References: appropriate credit to previous work: more than fine.

Clarity and context: very well written.

[No changes from my previous review].

I look forward to seeing this in press in Nature!

Yours,

Dr Russell Bicknell.

My comments

Remark: 120: Provisionally called 'crumble' after its appearance, ◇ Provisionally called 'crumble', after its appearance,

Response: Done.

Remark: 127: a citation here is a good idea, as you are generalising across Lichidae
Not sure a new paragraph is needed at 130?

Response: A reference was added.

Remark: 134: accords◇ aligns?

Response: Done. The word "accords" is in line number 143, so we hope we got it right.

Remark: 168-170: I think there is something missing here. Decapods in modern systems into trilobites therefore had similar functions. I think one more sentence here would make it clearer

Response: The paragraph was deleted.

Remark: 187: “typical of lichid trilobites” also needs a citation. Likely the same as the one at line 127?

Response: Reference was added and the sentence was slightly modified.

Remark: 199-201: the which, because of the inward midline curvature, distrupts flow a bit. Maybe reworking the sentence here?

Response: The sentence was modified.

Remark: 205: clusters of shell remains ◇ clusters of shells remain?

Response: The sentence was modified.

Remark: 207: gut was of a large diameter ◇ gut had a large diameter

Response: Done.

Remark: Paragraph starting at 228: You shift from using the abbreviated genus and species to the genus. Maybe stick to genus and species

Response: As the discussion concerns not only a single species (“... and related taxa”) we feel our usage is justified.

Remark: 265: what is they referring to here? I prefer to use the noun as, in this case, ‘they’ could refer to the burrows, the carcass, or the traces

Response: The sentence was modified.

Remark: Comments on Reviewer 1 review

I have considered the reviewer 1 comments and would like to highlight some points I have some agreement or disagreement with.

“These findings are however not new”: I disagree with this point. As far as I know, this is the first time a trilobite with gut contents that are shelly have been documented.

“The authors also embark into unnecessary discussions on molting, life cycle, Ph conditions

that are clearly out of the scope of the main target”: I think this is unfair, as the exploration of other topics makes this a really interesting piece of work. I think these additions expand the scope and allow for more discussion.

“Numerous sentences are not understandable at all especially in the discussion part.”: I do not agree here. There are some minor changes I suggest this time, but honestly, I find this an enjoyable read.

“I would recommend the authors either to focus on mainly technical aspects (e.g. how synchrotron techniques reveal trilobite gut contents) and submit their MS to a specialized

journal, or rewrite their MS on a more solid basis (adequate refs, less speculation) and publish

it in a good palaeontology journal.”: I strongly disagree here. This is the kind of research that belongs in top-tier journals as it is innovative, it is exciting, and it opens up more avenues to explore virtual arthropod palaeontology.

“please don’t use “crumble”; choose another term”: I personally think this is a fine enough word to use in this context, and understood what was intended in the text.

“Size applied to individuals”: I have seen size used in taxonomic descriptions.

Admittedly it IS a relative concept and should be limited in taxonomic use. But in this context, I think it is fine.

“Limulus has strong non-biomineralized gnathobases with strongly sclerotized teeth that are able to break muscles (shell) into pieces. Same with Sidneyia. The gnathobases of Sidneyia are non-biomineralized elements that were able to break small trilobites into pieces (their remains are found in the gut). Again, this paper and other related one are not cited!!”: I am not sure what the reviewer has issue with here. The author state the trilobites have non-biomineralised (maybe sclerotised is better?) gnathobases, which is the same as Limulus and Sidneyia. I am not sure what this comment is trying to achieve.

Response: Thank you for addressing some points of Referee no. 1 review, which we also think are unfair. We changed the word “non-biomineralised” to “sclerotised”.

“Not necessarily inorganic. Durophagy refers to animals that have mouth parts or other appendages adapted to break hard skeletons (biomineralized or not)”: This I do agree with and the authors have made changes to accommodate this point.

Remark:*** “Trilobites had no capacity to detect whether an invertebrate was thick or thin shelled”: I do have to agree with Reviewer 1 here. I think what is implied is that they could not pick and choose thin from thick shells. Indeed, Limulus cannot really either. The limit is whether it can crush the shell, but that is dependant on the individual (I had fed limulus bivalves as an experiment once so have some hands-on experience here). I think maybe specifying that trilobites had no way of ‘choosing prey’ maybe be useful here.

Response: We do not write about the capacity of the trilobite to detect the character of its prey but about the ability of the trilobite to consume certain particles. Fair enough, we slightly modified the sentence.

“The example of Sidneyia (Zacai et al. 2016) shows that durophagy existed much earlier.

Please mention this important point”: I do not think this is relevant. And if it were, then there is the Emu Bay Shale with many shelly coprolites, most likely from trilobites (Bicknell et al., 2022 Palaeo3), as the authors point out in the rebut.

“If you had read Zacai et al. 2016 and other related papers (Bicknell, etc..) you would know

that they were present.”: this reads rather condescendingly to the authors. I am sure they had read around the literature while writing this, and I find this kind of comment rather unfair on the authors.

“There is absolutely no need of specialized appendages. Look at Limulus or Sidneyia. Food is processed by the gnathobases of a series of homonomous appendages.”: this is wrong. Limulus has 3 kinds of gnathobasic spine sets for processing prey, pushing legs being the most stout for the strongest crushing. (Bicknell et al, 2021 Proc B).

“You forget that the whole trilobite biomineralized exoskeleton is preserved. There is absolutely no reason to suppose that special PH-conditions prevailed

within the gut.”: I am not sure how the exoskeleton preservation relates to the gut

here.

Remark: *** “Delete or rearrange the discussion with examples taken from detailed studies

on extant arthropod gut contents. Why “dangerous”. Avoid these terms”: I agree here. Dangerous is a possibly odd choice of word and has a bit of a humanised aspect. Though the authors have modified to “inhospitable” which works very well. That being said “dangerous prey” is a term popularised by Vermeij, so maybe it is appropriate here.

Response: The sentence was modified.

“The authors don’t seem to know about the complexity of the moulting process.”: this is a very personal statement and was not needed in review.

Remark: *** “Everybody knows that arthropods that have just discarded their old carapace are relatively soft and for this reason vulnerable”: ‘Everyone’ is a broad statement from this reviewer. I personally think the inclusion of the comment on moulting was fair. Though yes, aquacultures was a bit left field. I would consider adding this back in, if length permits

Response: Adding this comment back in would disrupt the flow of the text. So, we decided to keep it as it is.

Remark: *** “You believe in reincarnation? Avoid these terms.”: I do not think the reviewer needed to say this in such a way. Though they do raise a point: is there another way of saying this? “Post-mortem events” possibly? It is up to the authors though

Response: We changed the chapter title as requested.

Referee #3

The authors show the gut contents that contain many exoskeletal fragments, being the first reported in trilobites. The beautiful CT reconstruction has helped to display the gut contents piece by piece. In addition, many issues, such as ventriculi, premoulting stage, quick digestion, ligament, enzymatic activity, and trace fossils, are involved to discuss the functional process of the gut system. All these ideas working together are quite amazing, however, the connections among these discussions are not smooth but relatively robust.

Exoskeletal fragments preserved in gut systems have been reported in some early organisms such as arthropods and annelids. It is true that the report of fragments in trilobites is a unique case but this single point is not that rare in the early fossil record. The most fascinating part is that the authors have discussed more about the functional aspect of the gut system, however, the evidence present in this paper is not convincing. Some arguments in the paper contradict each other.

Remark: Line 49

Fig 3 appears here but Fig 2 is missing in the previous part. Adding Fig 2 to the correct place such as line 45.

Response: A reference to Fig. 2 was added.

Remark: Lines 55-56

“though three indistinct clusters can be differentiated in detail (Extended Data Fig. 1b,c).” Should it be Extended Fig. 1a,b?

Response: No, it is correct. The gut fill is visible in ED Fig. 1b and 1c.

Remark: Lines 70-71

The marginal plates do not have a figure citation, so it is better to cite “Fig. 2b” at the end of this sentence for pinpointing the marginal plates.

Response: Reference to figures was added.

Remark: Lines 81-90

The authors have suggested “two sack-like ventriculi” and described them with a connection by a short, flat, and bent segment, which is secondarily dislocated to the left. However, I don’t see the “short, flat and bent segment” in the figures and can’t catch this important information. If authors make a line drawing of the outline of the segment and ventriculi in the figures, it would be very helpful.

Response: A line drawing would be problematic because the shapes of all parts of the digestive tract are inferred from the accumulations of discrete fragments inside them. Therefore, the drawing would only be an interpretation of the original shapes and would not provide more information than the rendered scan images. Perhaps a dotted 3D drawing would partly serve the purpose, but a drawing from the same or similar perspective, which seems to be the most instructive, would provide equal information as the reconstruction drawing in Fig. 5.

Remark: Authors are stating that the ventral ventriculus (line 90) is connected to the dorsal ventriculus by a short segment, then the dorsal ventriculus is connected to the midgut (line 91-92), and finally the hindgut (line 94). Based on this sequence, the morphology of the gut system is comparable to the modern examples (lines 105-109). This is one of the major arguments in the paper, but I can’t see the evidence with confidence. As the exoskeletal fragments are close to each other, the boundary between clusters, which authors have suggested, is not that clear. That is why I hope the authors can provide a line drawing of ventriculi.

Response: We understand that it can be difficult to see in 2D images, even though we did our best to select views in different angles. The referees can download a 3D pdf from <https://uppsala.box.com/s/n71426jvkbmg7ljaxh47nnl252e1ob2l> which should make the morphology easier to understand. We have also modified the text for better clarity.

Remark: How is the boundary between the ventriculi (authors have argued) and the posterior gut tract different from the boundaries of clusters occurred in the posterior gut tract (Fig. 2a; a boundary in front of the cluster containing the last four blue pieces of fragments that are located at four corners)? Based on Fig. 2a2, the suggested ventriculi are on the same level and connected transversely. The separation between dorsal and ventral ventriculi is likely the taphonomic artifact because the gut is dislocated.

Response: The 3D pdf will help answering this question because the ventriculi are not on the same level. The gut is clearly a continuation of the dorsal ventriculus. We are not arguing that the boundary between the foregut and midgut differs from the

boundaries in the hindgut. On the contrary, all of them are very faint, best spotted in motion.

The separation between the dorsal and ventral ventriculi cannot be a taphonomical effect. If those were originally a single structure, the post-mortem compression would lead to a homogenous mass pressed into the ventral surface of the glabella. Having two discrete accumulations of fragments in the head region is a direct indication of two structures separated by soft tissues in life.

Remark: Fig. 3

The dorsal (glabellar) and ventral (hypostome) ventriculi are suggested. By looking at the fig 3, the ventral ventriculus has a curvature by following the inner concave surface of the hypostome (Fig. 3c), and the dorsal ventriculus has a curvature that follows the inner convex surface of the glabellar (Fig. 3b). I use “concave and convex” for the fig 3 to recognize the ventriculi’ morphology. As ventriculi are restricted by the inner surfaces of the glabella and hypostome, it is hard to make an argument that the dorsal and ventral ventriculi represent their original morphology. Authors need to distinguish why it is not part of taphonomy.

Response: See above, the text was modified.

Remark: Lines 111-116

“However, the results of the analysis show a statistically insignificant difference (Line 111-112)”. This insignificant difference is interpreted as “Thus, food may have been ingested at speed, but digested at leisure, as documented in other fossil ecdysozoans (lines 115-116)”. This has been used to argue why the function between the suggested two ventriculi is not that efficient. This issue gets back to the comment directly above. If the suggested ventriculi did not exist, the inefficient digestion becomes more reasonable. How do authors distinguish the recognized detail of ventriculi from the taphonomic artifact?

Response: See above. There are two discrete clusters deformed by the hypostome dislocation but still existing.

Remark: Lines 120-123

“Crumble” is limited to the axial part reflecting the decaying tissue of the gut. Authors have agreed it represents the decaying tissue of the gut, then it is understandable for the question below. The collapse of the decaying tissue would become closer to the trunk segments and likely cast on the inner surface of the trunk segments. Do authors agree on this?

Response: Please note that we describe a unique process which happened in a nodule that formed very quickly after the death of the trilobite. The nature of this process is unlike the usual decay processes we are used to in common facies, such as when the fossil is covered by a muddy substrate which will later form a shale. Two consecutive taphonomic processes took place: first a compression of the dead organism in a still soft substrate immediately after burial, and then the decay of tissues in an already forming nodule, i.e. in a solid matter produced by silicification of the muddy substrate at a very shallow depth below the sea bottom. There is not enough space in the manuscript to fully describe the process, but bear in mind that there is a substantial difference between the preservation in the nodules and the surrounding shales. After the formation of the nodule, there were no further deformations or collapse of the cavities left after the decayed tissues. That is why the cavities are preserved and filled by the “crumble”, i.e. fossilized products of the

decayed tissues due to bacterial activity. The tissue decay was happening in cavities entombed in solid matter. That is why their shape didn't change further and it reflects the original shape of the cavities at the time when the nodule was formed, which was apparently very early after the specimen was buried (which is also supported by the behaviour of the scavengers, see chapter Post-mortem events). This bit also explains some of your questions below.

Remark: Line 125-129

Digestive caecae are introduced. Three pairs are located below the occipital ring and two thoracic segments. They are shown in the reconstructions (Fig. 5a). However the less extensive diverticulae in the posterior part of the trunk are not reconstructed. Making a distinction between the anterior and posterior diverticulae is likely needed.

Response: They are reconstructed. See the undulated outline of the gut in Fig. 5.

Remark: In Fig. 4c, the suggested digestive caecae or diverticulae are following the morphology of articulating half rings of the trunk segments. How do authors distinguish the diverticulae from taphonomy because of the collapse of the decaying gut? In 2D fossils, we can recognize the ink-like impressions to recognize the diverticulae. This method is clearly not working in 3D fossils studied herein. If authors can show the diverticulae lifted away from the trunk segment, this evidence would be more convincing. Otherwise, it is likely part of the collapse of the gut tissues but not diverticulae.

Response: As explained above, the decaying tissue had nowhere to expand. Everything was enclosed within solid, silicified sediment.

Remark: Lines 211-212

Undissolved sharp-edged shells suggest an almost pH-neutral inner environment. However, the authors suggested quick digestion in line 115. "The inferred pH of the digestive ...is a unique direct evidence of palaeophysiology" in lines 217-218.

I can't catch the theory behind these arguments. Both quick digestion and neutral pH condition can result in slow digestion of the shells, which makes the neutral pH not the unique evidence.

Response: You are right. The argumentation has weaknesses here, so the sentence discussing the digestion rate was deleted.

Remark: Lines 243-247

Based on the slightly vaulted trunk and partly loosened librigenae, authors suggest it is a premoult phase. This is hard to follow. As the head is distinctly convex and leaves the librigenae in a bedding plane lacking the support of the cranium, the lateral force performed during the preservation would likely give a shear force existed between the librigenae and cranium, resulting in the displacement of the librigenae. This isn't evidence for moulting or premoult stage especially when the soft tissues are preserved. Do authors see any part of the exoskeleton that shows double exoskeletal thickness? If it is in a stage that is so close to the moulting, the exoskeleton should be thicker because a new exoskeleton is formed below the old exoskeleton. The vaulting trunk is still the taphonomic artifact or it may represent that animals react to the burial event (I forget the original paper where this has been said, so I just mention this can be the case).

Some well-preserved or enrolled specimens had similar patterns of dislocated librigenae in Whittington, 1971.

Whittington, 1971. Silurian calymenid trilobites from United States, Norway, and Sweden. *Palaeontology*, 14, 455-477.

Response: This is partly explained above. The position of the hypostome, librigenae and thorax are a result of the very early compression. We do not doubt it and we described it so. This taphonomical state was fixed in the "moment" soon after the trilobite was buried. Please note, that the compression has squeezed and moved the hypostome and very slightly also the librigenae. Other parts of the exoskeleton remained in place without signs of movement or displacement. It clearly illustrates a different state of articulation. The hypostome and librigenae were already weakly articulated in contrast to other parts of the exoskeleton. Such weakening of the articulation of certain parts that are detached first is a sign of an upcoming ecdysis, i.e. a premouling phase. If the compressing force was strong enough to detach a normally articulated exoskeleton, all of its parts would be affected in the same way. Another proof is the vaulting of the thorax between the 5th and 6th segment. This abnormality could not have been formed by the same compression force which affected the hypostome and librigenae. And why were other thoracic segments not affected? If we study the moulting patterns in trilobites this configuration reflects a typical primary phase when the ventral part (hypostome) and all the cephalic parts with doublure are getting detached. The rest of the exoskeleton is slightly vaulted as a prerequisite for one of the next stages when this area also becomes weakly articulated and the thorax breaks in half. So, that is why these are unique but very clear indications of not the moulting but the premouling stage.

Remark: Line 255

"Minor post-mortem deformation is manifested by the hypostome and gut displacements". Authors have stated the post-mortem deformation. How does the "partly loose librigenae" in line 244 correlate with the moulting? Why is the deformation only affecting the hypostome but not the librigenae? This statement contradicts the premouling theory.

Response: See above.

Remark: Lines 256-257

A "ligament" is suggested between the hypostome and anterior-ventral side of the cephalon. However, the evidence provided by the authors does not support the "ligament" hypothesis. Check the list of major issues at the end of the comments.

Forty (1990) had described three types of hypostome attachment: natant, conterminant, and impendent. The current one can be natant or conterminant. I don't insist authors use these terms as they are too specialized. Both papers below are helpful for understanding the feeding issues.

Fortey, 1990. Ontogeny, hypostome attachment and trilobite classification. *Palaeontology*, 33, 529-576.

Fortey and Owens 1999. Feeding habits in trilobites. *Palaeontology*, 42, 429-465.

Response: Yes, you are right. However, in line 40 we state that the hypostome is conterminant, using Fortey's (1990) terminology.

The incorporation of ligaments in articulation of arthropod exoskeletons is a basic feature. Thus, there is no discussion about its presence in trilobites (see for example Lerosey-Aubril, R., Hegna, T. A., & Olive, S. 2011. Inferring internal anatomy from the trilobite exoskeleton: the relationship between frontal auxiliary impressions and the digestive system. *Lethaia*, 44(2), 166-184). We can logically assume that some types of hypostomes were connected to the rostra by ligaments. It must have been the case with this type of hypostome because it was connected to the rostral margin (their corresponding marginal shapes fit together) and the connection was apparently relatively firm and supposedly flexible. The ligament is a general term for such an interconnecting tissue, widely used in literature. Our specimen even has remains of such structure preserved. The dislocation of the hypostome by external post mortem forces caused the deformation of the ligament. The deformation was not brittle, which supports our interpretation, as the ligaments are plastic in nature. The ligament tissue was stretched and torn, and this state was "frozen" in the nodule. Subsequently, the ligament decayed to form the crumble via the mechanism described above. As it was a very fine structure and the cavities filled with crumble were very small and indistinct, in the rendered images the ligament looks like "raindrops" scattered between the rostrum and the hypostome.

Because the crumble represents a product of tissue decay inside the nodule, the tissue stretched between the rostrum and the hypostome is a direct evidence of a connecting ligament tissue in trilobites.

Remark: Lines 267-272

Burrows are only restricted to the lobate anterior part of the glabella, in front of the ventriculi. Authors then argue that the burrows are located in front of the digestive system and thus avoid the decaying digestive tract. I understand that authors have used "crumble" to describe the digestive tract, but I find there may be alternative explanations about the relationship between "ligament" and the "crumble". Please check the possible explanations at the end of the comments.

Response: We highly appreciate the motivation for brainstorming induced by the referee's remarks. We carefully went through our arguments and re-examined all the data, photos and material we have available. And we hope that our explanations based on observations clearly justify our attitude. Remember, that the ligament is also preserved in form of "crumble". The "crumble" represents a mode of preservation of the decaying tissues, it is not used to describe the gut tissue only. The chapter was considerably rewritten.

Remark: Line 271

Antennal glands are suggested. We don't have any evidence for the position of antennal glands in any early arthropod. Even if modern arthropods can have the antennal glands there, we can't simply apply this in trilobites. In trilobites, we can use the muscle scars to discuss the position of muscles because muscle scars are observable. There is nothing left of the antennal glands in the fossil record, so it should be cautious about this. If this is purely for the discussion of enzymatic activity, it is overinterpreted by discussing the position and size of the glands.

Response: You are completely right that our argumentation here is not very robust. We modified the paragraph considerably.

Remark: Lines 283-286

This section suggests evidence of an inhospitable condition, which is then connected with the enzymatic activity. Authors have seen the trace fossils present dominantly in the frontal lobe of the glabella and other trunk pleural regions. There is an alternative explanation about this, please check the list of major issues below.

Many Cambrian arthropods or annelids have gut contents preserved with animal fragments. I list a few here. I don't deny the importance of the work done by authors. If the authors can provide confident evidence for the issues discussed in the paper, it is definitely an innovative thing for early arthropod studies. However, there are some issues that authors need to think about. Please check the list of major issues below.

I here list a few more papers that have discussed the gut contents.

1) Zhu et al. 2004. Direct evidence for predation on trilobites in the Cambrian. *Biology Letters*, 271, S277-S280.

2) Bruton 1981. The arthropod *Sidneyia inexpectans*, middle Cambrian, Burgess shale, British Columbia. *Phil Trans R Soc Lond*, 295, 619-653.

3) Vannier 2012. Gut contents as direct indicators for trophic relationships in the Cambrian marine ecosystem. *Plos One*, 7, e52200.

Response: Of course, we studied the references above during our research. Concerning ichnofossils, please note, that there is a complex study about feeding traces in the nodules of the Šárka Formation where different aspects are discussed (Kraft et al 2020).

Remark: I list a few major issues that authors may need to consider. As there is only one specimen for the study, I think the authors need to provide more convincing evidence.

Response: We think that even though the specimen is single it provides very robust evidence due to its exceptional preservation and the used imaging method. This is not an exceptional methodological case; many key discoveries were based on such single specimens.

Remark: (I) Dorsal and ventral ventriculi

The connection between the dorsal and ventral ventriculi is an important character but I don't see this evidence clearly in the figures. Authors have mentioned both ventriculi are close to either the dorsal glabella or the ventral hypostome, thus the lobe-like shapes of the ventriculi are affected by the rounded inner surfaces of both glabella and hypostome.

Response: See above.

Remark: (II) "Ligament" and "crumple"

The authors have used the "ligament" and "crumple" to describe the two separately located structures. There is another possibility that these two structures are continuous and connected with each other in life.

The evidence to challenge the "ligament" hypothesis is shown here. (1) Many small branches or filamentous extensions, like cotton with filaments connecting with each other, are likely not the component of a ligament that should be relatively regular in shape. (2) The coarse appearance and the high thickness are all against the

“ligament” hypothesis, especially considering the “ligament” is stretched (authors mentioned it in line 257). (3) The lateral fragment of the “ligament” (Extended Data Fig. 8b), separated from the centrally located “ligament”, is likely extended into the lateral border of the left librigena (Extended Data Fig. 8e2).

If the ligament is only connecting the hypostome and anterior border of the cephalon, why the ligament extends into the lateral border? (5) “Ligament” perfectly extends into the anterior and lateral borders of the cephalon, indicating a soft tissue with high flexibility. Does a ligament bear such high flexibility? (6) “Ligament” has a perfect shape that matches the inner surface of the frontal lobe of the glabella (Extended Data Fig. 8e2 and 8e3). Again, soft tissue is favored.

The “crumble” is the decaying tissue of the gut. I agree with this. However, the anterior portion of the crumble has many small branches or filamentous extensions, which are extremely similar to those in the posterior portion of the suggested “ligament”. I think both “ligament” and “crumble” are connected with each other because they share a similar pattern of filamentous extensions, indicating a complete gut tract.

Authors have mentioned that the hypostome is dislocated from the anterior position to the posterior or the ventral position, the hypostome likely squeezes or cuts off the structure that is located between the hypostome and glabella. As the force is performed by the anterior margin of the hypostome, the gut tract is then cut off rather than two different structures.

In short, the downward displacement of the hypostome has forced and cut the gut tract into two pieces, the anterior “ligament” and the posterior “crumble”.

If this is the case, the gut tract will have an expanded frontal crop (expanded anterior portion of the gut tract). By following this possible explanation, the suggested dorsal and ventral ventriculi do not exist as they are inside the wide crop. That is why the connecting segment between the two ventriculi is not clearly displayed.

Response: This is a point of misunderstanding. It should be stated again: the “crumble”, rendered in dark green, occurs in many places where soft tissue decayed in the forming nodule and the products of this process fossilized (see explanation above). In the Šárka Formation nodules, it often occurs between the dorsal part and the doublure of the trilobite exoskeleton where it is often accompanied by feeding traces (see Kraft et al. 2020). In this trilobite, the “crumble” is found only below some segments of the doublure. The “crumble” of the ligament is represented by small discrete cavities between the posterior margin of the rostrum and the anterior margin of the hypostome, as clearly labelled in Fig. 4. All remaining “crumble” was formed from other tissues.

Remark: (III) Why are traces clustered near the crop or foregut?

As the gut tube is broken, the nutrients released from the crop would supply the surrounding microbes and allow the organisms to feed near the broken region. This is nothing related to the enzymatic activities of the gut tract. Suggesting a favorable way for trace makers is relatively easy compared to suggesting an unfavorable way.

Response: The gut was not broken. If it had been, its content would be prolapsed when compressed, but this is not the case. The entire digestive tract is only deformed and slightly displaced. The 3D pdf that will be a part of the supplementary data shows this clearly.

Remark: The favorable way involves one hypothesis: nutrients are there. The unfavorable way involves two hypotheses: toxic condition and its producer: enzymes. If we follow the method of phylogenetic studies, one hypothesis is likely better.

Response: Of course, that nutrients were there. However, possible sources of nutrients occurred in all parts of the carcass. The enzymes continue to function after the death of the organism through the process of autolysis, which is a passive breakdown of necrotic tissues via the organism's own enzymes. This process naturally begins where enzymes are present in life, i.e. the digestive tract. The scavengers must have chemically sensed the trilobite carcass and started to approach it shortly after it died (because, for example, muscle tissue was still not decayed enough for the scavengers to target it), when the process of autolysis of the gut was still ongoing. Thus, they had to avoid this area to prevent being digested. The distinct selectivity of the trace makers targets, burrow courses and their terminations, all together frame the story not only about decay processes of the trilobite but also bring forward arguments supporting a fast formation of the nodule, reason for the existence of 'crumble' and to understanding the whole nodule as a snapshot. This is one of the keys to palaeophysiology.

Reviewer Reports on the Second Revision:

Referees' comments:

Referee #1 (Remarks to the Author):

no further comments

Referee #2 (Remarks to the Author):

My review remains the same as the previous one. The authors have addressed my comments and I look forward to seeing this published. I have no more comments, other than this should be published.

Referee #3 (Remarks to the Author):

To Authors,

I have carefully manipulated the 3D PDFs to observe the suggested ventriculi but failed to see the "short" connecting segment. The refined process of digestion by the sequence, the suggested ventral ventriculus -- connecting segment -- the dorsal ventriculus, is still a mixed state that is taphonomically collapsed together. Gut fragments preserved like this are common in early metazoans, rendering fragments out but lacking the support of two ventriculi can not make it a case uniquely distinguished from those examples already known.

I also want to admit that the 3D files are very impressive.

Those undebated remarks and responses have been removed from this round. Only the debated ones listed below.

Originality and significance:

The gut structure presented in this paper does not make a big difference from other known examples. Only rendering the fragments out is not that innovative because fragments like this are common in early animals. The interaction between the trace makers and the trilobite gut is a unique case.

Data & methodology:

Data quality is very good and the most advanced technique is also applied.

Appropriate use of statistics and treatment of uncertainties:

Good.

Conclusions:

As 3D data failed to support the suggested “ventriculi”, the precise process of palaeophysiology can not be summarized with confidence.

Suggested improvements:

A few concerns remained and listed below.

References:

Appropriate.

Remark: Lines 81-90

The authors have suggested “two sack-like ventriculi” and described them with a connection by a short, flat, and bent segment, which is secondarily dislocated to the left. However, I don’t see the “short, flat and bent segment” in the figures and can’t catch this important information. If authors make a line drawing of the outline of the segment and ventriculi in the figures, it would be very helpful.

Response: A line drawing would be problematic because the shapes of all parts of the digestive tract are inferred from the accumulations of discrete fragments inside them. Therefore, the drawing would only be an interpretation of the original shapes and would not provide more information than the rendered scan images. Perhaps a dotted 3D drawing would partly serve the purpose, but a drawing from the same or similar perspective, which seems to be the most instructive, would provide equal information as the reconstruction drawing in Fig. 5.

Some dotted lines can be marked in the Fig 2a for comparison. These lines are definitely not affecting the result and only serve for marking relative positions. Reconstruction is an art work is clearly different from the important dotted lines. With these lines, readers can trace where the possible connecting segment is.

Check comment below.

Remark: Authors are stating that the ventral ventriculus (line 90) is connected to the dorsal ventriculus by a short segment, then the dorsal ventriculus is connected to the midgut (line 91-92), and finally the hindgut (line 94). Based on this sequence, the morphology of the gut system is comparable to the modern examples (lines 105-109). This is one of the major arguments in the paper, but I can’t see the evidence with confidence. As the exoskeletal fragments are close to each other, the boundary between clusters, which authors have suggested, is not that clear. That is why I hope the authors can provide a line drawing of ventriculi.

Response: We understand that it can be difficult to see in 2D images, even though we did our best to select views in different angles. The referees can download a 3D pdf from <https://uppsala.box.com/s/n71426jvkbmg7ljaxh47nnl252e1ob2l> which should make the morphology

easier to understand. We have also modified the text for better clarity.

3D PDFs are very helpful. Check the comment below.

Remark: How is the boundary between the ventriculi (authors have argued) and the posterior gut tract different from the boundaries of clusters occurred in the posterior gut tract (Fig. 2a; a boundary in front of the cluster containing the last four blue pieces of fragments that are located at four corners)? Based on Fig. 2a2, the suggested ventriculi are on the same level and connected transversely. The separation between dorsal and ventral ventriculi is likely the taphonomic artifact because the gut is dislocated.

Response: The 3D pdf will help answering this question because the ventriculi are not on the same level. The gut is clearly a continuation of the dorsal ventriculus. We are not arguing that the boundary between the foregut and midgut differs from the boundaries in the hindgut. On the contrary, all of them are very faint, best spotted in motion.

The separation between the dorsal and ventral ventriculi cannot be a taphonomical effect. If those were originally a single structure, the post-mortem compression would lead to a homogenous mass pressed into the ventral surface of the glabella. Having two discrete accumulations of fragments in the head region is a direct indication of two structures separated by soft tissues in life.

By carefully manipulating the 3D PDFs, I can not find the evidence supporting the two ventriculi.

“The gut is clearly a continuation of the dorsal ventriculus”. The gut connects to not only the dorsal ventriculus but also the ventral ventriculus based on my observation. The question here is how long (sagittal) the connecting segment is. In Fig 2, the most distinct cluster of the fragments is about one fourth of the total gut length, and the first distinct boundary behind this cluster is clearly visible. The connecting segment stops nearly at the posterior end of this cluster (Fig. 2a2, 2a3), which is also reaching the posterior margin of the posterior border of the hypostome (3D pdf file). This is consistent with the transverse U-shape discussed by authors. Such a long connection is not a evidence for two ventriculi.

If authors continue to shorten the length of connecting segment upward (in Fig. 2a), the straight gut tract will connect clearly with both “ventriculi”. This will contradict with the statement of “The gut is clearly a continuation of the dorsal ventriculus”.

The dislocation (lines 93-95) has moved the connecting segment from transverse (Fig 5c; viewed from dorsal up) to longitudinal (sagittal; U-shape and Fig. 2a) direction is not likely because bending a structure in 90 degrees is a distinct deformation. I have tried to see if there is an angle about 45 degrees or somewhat similar for the connecting segment, I failed.

In trilobites, a triangular crop is located at the position where the suggested “two ventriculi” are located. If the crop is collapsed because of decay and then falls on the surface of glabella and hypostome (authors have discussed the gravitational collapse in line 635), it would become the current state. The gut tract connects to the compressed triangular crop becomes reasonable.

Remark: Fig. 3

The dorsal (glabellar) and ventral (hypostome) ventriculi are suggested. By looking at the fig 3, the ventral ventriculus has a curvature by following the inner concave surface of the hypostome (Fig. 3c), and the dorsal ventriculus has a curvature that follows the inner convex surface of the glabellar (Fig. 3b). I use “concave and convex” for the fig 3 to recognize the ventriculi’ morphology. As ventriculi are restricted by the inner surfaces of the glabella and hypostome, it is hard to make an argument that the dorsal and ventral ventriculi represent their original morphology. Authors need to distinguish why it is not part of taphonomy.

Response: See above, the text was modified.

Check comment above.

Remark: Lines 111-116

“However, the results of the analysis show a statistically insignificant difference (Line 111-112)”. This insignificant difference is interpreted as “Thus, food may have been ingested at speed, but digested at leisure, as documented in other fossil ecdysozoans (lines 115-116)”. This has been used to argue why the function between the suggested two ventriculi is not that efficient. This issue gets back to the comment directly above. If the suggested ventriculi did not exist, the inefficient digestion becomes more reasonable. How do authors distinguish the recognized detail of ventriculi from the taphonomic artifact?

Response: See above. There are two discrete clusters deformed by the hypostome dislocation but still existing.

Check comment above.

Remark: Lines 120-123

“Crumble” is limited to the axial part reflecting the decaying tissue of the gut. Authors have agreed it represents the decaying tissue of the gut, then it is understandable for the question below. The collapse of the decaying tissue would become closer to the trunk segments and likely cast on the inner surface of the trunk segments. Do authors agree on this?

Response: Please note that we describe a unique process which happened in a nodule that formed very quickly after the death of the trilobite. The nature of this process is unlike the usual decay processes we are used to in common facies, such as when the fossil is covered by a muddy substrate which will later form a shale. Two consecutive taphonomic processes took place: first a compression of the dead organism in a still soft substrate immediately after burial, and then the decay of tissues in an already forming nodule, i.e. in a solid matter produced by silicification of the muddy substrate at a very shallow depth below the sea bottom. There is not enough space in the manuscript to fully describe the process, but bear in mind that there is a substantial difference between the preservation in the nodules and the surrounding shales. After the formation of the nodule, there

were no further deformations or collapse of the cavities left after the decayed tissues. That is why the cavities are preserved and filled by the “crumble”, i.e. fossilized products of the decayed tissues due to bacterial activity. The tissue decay was happening in cavities entombed in solid matter. That is why their shape didn't change further and it reflects the original shape of the cavities at the time when the nodule was formed, which was apparently very early after the specimen was buried (which is also supported by the behaviour of the scavengers, see chapter Post-mortem events). This bit also explains some of your questions below.

It will be very good and helpful if authors can provide the details of “Two consecutive taphonomic processes took place: first a compression of the dead organism in a still soft substrate immediately after burial, and then the decay of tissues in an already forming nodule, i.e. in a solid matter produced by silicification of the muddy substrate at a very shallow depth below the sea bottom.” in the Taphonomic aspects of the Supplementary Information. This compensates the taphonomy of Burgess Shale-type fossils or will be another way to understand the BST fossils, increasing the broad effect of this study.

Remark: Line 125-129

Digestive caecae are introduced. Three pairs are located below the occipital ring and two thoracic segments. They are shown in the reconstructions (Fig. 5a). However the less extensive diverticulae in the posterior part of the trunk are not reconstructed. Making a distinguishment between the anterior and posterior diverticulae is likely needed.

Response: They are reconstructed. See the undulated outline of the gut in Fig. 5.

Please use one dark line to point one of the less extensive diverticulae as that one for the distinct diverticulum, or enlarge the less extensive diverticulae to be visible as they are quite clear in the 3D data.

Check my comment blow.

Remark: In Fig. 4c, the suggested digestive caecae or diverticulae are following the morphology of articulating half rings of the trunk segments. How do authors distinguish the diverticulae from taphonomy because of the collapse of the decaying gut? In 2D fossils, we can recognize the ink-like impressions to recognize the diverticulae. This method is clearly not working in 3D fossils studied herein. If authors can show the diverticulae lifted away from the trunk segment, this evidence would be more convincing. Otherwise, it is likely part of the collapse of the gut tissues but not diverticulae.

Response: As explained above, the decaying tissue had nowhere to expand. Everything was enclosed within solid, silicified sediment.

I have done my best to understand the 3D gut caecae by manipulating the 3D models, but it is really hard for me to correlate the rendered 3D gut caecae with that of other well-known early arthropods. Initially I found this type of preservation differs distinctly from the BST fossils (see the INITIAL COMMENT OF TAPHONOMY), but the difference is not that sharp. The crumble in the axial region is

likely the 3D morphology of the 2D ink impressions in BST fossils. Without the reference 13 cited in your paper in hand, I do not know whether the reference 13 has discussed this or not. If authors use one to two sentences to describe the relationship between the two, it will be more convenient for audience to understand why they have the appearance now and omits the confusion raised by taphonomy (INITIAL COMMENT OF TAPHONOMY). Otherwise directly making the argument that they are gut caecae is a little bit of abruptness.

Again, the preservation in nodule is somewhat similar to that BST type fossils at least at some stages, then the crop collapse issue mentioned above is very similar to the generally compressed crop possibly consisting of the mouth, oesophagus and crop.

INITIAL COMMENT OF TAPHONOMY (supplementary explanation only)

(1) "That is why the cavities are preserved and filled by the "crumble", i.e. fossilized products of the decayed tissues due to bacterial activity." Authors particularly consider that the "crumble" in the axial region of the exoskeleton is the decaying tissue of the gut (Line 124-125).

(2) Lines 632-637 in Supplementary Information.

Authors have clearly stated that "The dislocation of the hypostome and the intestine pressed to the internal surface of the exoskeleton...caused by gravitational collapse..." in the Taphonomic aspects. For this situation, the muscles, gut, nerve cord, and the tissues on the internal surface of the exoskeleton are all likely pressed to the internal surface of the exoskeleton.

The crumble in the axial region of the exoskeleton is likely not the only remain of the "decaying tissue of the gut" mentioned in point (1), it can be remains of muscles or other tissues. I say this because the outline of the crumble in the axial region does not show a very clear shape that is comparable to the well-known gut caecae in early arthropods. What I have mentioned in previous remarks is corresponding to this issue.

I make up an example here. Gut caeca with a box shape can leave a box-like empty space. Muscle surrounding the gut caeca can also leave another box-like empty shape. However, both two empty boxes are cutting with each other and thus show an outline that is the combination of gut caeca and muscle.

This is the concern I had in understanding the taphonomy if the material in nodule is distinctly different from the general BST fossils.

Remark: Lines 243-247

Based on the slightly vaulted trunk and partly loosen librigenae, authors suggest it is a premoult phase. This is hard to follow. As the head is distinctly convex and leaves the librigenae in a bedding plane lacking the support of the cranium, the lateral force performed during the preservation would likely give a shear force existed between the librigenae and cranium, resulting in the

displacement of the librigenae. This isn't evidence for moulting or premoulting stage especially when the soft tissues are preserved. Do authors see any part of the exoskeleton that shows double exoskeletal thickness? If it is in a stage that is so close to the moulting, the exoskeleton should be thicker because a new exoskeleton is formed below the old exoskeleton. The vaulting trunk is still the taphonomic artifact or it may represent that animals react to the burial event (I forget the original paper where this has been said, so I just mention this can be the case).

Some well-preserved or enrolled specimens had similar patterns of dislocated librigenae in Whittington, 1971.

Whittington, 1971. Silurian calymenid trilobites from united states, Norway, and Sweden. *Palaeontology*, 14, 455-477.

Response: This is partly explained above. The position of the hypostome, librigenae and thorax are a result of the very early compression. We do not doubt it and we described it so. This taphonomical state was fixed in the "moment" soon after the trilobite was buried. Please note, that the compression has squeezed and moved the hypostome and very slightly also the librigenae. Other parts of the exoskeleton remained in place without signs of movement or displacement. It clearly illustrates a different state of articulation. The hypostome and librigenae were already weakly articulated in contrast to other parts of the exoskeleton. Such weakening of the articulation of certain parts that are detached first is a sign of an upcoming ecdysis, i.e. a premoulting phase. If the compressing force was strong enough to detach a normally articulated exoskeleton, all of its parts would be affected in the same way. Another proof is the vaulting of the thorax between the 5th and 6th segment. This abnormality could not have been formed by the same compression force which affected the hypostome and librigenae. And why were other thoracic segments not affected? If we study the moulting patterns in trilobites this configuration reflects a typical primary phase when the ventral part (hypostome) and all the cephalic parts with doublure are getting detached. The rest of the exoskeleton is slightly vaulted as a prerequisite for one of the next stages when this area also becomes weakly articulated and the thorax breaks in half. So, that is why these are unique but very clear indications of not the moulting but the premoulting stage.

Deformation is common along the anteroposterior axis (or transverse articulations between thoracic segments) because thoracic segments are articulated along the transverse axis. In addition, the thoracic segment has a gentle slope toward the pleural spine, the force can be shifted and we can not see the clear evidence of deformation of the thoracic segment.

As there are still some difficulties, or debates, of recognizing the moulting-related phases and authors have cautiously argued this issue in the text, I am OK to this.

For this issue, I personally agree with referee #1.

Remark: (I) Dorsal and ventral ventriculi

The connection between the dorsal and ventral ventriculi is an important character but I don't see

this evidence clearly in the figures. Authors have mentioned both ventriculi are close to either the dorsal glabella or the ventral hypostome, thus the lobe-like shapes of the ventriculi are affected by the rounded inner surfaces of both glabella and hypostome.

Response: See above.

Check my comment above.

Author Rebuttals to Second Revision:

Response to referee no.3

To avoid confusion, the most recent remarks and responses are labelled in capital letters. We only responded to concrete remarks not to statements.

***REMARK:** I have carefully manipulated the 3D PDFs to observe the suggested ventriculi but failed to see the “short” connecting segment. The refined process of digestion by the sequence, the suggested ventral ventriculus -- connecting segment -- the dorsal ventriculus, is still a mixed state that is taphonomically collapsed together. Gut fragments preserved like this are common in early metazoans, rendering fragments out but lacking the support of two ventriculi can not make it a case uniquely distinguished from those examples already known.*

RESPONSE: The word “short” was removed from the text as it seems to confuse the readers. Our reasoning for the usage of the word “short” was in the dorsoventral sense not the anteroposterior.

We do not agree that “gut fragments preserved like this are common in early metazoans“. 3D gut fills are rare and any remains of food contents are unique. On the other hand, 2D imprints of guts are much more frequent. However, they lack the ability to reflect the original topologies of structures in space. Our specimen allowed us to reliably reconstruct the shape of the digestive tract, even though it is slightly but primarily deformed. This is something a fossil preserved in 2D could never provide, e.g. that even if there were two ventriculi present they would have been obscured by the taphonomical flattening. Therefore, anatomical comparisons between 2D and 3D preserved fossils are limited. Our observations are described and documented in figures, the reconstructed shape is supported by the character of the deformation in the very early diagenetic nodule. For arguments on the formation of nodules we added the reference to Kraft and Bruthansová 2023 to the Supplementary information text, more below.

I also want to admit that the 3D files are very impressive.

Those undebated remarks and responses have been removed from this round. Only the debated ones are listed below.

Originality and significance:

The gut structure presented in this paper does not make a big difference from other known examples. Only rendering the fragments out is not that innovative because fragments like this are common in early animals. The interaction between the trace makers and the trilobite gut is a unique case.

Data & methodology:

Data quality is very good and the most advanced technique is also applied.

Appropriate use of statistics and treatment of uncertainties:

Good.

Conclusions:

As 3D data failed to support the suggested “ventriculi”, the precise process of palaeophysiology can not be summarized with confidence.

Suggested improvements:

A few concerns remained and listed below.

References:

Appropriate.

Remark: Lines 81-90

The authors have suggested “two sack-like ventriculi” and described them with a connection by a short, flat, and bent segment, which is secondarily dislocated to the left. However, I don’t see the “short, flat and bent segment” in the figures and can’t catch this important information. If authors make a line drawing of the outline of the segment and ventriculi in the figures, it would be very helpful.

Response: A line drawing would be problematic because the shapes of all parts of the digestive tract are inferred from the accumulations of discrete fragments inside them. Therefore, the drawing would only be an interpretation of the original shapes and would not provide more information than the rendered scan images. Perhaps a dotted 3D drawing would partly serve the purpose, but a drawing from the same or similar perspective, which seems to be the most instructive, would provide equal information as the reconstruction drawing in Fig. 5.

REMARK: *Some dotted lines can be marked in the Fig 2a for comparison. These lines are definitely not affecting the result and only serve for marking relative positions. Reconstruction is an art work is clearly different from the important dotted lines. With these lines, readers can trace where the possible connecting segment is.*

Check comment below.

RESPONSE: Fig.2a is not the right one to demonstrate the two, clearly separated ventriculi, because those are dorsal, ventral and lateral views. As this part of the digestive tract is deformed by an obliquely squeezed hypostome an oblique view is much better. Such a view occurs in other figures, most of them already referred in the text, we added a few more references. We also decided to add a 3d pdf file among the supplementary data to allow the reader to check our observations.

Remark: *Authors are stating that the ventral ventriculus (line 90) is connected to the dorsal ventriculus by a short segment, then the dorsal ventriculus is connected to the midgut (line 91-92), and finally the hindgut (line 94). Based on this sequence, the morphology of the gut system is comparable to the modern examples (lines 105-109). This is one of the major arguments in the paper, but I can't see the evidence with confidence. As the exoskeletal fragments are close to each other, the boundary between clusters, which authors have suggested, is not that clear. That is why I hope the authors can provide a line drawing of ventriculi.*

Response: We understand that it can be difficult to see in 2D images, even though we did our best to select views in different angles. The referees can download a 3D pdf from <https://uppsala.box.com/s/n71426jvkimg7ljxh47nnl252e1ob2l> which should make the morphology easier to understand. We have also modified the text for better clarity.

3D PDFs are very helpful. Check the comment below.

Remark: How is the boundary between the ventriculi (authors have argued) and the posterior gut tract different from the boundaries of clusters occurred in the posterior gut tract (Fig. 2a; a boundary in front of the cluster containing the last four blue pieces of fragments that are located at four corners)? Based on Fig. 2a2, the suggested ventriculi are on the same level and connected transversely. The separation between dorsal and ventral ventriculi is likely the taphonomic artifact because the gut is dislocated.

Response: The 3D pdf will help answering this question because the ventriculi are not on the same level. The gut is clearly a continuation of the dorsal ventriculus. We are not arguing that the boundary between the foregut and midgut differs from the boundaries in the hindgut. On the contrary, all of them are very faint, best spotted in motion.

The separation between the dorsal and ventral ventriculi cannot be a taphonomical effect. If those were originally a single structure, the post-mortem compression would lead to a homogenous mass pressed into the ventral surface of the glabella. Having two discrete accumulations of fragments in the head region is a direct indication of two structures separated by soft tissues in life.

REMARK: By carefully manipulating the 3D PDFs, I can not find the evidence supporting the two ventriculi.

RESPONSE: Please note that the deformation has to be considered in your observation. Having this in mind, the two separate clusters of shell fragments interconnected by a displaced segment marked also by shell fragments become well visible. This is not observable in the anterior/posterior, dorsal/ventral or lateral views as the connecting segment was pushed from the anterior to an antero-lateral position. which can also explain your reservations below.

REMARK: “The gut is clearly a continuation of the dorsal ventriculus”. The gut connects to not only the dorsal ventriculus but also the ventral ventriculus based on my observation. The question here is how long (sagittal) the connecting segment is. In Fig 2, the most distinct cluster of the fragments is about one fourth of the total gut length, and the first distinct boundary behind this cluster is clearly visible. The connecting segment stops nearly at the posterior end of this cluster (Fig. 2a2, 2a3), which is also reaching the posterior margin of the posterior border of the hypostome (3D pdf file). This is consistent with the transverse U-shape discussed by authors. Such a long connection is not a evidence for two ventriculi.

RESPONSE: As stated above, Fig. 2 is not suitable for this observation. Better inspect Fig. 3d. This figure provides a better impression about the topology of clusters in relation to the cephalon and the hypostome. It also clearly documents that there is no connection between the ventral ventriculus and the gut.

REMARK: *If authors continue to shorten the length of connecting segment upward (in Fig. 2a), the straight gut tract will connect clearly with both “ventriculi”. This will contradict with the statement of “The gut is clearly a continuation of the dorsal ventriculus”.*

RESPONSE: See the previous explanation. We don't fully understand your comment “if authors continue to shorten the length of connecting segment upward...”

REMARK: *The dislocation (lines 93-95) has moved the connecting segment from transverse (Fig 5c; viewed from dorsal up) to longitudinal (sagittal; U-shape and Fig. 2a) direction is not likely because bending a structure in 90 degrees is a distinct deformation. I have tried to see if there is an angle about 45 degrees or somewhat similar for the connecting segment, I failed.*

RESPONSE: We do not fully understand your argument. The deformation which affected the hypostome happened in an upside-down position of the trilobite carcass. A dextral pressure disoriented the connecting segment and a dorsal pressure (i.e. on the ventral side of the carcass) squashed the contents (shell remains) sinistro-posteriorly. A combination of these two movements affected the stretchable walls of the digestive tract resulting in a seemingly axial connecting segment. This is our explanation based on the configuration of the shell remains in the connecting segment. However, as it is more speculative than provable we avoided to discuss it in the text where we only described the observable features.

REMARK: *In trilobites, a triangular crop is located at the position where the suggested “two ventriculi” are located. If the crop is collapsed because of decay and then falls on the surface of glabella and hypostome (authors have discussed the gravitational collapse in line 635), it would become the current state. The gut tract connects to the compressed triangular crop becomes reasonable.*

RESPONSE: It could not have worked that way. In a gravitational collapse the crop contents could not end up in two distinct clearly and widely separated clusters. If one of them would logically lie on the bottom of the crop, the other one would have to be “hanging” from its “roof”. If all the shell remains were originally enclosed in a single large

crop, all of its contents would gravitationally fall down into the glabellar space. There would be no dorsal cluster of remains attached to the hypostome nor a gap between the two clusters. You may argue that the U shape could have been formed by gravitational collapse in an obliquely or perpendicularly buried specimen. But that is not the case, because the precise orientation of the trilobite carapace on its dorsal side can be inferred from the intestine contents distributed evenly along the entire width of the axial lobe and not accumulated on one side. In addition, there was no further collapse after the decay processes happening in the forming nodule (see arguments based on burrows, and Kraft and Bruthansová 2023).

Remark: Fig. 3

The dorsal (glabellar) and ventral (hypostome) ventriculi are suggested. By looking at the fig 3, the ventral ventriculus has a curvature by following the inner concave surface of the hypostome (Fig. 3c), and the dorsal ventriculus has a curvature that follows the inner convex surface of the glabellar (Fig. 3b). I use “concave and convex” for the fig 3 to recognize the ventriculi’ morphology. As ventriculi are restricted by the inner surfaces of the glabella and hypostome, it is hard to make an argument that the dorsal and ventral ventriculi represent their original morphology. Authors need to distinguish why it is not part of taphonomy.

Response: See above, the text was modified.

Check comment above.

Remark: Lines 111-116

“However, the results of the analysis show a statistically insignificant difference (Line 111-112)”. This insignificant difference is interpreted as “Thus, food may have been ingested at speed, but digested at leisure, as documented in other fossil ecdysozoans (lines 115-116)”. This has been used to argue why the function between the suggested two ventriculi is not that efficient. This issue gets back to the comment directly above. If the suggested ventriculi did not exist, the inefficient digestion becomes more reasonable. How do authors distinguish the recognized detail of ventriculi from the taphonomic artifact?

Response: See above. There are two discrete clusters deformed by the hypostome dislocation but still existing.

Check comment above.

Remark: Lines 120-123

“Crumble” is limited to the axial part reflecting the decaying tissue of the gut. Authors have agreed it represents the decaying tissue of the gut, then it is understandable for the question below. The collapse of the decaying tissue would become closer to the trunk segments and likely cast on the inner surface of the trunk segments. Do authors agree on this?

Response: Please note that we describe a unique process which happened in a nodule that formed very quickly after the death of the trilobite. The nature of this process is unlike the usual decay processes we are used to in common facies, such as when the fossil is covered by a muddy substrate which will later form a shale. Two consecutive taphonomic processes took place: first a compression of the dead organism in a still soft substrate immediately after burial, and then the decay of tissues in an already forming nodule, i.e. in a solid matter produced by silicification of the muddy substrate at a very shallow depth below the sea bottom. There is not enough space in the manuscript to fully describe the process, but bear in mind that there is a substantial difference between the preservation in the nodules and the surrounding shales. After the formation of the nodule, there were no further deformations or collapse of the cavities left after the decayed tissues. That is why the cavities are preserved and filled by the “crumble”, i.e. fossilized products of the decayed tissues due to bacterial activity. The tissue decay was happening in cavities entombed in solid matter. That is why their shape didn't change further and it reflects the original shape of the cavities at the time when the nodule was formed, which was apparently very early after the specimen was buried (which is also supported by the behaviour of the scavengers, see chapter Post-mortem events). This bit also explains some of your questions below.

REMARK: *It will be very good and helpful if authors can provide the details of “Two consecutive taphonomic processes took place: first a compression of the dead organism in a still soft substrate immediately after burial, and then the decay of tissues in an already forming nodule, i.e. in a solid matter produced by silicification of the muddy substrate at a very shallow depth below the sea bottom.” in the Taphonomic aspects of the Supplementary Information. This compensates the taphonomy of Burgess Shale-type fossils or will be another way to understand the BST fossils, increasing the broad effect of this study.*

RESPONSE: We modified our explanation from the previous response to the referee (above) and used it along with the reference (see below) to expand the Supplementary Information text.

This type of preservation cannot be compared to the Burgess Shale-type preservation as it is in full 3D without any secondary deformation caused by diagenetic and post diagenetic processes. In fact, this type of preservation is currently being described in a new study by the first author and colleagues. Its submission was postponed because we are waiting for the present manuscript to be accepted, so we can cite it and also keep its priority. Anyway, the preliminary results of this study were presented in the 14th isos in estonia this year and published in an abstract: Kraft,P., Bruthansová,j.2023. Preservation of fossils in the Šárka Formation (Darriwilian,Czech Republic). Estonian Journal of Earth Sciences,72(1),136.doi 10.3176/earth.2023.59

Remark: Line 125-129

Digestive caecae are introduced. Three pairs are located below the occipital ring and two thoracic segments. They are shown in the reconstructions (Fig. 5a). However the less extensive diverticulae in the posterior part of the trunk are not reconstructed. Making a distinguishment between the anterior and posterior diverticulae is likely needed.

Response: They are reconstructed. See the undulated outline of the gut in Fig. 5.

REMARK: Please use one dark line to point one of the less extensive diverticulae as that one for the distinct diverticulum, or enlarge the less extensive diverticulae to be visible as they are quite clear in the 3D data.

Check my comment blow.

RESPONSE: We consider such a label as misleading. You are right, that some structures, which we describe as "midgut glands or other less extensive diverticula" in the text, are clearly visible in the renders and the 3D data. However, in related trilobites typically three pairs of diverticula are present. Therefore, we refrain from defining the structures posterior to them, because we simply don't have an analogy available. We feel obliged to reconstruct them in Fig. 5, but we decided not to specify their function further nor label them, as that could be considered speculative.

Remark: In Fig. 4c, the suggested digestive caecae or diverticulae are following the morphology of articulating half rings of the trunk segments. How do authors distinguish

the diverticulae from taphonomy because of the collapse of the decaying gut? In 2D fossils, we can recognize the ink-like impressions to recognize the diverticulae. This method is clearly not working in 3D fossils studied herein. If authors can show the diverticulae lifted away from the trunk segment, this evidence would be more convincing. Otherwise, it is likely part of the collapse of the gut tissues but not diverticulae.

Response: As explained above, the decaying tissue had nowhere to expand. Everything was enclosed within solid, silicified sediment.

REMARK: *I have done my best to understand the 3D gut caecae by manipulating the 3D models, but it is really hard for me to correlate the rendered 3D gut caecae with that of other well-known early arthropods. Initially I found this type of preservation differs distinctly from the BST fossils (see the INITIAL COMMENT OF TAPHONOMY), but the difference is not that sharp. The crumble in the axial region is likely the 3D morphology of the 2D ink impressions in BST fossils. Without the reference 13 cited in your paper in hand, I do not know whether the reference 13 has discussed this or not. If authors use one to two sentences to describe the relationship between the two, it will be more convenient for audience to understand why they have the appearance now and omits the confusion raised by taphonomy (INITIAL COMMENT OF TAPHONOMY). Otherwise directly making the argument that they are gut caecae is a little bit of abruptness.*

RESPONSE: For a better understanding of the crumble and its related aspects we added an explanation to the Supplementary Information text, as stated above.

REMARK: *Again, the preservation in nodule is somewhat similar to that BST type fossils at least at some stages, then the crop collapse issue mentioned above is very similar to the generally compressed crop possibly consisting of the mouth, oesophagus and crop.*

RESPONSE: No, the BST and the Šárka nodule types of preservation are diametrically different. The comparison of these two types of preservation is misleading. For the compression see above.

REMARK: **INITIAL COMMENT OF TAPHONOMY (supplementary explanation only)**

(1) *“That is why the cavities are preserved and filled by the “crumble”, i.e. fossilized products of the decayed tissues due to bacterial activity.” Authors particularly consider that the “crumble” in the axial region of the exoskeleton is the decaying tissue of the gut (Line 124-125).*

(2) Lines 632-637 in Supplementary Information.

Authors have clearly stated that “The dislocation of the hypostome and the intestine pressed to the internal surface of the exoskeleton...caused by gravitational collapse...” in the Taphonomic aspects. For this situation, the muscles, gut, nerve cord, and the tissues on the internal surface of the exoskeleton are all likely pressed to the internal surface of the exoskeleton.

The crumble in the axial region of the exoskeleton is likely not the only remain of the “decaying tissue of the gut” mentioned in point (1), it can be remains of muscles or other tissues. I say this because the outline of the crumble in the axial region does not show a very clear shape that is comparable to the well-known gut cecae in early arthropods. What I have mentioned in previous remarks is corresponding to this issue.

I make up an example here. Gut ceca with a box shape can leave a box-like empty space. Muscle surrounding the gut ceca can also leave another box-like empty shape. However, both two empty boxes are cutting with each other and thus show an outline that is the combination of gut ceca and muscle.

This is the concern I had in understanding the taphonomy if the material in nodule is distinctly different from the general BST fossils.

RESPONSE: The crumble occurs in places where the tissue decaying by bacterial activity was situated during the formation the nodule. It is a highly selective feature. For example, legs, muscles, epidermis and many other types of tissues are also not preserved.

Remark: Lines 243-247

Based on the slightly vaulted trunk and partly loosen librigenae, authors suggest it is a premoult phase. This is hard to follow. As the head is distinctly convex and leaves the librigenae in a bedding plane lacking the support of the cranidium, the lateral force performed during the preservation would likely give a shear force existed between the librigenae and cranidium, resulting in the displacement of the librigenae. This isn't evidence for moulting or premoult stage especially when the soft tissues are preserved. Do authors see any part of the exoskeleton that shows double exoskeletal thickness? If it is in a stage that is so close to the moulting, the exoskeleton should be thicker because a new exoskeleton is formed below the old exoskeleton. The vaulting trunk is still the taphonomic artifact or it may represent that animals react to the burial

event (I forget the original paper where this has been said, so I just mention this can be the case).

Some well-preserved or enrolled specimens had similar patterns of dislocated librigenae in Whittington, 1971.

Whittington, 1971. Silurian calymenid trilobites from united states, Norway, and Sweden. Palaeontology, 14, 455-477.

Response: This is partly explained above. The position of the hypostome, librigenae and thorax are a result of the very early compression. We do not doubt it and we described it so. This taphonomical state was fixed in the “moment” soon after the trilobite was buried. Please note, that the compression has squeezed and moved the hypostome and very slightly also the librigenae. Other parts of the exoskeleton remained in place without signs of movement or displacement. It clearly illustrates a different state of articulation. The hypostome and librigenae were already weakly articulated in contrast to other parts of the exoskeleton. Such weakening of the articulation of certain parts that are detached first is a sign of an upcoming ecdysis, i.e. a premouling phase. If the compressing force was strong enough to detach a normally articulated exoskeleton, all of its parts would be affected in the same way. Another proof is the vaulting of the thorax between the 5th and 6th segment. This abnormality could not have been formed by the same compression force which affected the hypostome and librigenae. And why were other thoracic segments not affected? If we study the moulting patterns in trilobites this configuration reflects a typical primary phase when the ventral part (hypostome) and all the cephalic parts with doublure are getting detached. The rest of the exoskeleton is slightly vaulted as a prerequisite for one of the next stages when this area also becomes weakly articulated and the thorax breaks in half. So, that is why these are unique but very clear indications of not the moulting but the premouling stage.

REMARK: *Deformation is common along the anteroposterior axis (or transverse articulations between thoracic segments) because thoracic segments are articulated along the transverse axis. In addition, the thoracic segment has a gentle slope toward the pleural spine, the force can be shifted and we can not see the clear evidence of deformation of the thoracic segment.*

As there are still some difficulties, or debates, of recognizing the moulting-related phases and authors have cautiously argued this issue in the text, I am OK to this.

For this issue, I personally agree with referee #1.

RESPONSE: Thank you! We describe what we observe, but remain careful in our interpretations.

Remark: *(I) Dorsal and ventral ventriculi*

The connection between the dorsal and ventral ventriculi is an important character but I don't see this evidence clearly in the figures. Authors have mentioned both ventriculi are close to either the dorsal glabella or the ventral hypostome, thus the lobe-like shapes of the ventriculi are affected by the rounded inner surfaces of both glabella and hypostome.

Response: See above.

Check my comment above.